# Laminar differences in decision-related neural activity in dorsal premotor cortex

Chandramouli Chandrasekaran[1], Diogo Peixoto [2,3], William T. Newsome[2,4,5] & Krishna V. Shenoy[1,2,4,5,6,7]

Dorsal premotor cortex is implicated in somatomotor decisions. However, we do not understand the temporal patterns and laminar organization of decision-related firing rates in dorsal premotor cortex. We recorded neurons from dorsal premotor cortex of monkeys performing a visual discrimination task with reaches as the behavioral report. We show that these neurons can be organized along a bidirectional visuomotor continuum based on task-related firing rates. "Increased" neurons at one end of the continuum increased their firing rates ~150 ms after stimulus onset and these firing rates covaried systematically with choice, stimulus difficulty, and reaction time—characteristics of a candidate decision variable. "Decreased" neurons at the other end of the continuum reduced their firing rate after stimulus onset, while "perimovement" neurons at the center of the continuum responded only ~150 ms before movement initiation. These neurons did not show decision variable-like characteristics. "Increased" neurons were more prevalent in superficial layers of dorsal premotor cortex; deeper layers contained more "decreased" and "perimovement" neurons. These results suggest a laminar organization for decision-related responses in dorsal premotor cortex.

[1] Department of Electrical Engineering, Stanford University, Stanford, CA 94305, USA. [2] Department of Neurobiology, Stanford University, Stanford, CA 94305, USA. [3] Champalimaud Neuroscience Programme, Avenida Brasília, Lisbon 1400-038, Portugal. [4] Howard Hughes Medical Institute, Stanford University, Stanford, CA 94305, USA. [5] Stanford Neurosciences Institute, Stanford University, Stanford, CA 94305, USA. [6] Department of Bioengineering, Stanford University, Stanford, CA 94305, USA. [7] Bio-X Program, Stanford University, Stanford, CA 94305, USA. Correspondence and requests for materials should be addressed to C.C. (email: mouli@stanford.edu) or to K.V.S. (email: shenoy@stanford.edu)

I
magine coming to a stoplight in your car. The sight of the stoplight results in patterns of neural activity that lead you to press the brakes to stop the car. This process of choosing and performing appropriate actions based on sensory inputs is termed perceptual decision-making and in this scenario involves the somatomotor system of the brain[1–5]. Lesion experiments in monkeys[6], clinical case studies of human patients[7], and physiological studies in monkeys using reach-target selection[8, 9] and perceptual decision-making tasks[10, 11] all suggest that somatomotor decision-related signals are present in dorsal premotor cortex (PMd)[1]. Despite these important studies, many questions about neural circuit dynamics in PMd during somatomotor decisions remain unresolved. In this study, we address two of these questions.

First, we lack a detailed description of the temporal patterns of firing rates (FRs) in PMd during perceptual decisions. An understanding of the temporal patterns of FRs is an important first step for building mechanistic models of decision-making[12–15]. Classical examinations of motor structures including PMd using task designs that involve instructed delays[16–21] and recent studies focused on establishing a role for PMd in reach target selection[8] or perceptual decision-making[10, 11] have established that FR patterns demonstrate substantial temporal variation or "complexity"[22], consistent with other reports of heterogeneous, temporally complex FRs in several brain regions during perceptual decision-making tasks[14, 23–27]. By temporal variation, we mean that PMd FRs exhibit mixed responses to the visual stimulus and movement onset that can involve both increases and decreases in FRs[8, 28]. One hypothesis proposed in studies of the frontal eye field is that these types of FRs can be viewed as organized along a visuomotor continuum[29–35]. Here, our first goal was to further investigate the temporal variation in PMd FRs during perceptual decisions and examine if they are also well described by a visuomotor continuum[29]. In particular, we wanted to examine how FR patterns of cells in different parts of the continuum covary with sensory parameters (e.g., stimulus difficulty), behavioral markers of task performance (e.g., reaction time (RT), choice), and temporal events (e.g., stimulus onset, movement onset) and thereby delineate putative decision-related neurons. To elicit a broad range of decision-related FR patterns that we wished to investigate, we used a variant of a recently reported RT static visual checkerboard discrimination task[11].

Our second goal was to ask whether putative decision-related neurons tend to be anatomically localized, such that anatomically informed neural network models can be created[13–15, 30, 36–38]. Anatomical studies suggest that superficial layers of PMd receive substantial corticocortical input from frontal association areas[39], and connectivity studies in the rodent motor cortex suggest a prominent descending projection from superficial to deep layers[40]. Therefore, we investigated the hypothesis that there are differences in the FR dynamics in the superficial layers compared to the deeper layers of PMd during the process of decision formation. To study this question, we used multi-contact electrodes that provide recordings across the different layers of the cortex.

We found that FR patterns in PMd demonstrated substantial temporal variation during the decision-formation process. This variation could be readily viewed along a systematic visuomotor continuum[9, 29, 33]. This conclusion was supported by a variety of analyses ranging from simple indices computed on a single neuron basis to methods such as principal components and K-means clustering that exploit covariance structure in the neural population[26]. The activity of PMd neurons that increased their FRs after visual stimulus onset had the strongest and earliest covariation with stimulus difficulty, choice, and RT—consistent with the characteristics of a candidate decision variable predicted by decision-making models[41–44]. Finally, using relative depth

information we found that there are more putative decision-related neurons in superficial layers of PMd, thus providing a constraint for anatomically informed neural network models[14, 30, 37].

## Results

**Monkeys discriminate visual checkerboard cues.** Two trained monkeys (T and O) discriminated the dominant color of a central, static checkerboard cue composed of mixtures of red and green squares and used an arm movement to report the decision (Fig. 1a[11]). The use of a static checkerboard cue meant that the sensory stimulus itself had no temporal dynamics, and all sensory evidence was provided at the onset of the stimulus and did not vary over time. Figure 1b depicts a trial timeline. The trial began when the monkey touched the center target and fixated on the cross. After a variable target viewing period, the red-green checkerboard cue appeared. The task of the monkey was to make an arm movement to the target (red vs. green) that matched the dominant color of the checkerboard cue. We parameterized difficulty of the discrimination (Fig. 1c) by a color coherence measure (C) defined as the absolute difference in the number of red and green squares normalized by the total number of squares in the checkerboard ($100 \times |R - G|/(R + G)$). A corresponding signed color coherence measure (SC) is defined as $100 \times (R - G)/(R + G)$.

We first examined the effects of changes in color coherence on the behavior of the monkeys. On average across sessions, decreases in color coherence (example stimuli shown in Fig. 1c) resulted in more errors (Fig. 1d, fitting the proportion correct as a function of coherence (C) using a psychometric function; average $R^2$, T: 0.99 (over 75 sessions), O: 0.99 (over 66 sessions), slope ($\beta$), M $\pm$ SD over sessions, T: $1.30 \pm 0.16$, O: $1.22 \pm 0.16$). In Fig. 1d, the measured data points (mean across sessions) are shown as *gray circles* with $2 \times$ SEM. *Black lines* are drawn in between these data points to guide the eye. Monkey T was more sensitive than monkey O (thresholds ($\alpha$) are computed on a per session basis at 81.6% correct (M $\pm$ SD): T, $10.77 \pm 1.26\%$, O: $15.42 \pm 1.87\%$, unpaired $t_{140} = -17.65$, $p = 2.67e-37$, two-tailed test, Wilcoxon rank sum comparing median thresholds, $p = 2.91e-23$).

Decreases in color coherence also resulted in slower RTs (Fig. 1e, using a regression to test if mean RT increases as $\log_e$ coherence decreases (harder stimulus difficulties) as in ref. [45]; average $R^2$, T: 0.94, O: 0.65; slope of regression: M $\pm$ SD over sessions, T: $-42.1 \pm 8.1$ ms/$\log_e$ coherence (%), O: $-8.6 \pm 4.5$ ms/$\log_e$ coherence (%)). Again in Fig. 1e, the measured data points (mean across sessions) are shown in *gray circles* with $2 \times$ SEM and to guide the eye, *black lines* are drawn in between the points. Monkey T had a larger range of RTs compared to monkey O (comparing the RT range between easiest and hardest difficulties (M $\pm$ SD) estimated over sessions; T: $143 \pm 32$ ms and O: $52 \pm 17$ ms, Wilcoxon ranksum comparing median ranges of RT, $p = 4.85e-24$).

We also investigated the behavior of the monkeys by fitting the RT distributions and accuracy using the drift diffusion (DDM) and urgency gating models (UGM) developed to explain behavior in two alternative forced choice tasks[46–48] (Supplementary Note 1, Supplementary Figs. 1, 2 and Supplementary Tables 1–4). We performed this model-fitting analysis to identify if these candidate computational frameworks could help us interpret decision-related responses in PMd, and if the behavior was better explained by the DDM, estimate decision times for the monkeys. Quantitative modeling of how monkeys perform discrimination of static stimuli such as the checkerboard used here is lacking[47]. Both the UGM and the DDM provided reasonable fits. However, neither model was completely sufficient to describe the RT and accuracy of the monkeys (Supplementary Figs. 1, 2 and Supplementary Note 1). The UGM with an intercept and slope

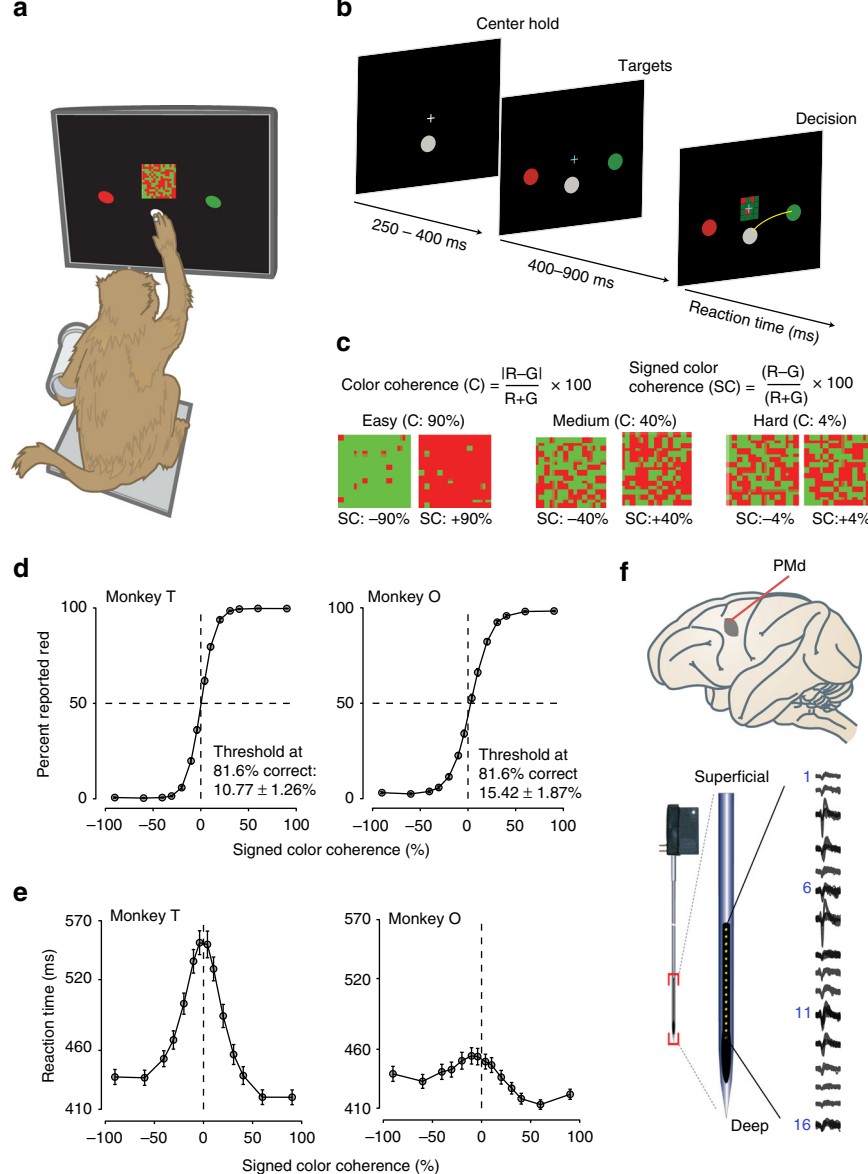

**Fig. 1** Recording locations, techniques, task, and discrimination behavior. **a** An illustration of the setup for discrimination. We gently restrained the arm the monkey was not using with a plastic tube and cloth sling. We tracked a reflective IR bead taped on the middle digit of the hand to mimic a touch screen and to provide an estimate of instantaneous arm position and tracked eye position using an infrared reflective mirror placed in front of the monkey's nose. **b** Timeline of the discrimination task. **c** Examples of different stimulus ambiguities used in the experiment parameterized by the color coherence of the checkerboard cue defined as (C = 100×|R − G|/(R + G)). The corresponding SC is defined as SC = 100×(R − G)/(R + G). Positive values of SC denote more *red* than *green* squares and vice versa. **d**, **e** Average discrimination performance (**d**) and RT (**e**) over sessions of the two monkeys as a function of the SC of the checkerboard cue. RT plotted here includes both correct and incorrect trials for each session and then averaged across sessions. *Gray markers* show measured data points along with 2 × SEM estimated over sessions. The *black line* segments are drawn in between these measured data points to guide the eye. For many data points in **d**, the *error bars* lie within the marker. *X*-axes in both **d**, **e** depict the SC in %. *Y*-axes depict the percent responded *red* in **d** and RT in **e**. Also shown in **d** are discrimination thresholds (M ± SD over sessions) estimated from a Weibull fit to the overall percent correct as a function of coherence. The discrimination threshold is the color coherence level at which the monkey made 81.6% correct choices. 75 sessions for monkey T (128,989 trials) and 66 sessions for monkey O (108,344 trials) went into the averages. **f** Location of PMd along with an example recording from a 16 electrode, 150-micron spacing U-probe. The brain in this figure is adapted by permission from Macmillan Publishers Ltd: Nature Reviews Neuroscience (Fig. 3 of ref. [75]), copyright 2004

term provided the best fit of all three models considered here[10]. Our results here further highlight the increasing realization that differentiating between these models of decision-making behavior using purely statistical techniques is currently very difficult[47–49]— explicit stimulus manipulations are necessary. Additional elaboration of these models is likely needed to better describe the behavior of these monkeys in this and likely other discrimination tasks.

**PMd neurons are organized along a visuomotor continuum.** Having observed that the behavior of the monkeys in this task is similar to other reports of monkeys performing decision-making tasks[3, 11, 45, 50], we next examined the temporal patterns of FRs of neurons in caudal PMd during the period of decision formation to delineate putative decision-related neurons (Fig. 1f, 546 units in T, 450 units in O; single neurons and multi-units that

responded during any epoch in the task). Of the 996, 801 PMd units reported here are single neurons (384 from O, 417 from T, mean ISI violation (<=2ms) = 0.43%, ~0.13 additional spikes per trial) and the remaining 195 units reported here are classified as multi-units (129 from T, 66 from O, mean ISI violation = 3.36%, ~1.4 additional spikes per trial). We did not exclude multi-units because they were often robustly modulated during the task and we wanted all the power we could get for analysis of laminar differences. Conclusions did not change when restricting the analyses to just the isolated single neurons. The number of neurons/units reported here is substantially larger than other recent studies of neuronal responses in PMd during decision-making tasks[10, 11]. We leverage this large physiological data set recorded in PMd to attempt to address the two goals we proposed in the introduction.

Qualitatively, electrophysiological characteristics and effects reported here were similar in both monkeys. The main figures report data pooled across both monkeys and also single and multi-units for more clarity. Analyses are also shown in Supplementary Figures for each monkey where appropriate to emphasize the generality of our results.

Caudal PMd units did not typically modulate their FRs to target onset in this task. Modest changes in FRs in response to

target onset might be either a result of the targets always being shown at predictable spatial locations. Suppressive effects on FRs may also occur when there is movement uncertainty due to the presence of multiple targets[9]. Unlike the modest modulations observed to target onset, visual checkerboard cue onset resulted in a robust modulation of FRs of some PMd units. For example, the unit in Fig. 2a (782 trials) increased its FR ~150–200 ms after checkerboard cue onset. Moreover, for this unit, higher color coherence stimuli were associated with a faster rate of divergence of FRs as a function of choice and lower color coherence stimuli resulted in a slower rate of divergence of FRs as a function of choice. In contrast, the unit in Fig. 2b (1454 trials) decreased its FR after checkerboard cue onset. The FRs of this unit only modestly changed with coherence; FRs ultimately covaried with the animal's choice for this unit, but only for the last ~100 ms before movement onset.

In contrast to FR modulation time-locked to checkerboard cue onset for the units shown in Fig. 2a, b, FR modulations in "perimovement" units appeared in the last ~150 ms before movement onset and were clearly time-locked to the movement (Fig. 2c, 655 trials). The response pattern of the "perimovement" unit shown in Fig. 2c only modestly changed with color coherence. These examples shown in Fig. 2a–c suggest that some units increase or decrease in FR in response to the visual stimulus and identification of the choice, whereas others are more tightly associated with the movement initiation process. The single examples suggest that FR patterns in PMd during the process of decision formation span a bidirectional visuomotor continuum[9, 29], with increased and decreased activity in response to visual stimuli at opposite poles and perimovement activity in the middle.

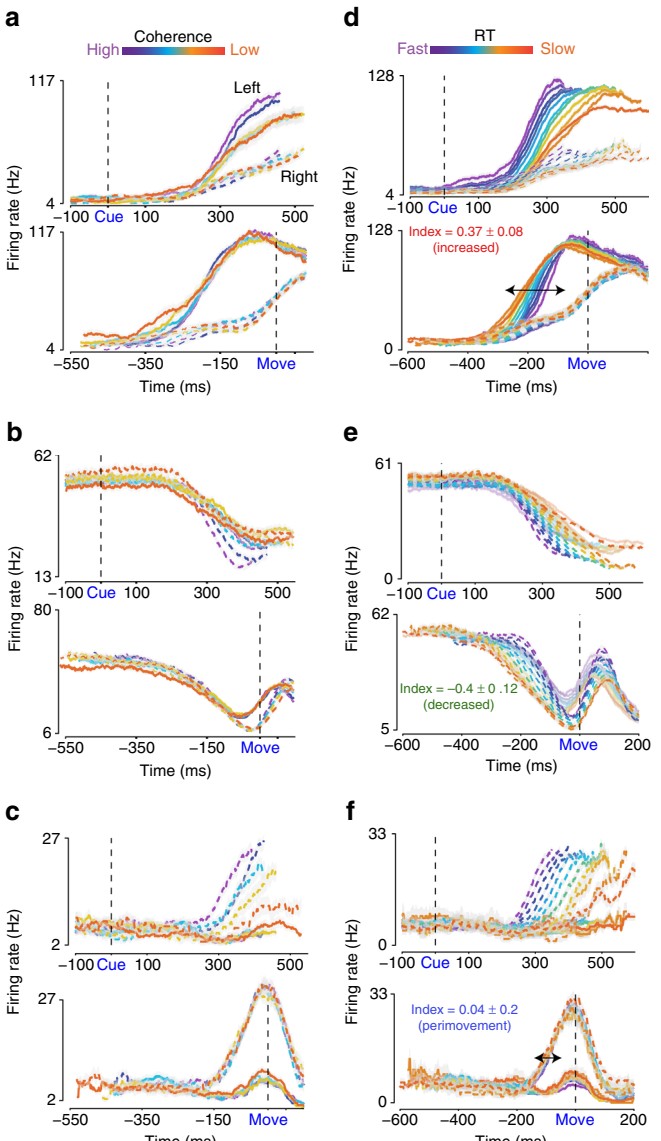

Fig. 2 FRs in PMd demonstrate an organized relationship to different elements of the decision-making process. **a–c** Three example units in PMd during the decision task sorted by color coherence and arm movement choice when aligned to checkerboard cue onset (labeled as "cue", *top panel*) or movement onset (*bottom panel*). *Solid lines* of different colors (—) depict left reaches; *dashed lines* of varying colors (---) depict right reaches. The colors ranging from *purple* to *orange* depict color coherence (high to low). The *black dashed lines* depict the onset of the checkerboard cue (*top panels*) or the onset of the arm movement (*bottom panels*). When aligned to checkerboard cue onset, FR is plotted until the median RT for the color coherence. When aligned to movement onset, the FR is plotted until the negative of the median RT for the color coherence. *Error bars* denote standard error of the mean (SEM) over trials. We estimated peristimulus and perimovement FRs by convolving spike trains with 75 ms causal boxcars. Averages include all trials sorted according to the choice of the monkey (left vs. right). Unit a: 782 trials, Unit b: 1454 trials, Unit c: 655 trials. Individual traces are averages of at least 30 trials. **d–f** FRs of the units that are shown in **a–c** when sorted by RT. The colors ranging from *purple* to *orange* depict RT bins (fast to slow, all color coherences combined). Line conventions for left and right reach directions are as in **a–c**. FR traces, when aligned to checkerboard cue onset and organized by RT, are shown until the *center point* of the RT bin. The visuomotor index (see Fig. 3a, and corresponding text), along with the label is shown for each unit along with CI for the index. The index is significantly different from 0 for units in **a**, **b** (CIs do not overlap with 0) but not **c** (CIs overlap with 0). The trials used in **d–f** are same as those used for the plots in **a–c** except re-sorted according to the various RT bins. In Fig. 2e, FR traces for the left reaches are deliberately reduced in opacity to better highlight the structure in the FR traces for right reaches and their organization with RT. FRs for left reaches also have the same structure. Additional examples of unit responses in PMd that increase, decrease, or are perimovement in nature are shown in Fig. 4 and Supplementary Fig. 11a

This proposal that PMd FRs lie along a bidirectional continuum was again apparent when the FRs of units in Fig. 2a–c were re-sorted and re-averaged as a function of RT (Fig. 2d–f). Figure 2d shows the FR of the unit shown in Fig. 2a now organized by RT. FRs of this unit when organized by RT again exhibited choice-dependent modulation ~150–200 ms after checkerboard cue onset. The longest RTs (500–1000 ms) are associated with FR modulations up to 350–400 ms before the onset of movement (Fig. 2d, *bottom panel*, see *arrow*[45]). Similarly, Fig. 2e shows the FRs of the unit in Fig. 2b now sorted by RT. FR of this unit again decreased after checkerboard cue onset; the duration of this decrease covaries with RT. The unit only recovered its FRs and selectivity for choice just before (~100 ms) movement onset. Finally, Fig. 2f shows the FRs of the perimovement unit in Fig. 2c now organized by RT. FR modulations for this unit again appeared only ~150 ms before movement initiation and regardless of RT they are tightly locked to movement initiation (*bottom*

*panel* of Fig. 2f, see *arrow*). We exploited this organization by RT when aligned to movement onset to broadly categorize the units.

**Fig. 3** Units in PMd that enhance FRs after checkerboard cue onset covaries strongest with the decision process. **a** Histogram of the visuomotor index demonstrating broad unit categories in PMd. Significant positive numbers reflect increased units (*shaded red*, *n* = 514 units), significant negative numbers for decreased units (*shaded green*, *n* = 141 units). Intermediate insignificant values of the visuomotor index reflect the perimovement structure of these units (*shaded blue*, *n* = 341 units). The index is defined on a per-unit basis and measured as the trial-by-trial correlation coefficient between RT and average FR in the −600 to −200 ms epoch before movement onset. *X*-axes depict index; *Y*-axes the number of units. **b** Average population level choice selective signal (|left—right|) in PMd for the population of increased units (*n* = 514 units) as a function of the color coherence aligned to checkerboard cue onset. All trials are included and sorted by the choice of the monkey. The colors ranging from *purple* to *orange* depict color coherence (high to low). *X*-axes depict time in ms; *Y*-axes depict the FR in Hz. *Shaded errors* denote SEM over units. For each unit, we also subtracted the absolute difference in baseline FR before averaging. The *gray shaded region* highlights the 150–350 ms epoch used to estimate the slopes plotted in Fig. 3c. **c** Slope of the choice selectivity signal (as shown in **b**) in the 150–350 ms epoch (for e.g. demarcated by the *shaded gray region* in **b**) after checkerboard cue onset for increased (514 units), decreased (141 units), and perimovement units (341 units) in PMd for the seven different color coherence levels. *Red color* denotes increased units. *Blue* and *green* colors denote the perimovement and decreased units. *Error bars* are SEM estimated over units. We compared the slopes of these curves to a slope estimated through shuffling across color coherences. **d** Average population level choice selective signal (|left—right|) in PMd for the increased units (*n* = 514 units) as a function of the RT aligned to checkerboard cue onset. Different colored lines here depict different levels of RT; *purple* colors depict fast RTs, *orange* colors depict slow RTs. *X*-axes depict time in ms; *Y*-axes the FR in Hz. For each unit, we also subtracted the absolute difference between left and right FRs during the hold period for each unit before averaging across units. **e** The population response of increased units (shown in *red*) begins to signal the eventual choice ~100–150 ms after checkerboard cue onset regardless of RT. Perimovement (shown in *blue*, 341 units) and decreased units (shown in *green*, 141 units) signal choice closer to movement onset. *Y*-axes plot the discrimination time defined as the first time at which the choice selective signal significantly departed from the FR before the onset of the checkerboard cue (i.e., the hold period) as estimated using a paired *t*-test that corrected for multiple comparisons. *X*-axes depict different RT bins. The *error bars* on this time estimate are calculated by bootstrapping FRs for each unit and then estimating the latency using this bootstrapped FR distribution for the subpopulation of units. Lines denote regression fits of the average latency to the center of the RT bin. **f** Covariation with RT was observed within a color coherence level for the increased units. For these units, the mean rate of rise of the choice selective signal was faster for faster RTs even within each level of color coherence (for every unit, RTs were split into fast and slow RTs using the median RT). *X*-axes depict different levels of color coherence. *Y*-axes the slope of the choice selective signals in the 150–350 ms after checkerboard cue onset as in **c**. *Error bars* denote SEM over units. **g** When aligned to movement onset, for increased units (*n* = 514) the average choice selective signal ~100 ms before movement onset only modestly covaries with color coherence. The colors ranging from *purple* to *orange* depict color coherence (high to low). *X*-axes depict time in ms. *Y*-axes the magnitude of the choice selectivity signal in Hz. *Shaded errors* denote SEM. **h** By the time of movement onset, the average choice selective signal in the −100 ms to move onset for any of the various broad classes of units in PMd does not strongly change depending on the color coherence of the checkerboard cue. The slopes of the curves are not significantly different from slopes estimated via shuffling. *X*-axes depict color coherence in %. *Y*-axes the FR in the −100 ms to 0 ms before movement onset

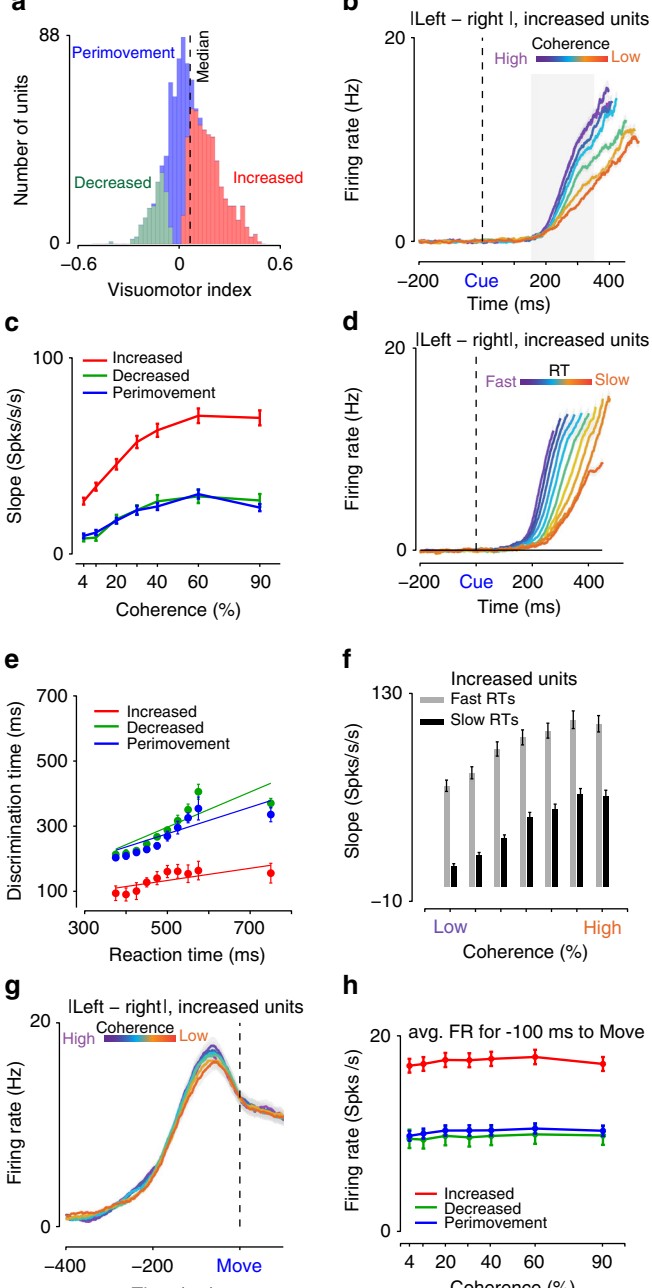

**Increased PMd neurons show stronger decision-related activity**. To summarize the activity of the population of units distributed along this continuum, we first developed an index to broadly separate the units into "increased", "decreased", and "perimovement" categories (Fig. 3a). The index was defined as the trial-by-trial correlation coefficient between RT and time-averaged FR in the −600 to −200 ms epoch before movement onset, after subtraction of the baseline FR during the holding period (measured in the 200 ms epoch before checkerboard cue cue onset). For computing the correlation coefficient, both left and right reach directions are included, and choice is not included as a factor. For trials with RT less than 600 ms, we truncated the start of the measurement window to coincide with the start of checkerboard cue presentation when measured relative to movement onset. We correlated the modified time-averaged FR to the RT across all trials to estimate the index.

We labeled units with significantly positive indices as "increased" (e.g., Fig. 2a, d, index (M ± CI): 0.37 ± 0.08), significantly negative indices as "decreased" (e.g., Fig. 2b, e, index:−0.4 ± 0.12), and units with indices not significantly different from zero as perimovement (e.g., Fig. 2c, f, index: 0.04 ± 0.2). The distribution of the index from negative to positive values reflects the continuum of responses from decreased to increased responses to checkerboard cue onset (Fig. 3a, Supplementary Fig. 3a). The median visuomotor index is positive and significantly different from zero, suggesting a slightly greater presence of increased compared to decreased and perimovement units (median = 0.065, two-tailed Wilcoxon sign-rank vs. 0, $p = 1.13e-57$, Fig. 3a). As per the index and the criteria we used, 514/996 units (51.6%) were categorized as increased, 141/996 (14.15%) as decreased, and the remaining 341/996 (34.25%) as perimovement. Increased, decreased, and perimovement units were observed in both monkeys (Supplementary Fig. 3c, d, median index for monkey T = 0.099, Wilcoxon signrank $p = 2.12e-22$; median for monkey O = 0.045, Wilcoxon signrank $p = 1.56e-37$).

We next used the visuomotor index to selectively average FRs of PMd units and show that the population responses of the increased units covary the strongest with color coherence, choice, and RT. We also correlated this visuomotor index to various metrics of interest such as the timing of choice selectivity and to other metrics proposed in recent studies of visuomotor continuums[26, 33]. Using these results, we contend that the FRs of increased, but not decreased and perimovement units are consistent with a candidate decision variable.

The first neural prediction made by computational models of decision-making for neurons that covary with an evolving decision is that on average the build-up of neural activity in favor of a choice, as measured in plots of trial-averaged FR vs. time, is faster for easier compared to harder color coherences[45]. The FRs of increased units were the most consistent with this prediction. For increased units, increases in color coherence were accompanied by an increase in the rate at which the average choice selective signal (defined as |left—right| averaged over units) developed (Fig. 3b–c, Supplementary Fig. 3e, slope of FR curve from 150–350 ms, M ± SE: 50.16 ± 3.33 spks/s²/100% color coherence, comparison to shuffled, sign test $p = 1.22e-27$). This 200 ms range over which we computed slopes is within the range of decision times estimated in Supplementary Fig. 1d for both monkeys, assuming that the DDM is the correct model to describe the behavior.

The rate of the build-up of neural activity in favor of a choice for the decreased and perimovement units also increased modestly as a function of color coherence, suggesting that they still contained vestiges of the decision-formation process (Fig. 3c, M ± SE, decreased: 24.73 ± 3.30 spks/s²/100% color coherence,

comparison to shuffled, sign test $p = 2.72e-08$; perimovement: 19.71 ± 1.86 spks/s²/100% color coherence, comparison to shuffled, sign test $p = 3.72e-15$).

The dependence on color coherence was stronger for increased units compared to the other two broad categories of units (positive correlation between the visuomotor index and the dependence of slopes on color coherence, Spearman's $r = 0.25$, $p = 6.35e-16$; comparison of slopes of lines in Fig. 3c, bar plots shown in Supplementary Fig. 3f, g, Wilcoxon ranksum $p = 3.505e-04$, increased vs. perimovement: Wilcoxon ranksum $p = 1.43e-09$, decreased vs. perimovement: Wilcoxon ranksum $p = 0.36$; permutation tests, 10,000 repeats: increased vs. decreased and increased vs. perimovement: $p = 2e-4$; decreased vs. perimovement: $p = 0.21$). Thus, increased units demonstrate the strongest modulation of FRs with the stimulus color coherence and choice. The order of these effects (increased > decreased ~ perimovement) and the positive correlation between the dependence on color coherence and the visuomotor index was again present in both monkeys (Supplementary Fig. 3g, monkey T: $r = 0.36$, $p = 3.2e-18$; monkey O: $r = 0.23$, $p = 1.02e-6$).

**Choice selective signals emerge earlier in increased neurons.** We next examined the latency of the population level choice selectivity signal for each of the broad unit categories. This latency which is termed 'discrimination time' was defined as the first time at which the choice selective signal significantly departed from the FR in the hold period before the onset of the checkerboard cue as estimated using a paired $t$-test that corrected for multiple comparisons[34]. Error bars on discrimination time were obtained through bootstrapping. Population level choice selectivity in increased units appeared ~100–150 ms after checkerboard cue onset, and this discrimination time increased only modestly with RT (mean slope of regression M ± SE: 0.2 ± 0.06 ms/1 ms of RT, Fig. 3d–e, Supplementary Fig. 4a–c). Furthermore, discrimination times were earlier for the increased compared to decreased and perimovement units (bootstrapped time estimates do not overlap). Decreased and perimovement units also exhibited a more pronounced increase in discrimination time with RT (decreased: M ± SE = 0.53 ± 0.15 ms/1 ms of RT, perimovement: 0.4 ± 0.11 ms/1 ms of RT, Fig. 3e, Supplementary Fig. 4a–c). These trends were observed separately in each monkey (Supplementary Fig. 4a, c) and also when the analysis was performed on a neuron-by-neuron basis (Supplementary Note 2, Supplementary Fig. 4d–f).

The differences between FR patterns for long vs. short RTs in Fig. 3d could emerge entirely due to mixing responses to different levels of visual stimulus difficulty. Long RTs are more likely to come from harder color coherences and short RTs from easier color coherences. Differences as a function of RT could just be an artifact of this separation. However, this explanation did not hold for our results. For increased units and for every color coherence level, the slope of the choice selective signal increased faster for short compared to long RTs (Fig. 3f, sign rank test comparing the slope of the choice selective signal for fast vs. slow RTs; maximum $p$-value = 9.21e-31; monkey T: 1.65e-19; monkey O:5.29e-13) arguing against explanations which posit that observed effects of RT are due to the spurious mixing of visual (i.e., sensory) responses to different color coherences.

**Choice selectivity does not depend on coherence at movement onset.** We next examined how FRs at the time of movement initiation depended on coherence and RT. At the time of movement onset, regardless of broad unit category, the magnitude of the average choice selective signal in the −100 ms to

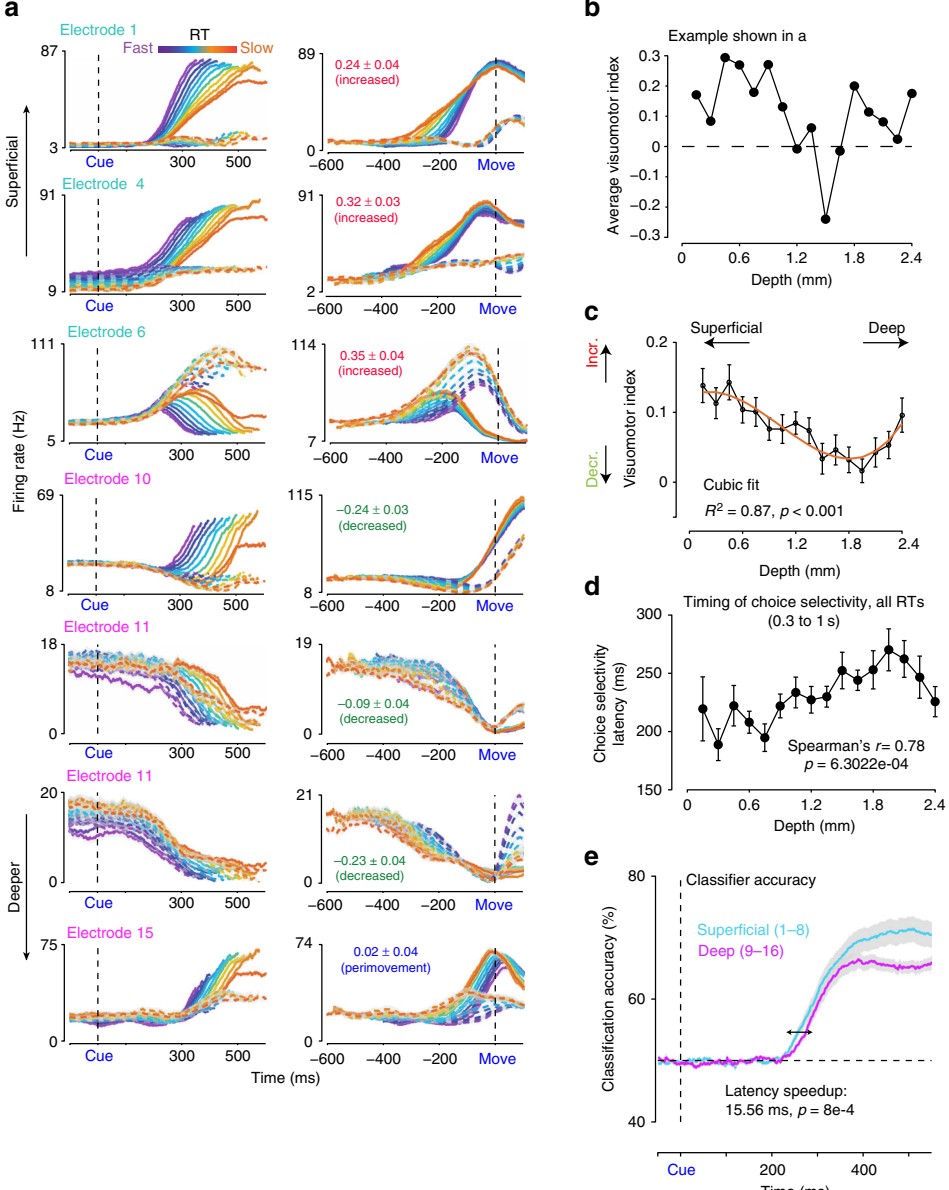

**Fig. 4** Laminar differences in the time course of decision-related signals in PMd. **a** PSTHs from seven different units recorded in the same session sorted as a function of RT and choice and ordered from superficial to deep electrodes. The FRs are aligned relative to checkerboard cue onset (*left*) or to movement onset (*right*). The colors ranging from *purple* to *orange* depict different RT bins. *Shaded errors* denote SEM. *Solid lines* depict left reaches and *dashed lines* depict right reaches. Peristimulus and perimovement FRs are estimated by convolving spike trains with 75 ms causal boxcars. Averages include all trials sorted according to the choice of the monkey (left vs. right). Also shown for each unit is the visuomotor index estimated for the unit along with confidence intervals and the label associated with the unit based on the visuomotor index. The rough pattern that this single example and a similar example shown in Supplementary Fig. 11 is that superficial electrodes have positive indices denoting increased units whereas the deeper electrodes are likely to have more perimovement and decreased units. *X*-axes depict time in ms. *Y*-axes depict FR in Hz. PSTHs and PMTHs reflect FR averaged over more than 100 trials per condition. >1800 trials were analyzed for these units. These trials were then separated according to the condition. **b** Visuomotor index as a function of cortical depth for the session shown in Fig. 4a. *X*-axes depict depth in mm. *Y*-axes depict the index. *Error bars* denote SEM. Visuomotor index for each electrode is estimated by averaging over all units recorded on that particular channel. **c** The visuomotor index estimated by pooling over sessions from both monkeys T and O decreases as a function of depth (68 sessions, 554 units). *X*-axes depict depth in mm. *Y*-axes depict the index. *Error bars* denote SEM. The line (shown in *orange*) is the fit of this average index vs. cortical depth using a cubic function ($ax^3 + bx^2 + cx + d$). $x$ is a variable denoting cortical depth. Reported *p*-value for the fit is obtained by a permutation test where we shuffled to remove the relationship between the index and cortical depth and computed the fit. We repeated this shuffling 1000 times and fit the cubic function to estimate a surrogate distribution of $R^2$ values. A simple linear regression was also significant ($R^2 = 0.61$, $p < .001$). We chose higher order fits because of the non-monotonic nature of the change in the index as a function of cortical depth. **d** Average choice selectivity as a function of cortical depth for the population of PMd neurons. *X*-axis depicts depth in mm. *Y*-axis the discrimination latency in ms. *Error bars* denote SEM estimated over sessions. **e** Classifier accuracy for superficial (electrodes 1–8) and deep (9–16) as a function of time for all RTs when aligned to checkerboard cue onset. The classification was performed on a session-by-session basis, and the number of units used for superficial and deep electrodes were equalized by setting the number of units used for the superficial and deep classifiers to be the same. We only used sessions where we had greater than 10 units recorded from the U-probes in this classification analysis. We used 50 ms bins stepped by 2 ms bins and used a linear classifier

movement epoch was only weakly modulated by color coherence/ RT (for instance in the PMTHs shown in Fig. 2d–f; M ± SE, sign-test to compare median slopes for the average FR in −100 ms to move as a function of color coherence and the slopes for a shuffled curve, increased: $0.30 \pm 0.24$ spks/s$^2$/100% color coherence, $p < 0.1$; decreased: $0.49 \pm 0.33$ spks/s$^2$/100% color coherence, $p < .16$; perimovement: $0.54 \pm 0.25$ spks/s$^2$/100% color coherence, $p < 0.09$; Fig. 3g, h, Supplementary Fig. 4g). These results were again observed in both monkeys (Supplementary Fig. 4j, sign test, monkey T: increased, $p < .07$, decreased: $p < .14$, perimovement: 0.59; monkey O: increased, $p < 0.51$, decreased: $p < 0.67$, perimovement: $p < 0.37$).

This observation that the magnitude of the choice selective signal at the time of movement onset only changes weakly with color coherence or RT is consistent with proposals of a commitment state in the network prior to the initiation of the choice[10, 45, 51]. In some contexts, additional analysis has been used to argue for a threshold implemented by single neurons[45], perhaps mimicking the thresholds used in behavioral models. Our analysis showing a lack of correlation between the magnitude of the choice selective signal (which is the difference in FRs for different reach directions) and coherence cannot be considered as evidence for the rise-to-threshold mechanism in PMd for decisions. Previous studies suggest that a rise-to-threshold mechanism provides a poor explanation of FRs of PMd neurons even in simple delayed-reach tasks that do not involve perceptual decisions[52].

Choice selectivity is distributed in a continuum in this PMd neural population and is not well described as clusters of high and low choice selectivity (Supplementary Note 3, Supplementary Fig. 4h). One concern is that the visuomotor index we use to partition and understand our large data set of neurons is too simplistic and collapses over many key features of the data. To ensure that increased, decreased, and perimovement units were not a spurious artifact specific to the index that we developed here, we also tested if our results were consistent when applying other methods for describing a visuomotor continuum[9, 33] as well as a novel technique reported by Erlich and collaborators[53] for estimating a correlation between RT and neural responses (Supplementary Note 4, Supplementary Figs. 5–9). These results were also not explained as an effect of differences in the kinematics of the eventual arm movement (e.g., speed) or eye position (Supplementary Note 5, Supplementary Fig. 10). Finally, the conclusions did not change even when we excluded the multi-units and only analyzed the single neuron responses (Supplementary Note 6, Supplementary Fig. 13).

**PMd superficial layers show earlier choice selectivity**. We next addressed the second goal of the study. We asked the following question. Are there differences between the FR dynamics of the superficial layers of PMd and the deep layers during the process of decision formation? Specifically, are the increased, decreased, and perimovement units systematically or randomly distributed as a function of cortical depth in PMd[54, 55]? Identifying how decision-related response properties of neurons vary as a function of anatomical and physiological properties (e.g., layer) is an important first step for building and testing cortical circuit models of decision-making[14, 56, 57].

The laminar multi-contact electrodes allowed us to record simultaneously from small populations of neurons and identify their cortical depth. We used this technique and examined PSTHS/PMTHs from single sessions to identify if there was any rough organization as a function of cortical depth. Figure 4a and Supplementary Fig. 11a show units from example sessions from monkey T during this task. In both single session examples,

increased units were more prevalent in the superficial electrodes, whereas the decreased and perimovement-related units were more common in the deeper electrodes. For instance, in Fig. 4a, for the units recorded from electrodes 1, 4, and 6, the index (shown with confidence intervals in the figure) is significantly positive and different from zero. In contrast, the units recorded from electrodes 10 and 11 have significantly negative indices. Finally, electrode 15 recorded a unit with an insignificant index, suggesting a perimovement unit. Figure 4b shows the visuomotor index for this session as a function of cortical depth. A similar trend is observed for the examples in Supplementary Fig. 11. Units recorded from superficial electrodes (1, 3, and 5) have positive and significant indices, whereas the units recorded from deeper electrodes (8, 11, 13, and 14) either have insignificant perimovement indices or negative indices. This separation was not perfect. Occasionally deeper electrodes recorded from units that demonstrated increases in FR before movement onset. For example, electrode 16 recorded from an increased unit. Supplementary Fig. 11b plots the visuomotor index as a function of the depth for this session. As the depth increases the visuomotor index also appears to decrease for this session.

The single session examples indicate that units at different cortical depths can have different FR modulations during decision-making and thus a laminar organization. This indication of differences in response profiles as a function of cortical depth held in the population of recordings. As one moves from superficial to deep electrodes, the index decreased as a function of depth (Fig. 4c, a cubic regression between the index averaged over 68 sessions (24 in T, 44 in O) vs. cortical depth, $r^2 = 0.87$, $p < .001$). At least, on average across both monkeys, the increased units are more likely in superficial parts of PMd, whereas decreased and perimovement units are found at relatively deeper depths in PMd. Effects of cortical depth on the visuomotor index were observed in both monkeys (Supplementary Fig. 11c). In monkey O, the decrease in the visuomotor index as a function of depth was somewhat less pronounced albeit still significant (cubic fit, monkey T: $r^2 = 0.86$, $p < .001$; monkey O: $r^2 = 0.67$, $p < .003$) and when pooled over both monkeys this effect remained significant. The same results were observed even when we used the more sophisticated correlation method developed in ref. [53] (correlation between the sign corrected correlation coefficient estimated in ref. [53] and cortical depth, Spearman's pooled $r = -0.83$, $p = 5.24e{-}5$, monkey T: $r = -0.45$, $p = 0.08$, monkey O: $r = -0.81$, $p = 3.5e{-}4$).

Given that the FRs of the increased units have the strongest relationship to a decision variable, the preponderance of increased units in superficial electrodes of PMd naturally resulted in differences in discrimination time as a function of cortical depth. Figure 4d shows the average time of significant choice selectivity (computed over all RTs from 300–1000 ms when aligned to checkerboard cue onset) as a function of cortical depth. As cortical depth increases, the discrimination time also increases (Spearman's $r = 0.77$, $p = 6.3e{-}04$, T: $r = 0.67$, $p = 0.005$, O: $r = 0.64$, $p = 0.012$, Supplementary Fig. 11d). These trends for visuomotor index and discrimination time were not observed in the posterior locations of our recording chamber (likely M1, Supplementary Note 7, Supplementary Fig. 12a).

We finally used the simultaneously recorded neurons from the U-probe recordings in PMd and performed a decoding analysis to identify if single-trial population level choice selectivity emerged earlier in the superficial compared to the deeper layers. We separated the units into two groups, superficial (electrodes 1–8) and deep (electrodes 9–16). We ensured that we restricted the classification analysis to the same trials and also equalized the number of units for both superficial and deep electrodes. We only included sessions where we had more than 10 neurons/multi-

units recorded for this classification analysis to give us sufficient power and be confident in the classification results. This criterion left us with 19 sessions in monkey T and 12 sessions in monkey O. When pooled over all these sessions, average and median classification accuracy increased earlier for the superficial compared to the deep electrodes (Fig. 4e, latency for 55% accuracy, ~15.57±4.8 ms earlier, $p < 8e-4$, bootstrap test vs. 0, 10,000 iterations). When examined individually in each monkey, classification accuracy showed trends similar to the pooled average although it was only significant in monkey T (superficial vs. deep, Supplementary Fig. 12b, c, T: ~14.41 ms earlier, $p = 3e-4$, O: ~15.23 ms earlier, $p = 0.15$).

## Discussion

The motivation for our study was to address key unresolved questions regarding the neural circuit dynamics in PMd underlying somatomotor decisions. We had two concrete goals: (1) to investigate the temporal pattern of FRs in PMd during decisions and use various parameters and predictions of computational models of decision-making to identify putative decision-related neurons, and (2) to investigate if these putative decision-related neurons were anatomically localized as a function of cortical depth. To achieve these goals, we used a visual RT discrimination task and multi-electrode recordings to demonstrate that FR dynamics of a diverse population of PMd units covary with decision formation and movement initiation. At the highest level, our results provide additional support for observations from lesion and physiological studies in monkeys performing target selection and perceptual decision-making tasks[5, 6, 8–11] and clinical examinations of stroke patients that suggested a role for PMd in selecting appropriate actions based on sensory cues[7].

We found that FR dynamics in PMd exhibit substantial variation within the population during decision formation[8–11]. Our first result shows that this variation can be summarized by viewing neural responses as organized along a bidirectional visuomotor continuum[9, 29, 31, 33, 34, 58]. We emphasize that we are not just showing that PMd units can increase or decrease their FR during somatomotor decision-making. By correlating response profiles to behavioral parameters such as choice, RT, and stimulus difficulty, our second result shows that a broad class of neurons within this continuum exhibits properties consistent with a role in the decision formation process. Specifically, increased units signal choice earlier than the decreased and perimovement units and are the most consistent with the predictions of computational models such as the DDM[29, 30, 45, 46, 59, 60] and the UGM[10, 43, 44].

In contrast to the increased units, the perimovement units we report only modulated their FRs in an all-or-none manner starting approximately 150 ms before the initiation of movement. These FR modulations appeared after decision-related modulations in the increased units. This temporal order and differences in degree of covariation with the decision-formation process is reminiscent of FR profiles in multi-module neural network models that can simulate both the temporal integration of sensory inputs and threshold detection aspects of an RT task[30, 36, 56]. In one such influential multi-module neural network model[36], neurons in the first module accumulate sensory evidence and show FR profiles similar to those observed for the increased units in our task. In contrast, the second module, which receives connections from the first accumulation module, demonstrates all-or-nothing changes in FR just prior to the initiation of the movement. All-or-nothing changes in this module implement the threshold for the initiation of the movement and the FRs of this second all-or-nothing module resembles the FRs of the perimovement units we find in PMd. In the original study[36], these modules were considered as implemented in different brain areas

with lateral intraparietal area (LIP) acting as an accumulator and the superior colliculus, which contains burst neurons that demonstrate perimovement discharges, as the threshold detection mechanism. Based on this neural network model[36], one interpretation of our results is that there are multiple computational modules even within a single area, PMd, during decision-making[30, 56]. How decreased units fit into this framework is currently unclear. Training randomly connected recurrent neural networks and then perturbing them may help provide additional insight into the role of these different types of FR dynamics in PMd during somatomotor decisions[23, 37].

The decreased neurons had tonic levels of FRs that began to reduce after the checkerboard cue onset and showed minimal choice selectivity. FRs of these neurons were similar for left and for right reaches and only diverged as a function of choice just before movement onset. These FR characteristics share some characteristics with fixation neurons documented in the frontal eye fields[35] and the superior colliculus[61]. Fixation neurons, like the decreased neurons described here, reduce their activity before the initiation of a saccade[35]. They also form a small proportion of neurons described in FEF—an observation consistent with the result that decreased neurons were the least common of the broad categories (~15% of the neural population). Finally, fixation neurons are thought to be colocalized with movement-related neurons in layer 5 of the FEF—a result consistent with our report here of finding these decreased and perimovement neurons largely in the deeper electrodes of the U-probes and thus the deeper layers of PMd[35].

Our primary analysis using indices that exploit features of the FRs and showing that PMd neurons are organized along a visuomotor continuum might be construed as supporting the standard practice in neuroscience which uses either preselection criteria[45] or indices and metrics to divide neurons into subcategories[9, 29, 62]. This is not our intention. Like others[25, 27], we recognize and indeed show that in an almost randomly sampled population of PMd neurons there is considerable variation in the signals carried by single neurons during decision-making. For instance, even within the broad category of increased units, there are interesting temporal features in the neuronal FR patterns (as seen in the clustering analysis shown in Supplementary Fig. 9). Said differently, the FR of every increased neuron is not easily described as a scaled version of a prototypical integrator (e.g., Fig. 4 and Supplementary Fig. 11). Across the entire PMd population, the FRs of some units are tightly associated with the visual stimulus, others more associated with movement, while yet others occupy the continuum between these extremes[9, 29, 33]. The visuomotor continuum we show here is one effort to provide some organizational principles for this temporal variation in PMd during a somatomotor decision.

How might one understand this variation and heterogeneity in the PMd population? The initial approach that we (and others) have used is to treat population activity in a brain region as a dynamical system[23, 25, 63]. In this view, the focus is on a population-level description using either supervised or unsupervised dimensionality reduction techniques. These techniques allow visualization and description of FR variance in lower dimensions[23, 25, 63]. This dynamical systems view is a largely agnostic approach, by which we mean that there are minimal assumptions and ad hoc sorting of neurons into subcategories is not performed. In a broad sense, the visuomotor continuum that we used here is a handcrafted dimensionality reduction technique that organizes FR variance along one axis. As we showed in the results, this handcrafted axis was tightly related to the first principal component, which explained nearly 50% of the variance. We also explored other recent methods of characterizing this neural population including the normalized latency and

covariance metrics adopted in other studies of visuomotor continuums[33].

Future work that analyzes trial-averaged and single-trial trajectories of these PMd FRs may provide additional insights into the underlying computational mechanisms, and additional organizing principles to subsume this temporal variation[23]. In parallel, we also expect that additional understanding of FR variation in PMd (and other structures) will emerge from systematic reverse engineering of the cortical circuit using techniques such as optical imaging[64], laminar recordings[57, 64], cell-type identification[9, 57], optogenetic perturbation[65], and the use of both conventional[66], as well as novel anatomical techniques[67]. This combined computational and experimental approach will hopefully provide an understanding of perceptual decisions at multiple levels of abstraction ranging from the single-neuron to the underlying dynamical system in PMd.

We recognize that our and other demonstrations of decision-related activity in PMd are correlational and cannot be used to identify whether the decision computation itself emerges in PMd[10, 11]. Additional causal experiments that involve inactivation or perturbation of PMd during a decision-making task are necessary to resolve this question[68] (Supplementary Note 8).

The second advance from our study was the observation that FR dynamics of PMd units are roughly organized as a function of cortical depth during perceptual decisions. The increased units, which show the strongest decision variable characteristics, were more likely in the superficial layers of PMd. The decreased and perimovement units were more likely in the deeper layers in PMd. Differences in FR dynamics in PMd as a function of cortical depth are consistent with reports of cognitive signals appearing in the superficial layers of PFC in monkeys performing visual[54] and working memory tasks[55]. The laminar separation of these broad unit categories was not absolute. For instance, increased units were also occasionally present in the deeper layers in PMd, a result observed in other studies of primate and rodent motor cortex. FR modulations are observed during the delay period of an instructed delay task in layer 5 motor cortical neurons that project to the spinal cord[20], and neurons in layer 5 of the rat motor cortex modulate their FRs during the hold, pre-movement, movement, and post-movement phases of a delayed motor task[57].

Our observation of an organization as a function of cortical depth in PMd complements descriptions of a functional gradient of responses parallel to the cortical surface[8]. In a reach target selection task, neurons in the more rostral parts of PMd (pre-PMd/rostral PMd, closer to the arcuate sulcus) appear to have stronger covariation with the target selection process compared to the most caudal portions of PMd that appear to have more movement-related signals (closer to primary motor cortex)[8]. This prior suggestion of a rostrocaudal organization in PMd along with our results, which show a depth dependent organization, provides additional evidence for an anatomically organized circuit in PMd that contains signals related to somatomotor decisions. Combining magnetic resonance imaging with dense electrode recordings is a promising strategy to recover an "electroanatomical" map of this circuit in PMd and M1[69].

Differences in timing of choice selectivity observed in PMd as a function of cortical depth may arise from differences in connectivity between PMd and other cortical areas and connectivity within PMd itself. Reciprocal connections are present between PMd and prefrontal, parietal as well as premotor areas[70] and these projections from prefrontal and pre-arcuate areas are thought to terminate in superficial layers of PMd[39]. Deeper layers of PMd are also thought to be connected to subcortical structures such as the spinal cord and striatal circuits[71]. Examinations of laminar connectivity in rodent motor cortex[40] also suggest a prominent descending projection from L2/3→L5. Based on these

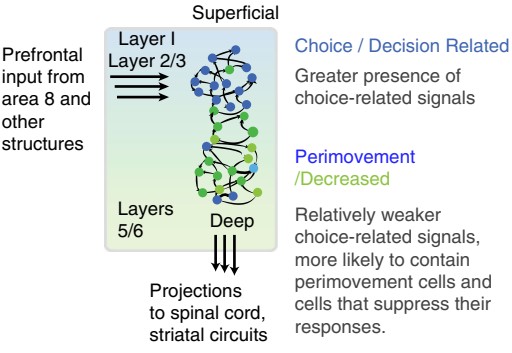

**Fig. 5** A schematic architecture for decision-making in PMd. The superficial layers of PMd are thought to receive inputs from prefrontal and prearcuate areas. The deeper layers of PMd are thought to project to subcortical and spinal circuits. Differences in anatomical connectivity is one hypothesis for why choice-related activity emerges earlier in the superficial compared to the deeper electrodes

anatomical studies, our results, and insights from multi-module computational models of decision-making[36, 56], we propose a summary schematic for decision-related dynamics in PMd (Fig. 5). The framework we propose is that decision-related activity will initially appear in the increased units either as a result of anatomical inputs from other brain regions or as a result of the circuit in premotor cortex. Activity in these increased units, which are more likely in the superficial layers or other brain areas, may in turn influence the dynamics in the decreased and perimovement units, which are more likely in the deeper layers and perhaps involved in the decision threshold mechanism[36]. We view this schematic as a starting point for further investigations into circuit dynamics underlying decision-making[13–15, 30], and perhaps guide the design of the next generation of anatomically guided recurrent neural network models[36, 37, 56].

## Methods

**Subjects**. Our experiments were conducted using two adult male macaque monkeys (*Macaca mulatta*; monkey T, 7 years, 14 kg; O, 11 years, 15.5 kg) trained to reach to visual targets for a juice reward. Monkeys were housed in a social vivarium with a normal day/night cycle. The protocols for our experiments were approved by the Stanford University Institutional Animal Care and Use Committee. We initially trained monkeys to come out of the cage and sit comfortably in a chair. After initial training, we performed sterile surgeries during which monkeys were implanted with head restraint holders (Crist Instruments, cylindrical head holder) and standard recording cylinders (Crist Instruments, Hagerstown, MD). Cylinders were centered over caudal PMd (+16, 15 stereotaxic coordinates) and placed surface normal to the cortex. We covered the skull within the cylinder with a thin layer of dental acrylic/palacos.

**Apparatus**. Monkeys sat in a customized chair (Crist Instruments, Snyder Chair) with the head restrained via the surgical implant. The arm not used for reaching was gently restrained using a tube and a cloth sling. Experiments were controlled and data were collected under a custom computer control system (xPC target and Psychophysics Toolbox). Stimuli were displayed on an Acer HN2741 computer screen placed approximately 30 cm from the monkey. A photodetector (Thorlabs PD360A) was used to record the onset of the visual stimulus at a 1 ms resolution. Every session, we taped a small reflective hemispheral bead (11.5 mm, NDI Digital passive spheres) to the middle digit of the right hand (left hand, monkey O). The bead was taped 1 cm from the tips of the fingers, and the position of this bead was tracked optically in the infrared (60 Hz, 0.35 mm root mean square accuracy; Polaris system; Northern Digital). Eye position was tracked with an overhead infrared camera (estimated accuracy of 1°, Iscan, Burlington, MA). To get a stable eye image for the overhead infrared camera that acquires the eye image, an infrared mirror transparent to visible light was positioned at a 45° angle (facing upward) immediately in front of the nose. This mirror reflected the image of the eye in the infrared range while letting visible light pass through. A visor placed around the chair prevented the monkey from touching the infrared mirror, the juice tube or bringing the bead to his mouth.

**Task structure**. Experiments consisted of a sequence of trials, each of which lasted a few seconds; successful trials resulted in a juice reward, unsuccessful trials in a time-out from 2–4 s. Monkeys used their unrestrained arm (monkey T used his right arm, monkey O his left arm) to reach to touch either red or green targets based on the dominant color in a central, static checkerboard cue composed of isoluminant red and green squares. For every trial, the monkey placed its unrestrained arm on a central target (diameter = 24 mm) and fixated on a small white cross (diameter = 6 mm). After ~350–400 ms had elapsed, two isoluminant colored targets appeared 100 mm to the right and left of the central target. The target configuration was randomized so that colors were not always tied to reach directions. On some trials, the target configuration was red on the left and green on the right; other trials had the opposite configuration. This manipulation allows us to examine if PMd FRs also covary with the color of the target chosen. After an additional hold period (varying from 400–900 ms), a static checkerboard cue (15 × 15 grid of squares; 225 squares in total; each square 2.5 mm × 2.5 mm) composed of isoluminant red and green squares appeared on the screen around the fixation cross (example stimuli are shown in Fig. 1d). The monkeys reached to the target whose color matched the dominant color in the central checkerboard cue. For example, when there was more green than red in the central checkerboard cue, the monkey had to choose the green target. To "choose" a target, the animals moved their hand from the central hold point and stably held a target for a short duration (minimum of 200 ms). The task was an RT paradigm, and thus the monkeys were free to initiate their reach whenever they felt there was sufficient evidence accrued to inform their choices. We did not impose a delayed feedback procedure in this task. That is, if the correct target was chosen, a juice reward was provided to the monkey immediately after the monkeys chose the target[45].

We parameterized the checkerboard cue at several different levels from almost fully red to almost fully green. We used 14 levels of red (ranging from 11 red squares to 214 red squares) in the central checkerboard cue. Each level of red had a complementary green level (e.g., 214 R, 11 G; and 214 G, 11 R squares). We defined seven levels of color coherence (defined as $C = 100 \times |R - G|/(R + G)$, 4–90%). The corresponding signed coherence (SC) was estimated without taking the absolute value of the difference ($SC = 100 \times (R - G)/(R + G)$). For monkey T, we used a uniform distribution of hold period durations between the onset of the targets and the onset of the checkerboard cue. Monkey O attempted to anticipate the checkerboard cue onset. To minimize this anticipation we used an exponential hold period duration (400–800 ms) between the onset of the targets and the onset of the checkerboard cue to reduce predictability.

**Training**. We used the following operant conditioning protocol to train our animals. First, the animal was rewarded for arm movements toward the screen and learnt to take pieces of fruit on the screen. Once the animal acquired the association between reaching and reward, the animal was conditioned to touch a central target for a juice reward. The position, as well as the color of this target, was then randomized as the monkey learned to touch targets at different locations on the screen. We then adopted a design in which the monkey first held the central hold for a brief period, and then a checkerboard cue, which was nearly 100% red or 100% green, appeared for 400–600 ms and finally the two targets appeared. The monkey received a reward for making a reach to the color of the target that matched the checkerboard cue. Two-target "Decision" blocks were interleaved with single target blocks to reinforce the association between checkerboard color and the correct target. After a couple of weeks of training with this interleaved paradigm, the animal reliably reached to the target matching the color of the central checkerboard cue. We then switched the paradigm around by adopting a design in which the targets appeared before the checkerboard cue onset. We initially trained on holding periods (where the monkeys view targets) from 300 to 1800 ms. We trained the animal to maintain the hold on the center until the checkerboard cue appeared by providing small amounts of juice. When the animal reliably avoided breaking central hold during the holding period, we stopped providing small amounts of juice. In subsequent weeks, we introduced more difficult checkerboard cues to the animal while reducing the maximal holding period to 900 ms. We then encouraged the animal to perform this discrimination as accurately and as fast as possible while discouraging impulsivity by adopting timeouts.

**Electrophysiological recordings**. Stereotactic coordinates, known response properties of PMd and M1, and neural responses to muscle palpation served as our guides for electrophysiological recordings. We placed the chambers surface normal to the cortex to align with the skull of the monkey and recordings were performed perpendicular to the surface of the brain. Recordings were made anterior to the central sulcus, lateral to the spur of the arcuate sulcus, and lateral to the precentral dimple. For both monkeys, we confirmed our estimate of the upper and lower arm representation by repeated palpation at a large number of sites to identify muscle groups associated with the sites. Monkey T used his right arm (O used his left arm) to perform tasks. Recordings were performed in PMd and M1 contralateral to the arm used by the monkey.

A subset of the electrophysiological recording was performed using traditional single electrode recording techniques. Briefly, we made small burr holes in the skull using handheld drills. We then used a Narishige drive with a blunt guide tube placed in firm contact with the dura. Recordings were obtained using FHC

electrodes to penetrate the overlying dura. Every effort was made to isolate single units during the recordings with FHC electrodes.

We performed linear multi-contact electrode (U-probe) recordings in the same manner as single electrode recordings with some minor modifications. We used a slightly sharpened guide tube to provide more purchase on the dura. We also periodically scraped away under anesthesia any overlying tissue on the Dura. Sharp guide tubes and scraping away dura greatly facilitated penetration of the U-probe. We typically penetrated the brain at very slow rates (~2–5 µm/s). Once we felt we had a reasonable sample population of neurons, potentially spanning different cortical layers, we stopped and waited for 45–60 min for the neuronal responses to stabilize. The experiments then progressed as usual. We used 180-µm thick 16-electrode U-probes with an inter-electrode spacing of 150 µm; electrode contacts were ~100 Kohm in impedance.

We attempted to minimize the variability in U-probe placement on a session-by-session basis so that we could average across sessions for the analyses in Fig. 4. Our approach was to place the U-probe so that the most superficial electrodes (electrodes 1, 2 on the 16 channel probe) were able to record multi-unit spiking activity. Any further movement of the electrode upwards resulted in the spiking activity disappearing and a change in the overall activity pattern of the electrode (suppression of overall LFP amplitudes). Similarly, driving the electrodes deeper resulted in multiphasic extracellular waveforms and also a change in auditory markers which were characterized by decreases in overall signal intensity and frequency content. Both markers suggested that the electrode entered white matter. Recording yields and this careful electrode placement were in general better in monkey T (average of ~16 units per session) than monkey O (average of ~9 units per session). We utilized these physiological markers as a guide to place electrodes and thus minimize variability in electrode placement on a session-by-session basis. Importantly, the variability in placement would act against our findings of depth-related differences in activity (Fig. 4). Random placement of U-probes on a day-to-day basis would flatten out the average visuomotor index and dilute or entirely remove significant differences in the discrimination time differences between superficial and deep electrodes.

The technique necessitated a careful watch over the electrode while lowering to ensure that it did not bend, break at the tip or excessively dimple the dura. We, therefore, were unable to use a grid system to precisely localize the location of the U-probes on different days and to provide a map of how laminar profiles change in the rostrocaudal direction. However, we recorded a small subset of units in a more posterior location in the recording chamber (putatively M1, n = 12 sessions, 95 units) that we also report in Supplementary Fig. 12a in this manuscript.

**Description of neural populations included in our database**. We report here the activity of 996 units recorded in PMd of two monkeys while they performed the task (546 units in T, 450 units in O; single neurons and multi-units which responded during any epoch in the task). The majority of the *units* in the database were well-isolated single neurons, and as per convention, we term them single neurons. Some of these single neurons were collected using high impedance (i.e., small electrode contact area, >6 MΩ) sharp FHC electrodes. When using these electrodes, every attempt was made to isolate and track single neurons and to stably record from them.

We also had excellent success recording from isolated single neurons from the U-probes. The U-probes are low impedance electrodes (~100 kΩ) with a small contact area and were thus excellent for isolation of single units. We used a conservative threshold to maximize the number of clearly defined waveforms and minimize contamination from spurious non-neuronal events. Online, we used the hoops provided by the software client for our Cerebus system (BlackRock MicroSystems) to delineate single neurons after the electrodes had been placed in the cortex for at least half an hour to 45 min. Every time a spike was detected by the threshold method, a 1.6 ms snippet was stored and used for subsequent evaluation of the clusters as well as adjustments needed for spike sorting. Recordings from some electrodes in the U-probes consisted of mixtures of 2 or more neurons well separated from the noise and from one another. In a majority of these cases, the waveforms were clearly separated, and these were labeled as single units. We verified these separations visually by viewing the waveforms in a principal component space using custom code written in MATLAB and adjusted the clusters that were isolated online using the MatClust MATLAB-based clustering tool that allows the drawing of polygons to label clusters or the Plexon Offline Sorter. Finally, we had two other types of recordings that we labeled as multi-units. The first category consisted of the multi neuron mixtures that were not separable using a principal components method. The second rarer class consisted of recordings with waveforms only weakly separated from the noise. Both of these types of units were classed as multi-units. We included all of these different units in our analysis here.

To reduce the subjectivity in our assessments we used the number of ISI violations (after clustering and sorting) to distinguish between single and multi-units. We labeled a unit as a single neuron if the percentage of ISI violations (refractory period of <=2 ms) was less than 1.5% and a unit with ISI violations exceeding 1.5% as a multi-unit. Using this criterion, 801 of the 996 PMd units reported here are identified as single neurons (384 from O, 417 from T, mean ISI violation = 0.43%, ~0.13 additional spikes per trial). Similarly, 195 out of 996 are classified as multi-units (129 from T, 66 from O, mean ISI violation = 3.36%, ~1.4 additional spikes per trial).

All main claims made in the paper were unchanged when only restricting the analysis to single neurons (Supplementary Note 6, Supplementary Fig. 13).

**General notes on physiological analyses.** Electrophysiological characteristics were similar in both monkeys, so we pooled data from both monkeys for more power as well as clarity. PMd units were tuned to different reach directions even when recordings were performed in the hemisphere contralateral to the arm. For example, some units recorded in the left hemisphere would be tuned for reaches to left targets; other units were tuned for reaches to right targets. We developed a tuning measure by testing if the FR in the −150 ms to move onset +50 ms time window was significantly different for left vs. right reaches. According to this metric (555/996 units (55.72%) were tuned to reaches to the left, 441/996 (44.28%) were tuned to reaches to the right). Thus, even within the same hemisphere, units were almost equally likely to be tuned to left reach or right reaches (although there is a slight tendency to be more tuned for left reaches overall). Tuning to both left and right reaches was often present within the same U-probe penetration. We, therefore, did not seek to place reach targets for each unit in the preferred and anti-preferred directions. Instead, we fixed the spatial locations of the two targets to the left and right of the central hand target and randomized the target color.

PMd units often switched their tuning preference across time within a trial. In such conditions, as suggested previously[26], we computed the mean absolute difference between FR for left vs. right reaches and plotted it as a function of time. This subtraction provided a "choice selectivity" signal that depends on color coherence (or RT). This is the signal plotted in Fig. 3b, d, g when aligned to checkerboard cue and movement onset. The FR in the hold period is subtracted from the choice selectivity signal for each unit before averaging.

We also occasionally observed in some PMd units a covariation between the hold period FR and overall RT. These effects will be the subject of another study and will not be discussed in this manuscript.

**Trials used in single and population unit PSTHs.** For averaging, we included both correct and incorrect trials and sorted them by the choice of the monkey. However, we excluded a small proportion of change of mind trials (~2–3% of trials) from averages. We chose to do this because, for the change of mind trials, the monkey would initiate in one direction and then change the direction mid-reach. The reaches were thus multiphasic in nature and did not possess a clear initial reach direction making choice attribution for these trials ambiguous. Including them in the averages inflated the variance of FR around movement onset. So we excluded this small proportion of explicit change of mind trials from the averages reported in the Figs. 2, 3, and 4 and the Supplementary Figures. We expect to report the behavioral and neural correlates of explicit change of mind trials elsewhere.

On average across all units, PSTHs/PMTHs rates were estimated using 1194 trials per unit (comprising all conditions and choices) for monkey T (5th and 95th percentiles: 242, 2182 trials) and using 928 trials per unit for monkey O (5th and 95th percentiles: 400, 1644 trials).

For PSTH plots and the analyses shown in Figs. 2–4, trials were separated according to color coherences (seven different color coherences) or by RT (using different overlapping bins 325–425 ms, 350–450 ms … 500–1000 ms). The use of these overlapping bins allows us to better visualize organization with RT. This gave us several curves for each of the plots when sorted by RT.

**General analysis notes.** For PSTHs, we used large numbers of trials as well as several units to compute our statistical tests (>100 units typically for comparisons) and assumed normality because of the central limit theorem. To be confident that potential non-Gaussian nature of the data did not impact our results, we compared medians of these distributions and where appropriate used shuffling and permutation to perform comparisons. No blinding or randomization of animals was performed in this study. All tests were two-tailed unless noted.

**Effects of coherence on accuracy and RT.** We analyzed the behavior using two different methods. Our first approach was to fit psychometric curves, which describe how discrimination accuracy changed as a function of coherence. In the same approach, to test if RT changed with coherence, we used a regression between RT and log coherence as in ref. [45]. Our second approach was to jointly fit both accuracy and RT using a DDM, which provides descriptions of choice and RT distributions[42].

For the analysis of the behavior, we used the same 75 sessions for monkey T (128,989 trials) and 66 sessions for monkey O (108,344 trials) from which we report electrophysiological data. Fits to psychometric curves and RT regressions were performed on a per-session basis and then averaged over sessions. The behavior for an average session was estimated from ~1500 trials. RT was estimated for each session by including both correct and incorrect trials for each SC.

**Psychometric curves for accuracy.** For every experiment, we estimated the monkey's sensitivity to the checkerboard cue by estimating the probability ($p$) of a correct choice as a function of the color coherence of the checkerboard cue ($C$). We

fit this accuracy function using a Weibull cumulative distribution function:

$$p(c) = 1 - 0.5e^{-\left(\frac{c}{\alpha}\right)^{\gamma}}.$$

The discrimination threshold $\alpha$ is the color coherence level at which the monkey would make 81.6% correct choices. The second parameter, $\gamma$, describes the slope of the psychometric function. The mean alpha parameter across sessions is used as the threshold. We fit threshold and slope parameters on a session-by-session basis and averaged the estimates. The mean and standard deviation of the threshold estimates are reported in Fig. 1d. $R^2$ values from the fit are provided in the text.

**RT vs. coherence.** To examine if RT changed with coherence, we adopted the procedure from ref. [45] and used a linear regression between RT and log coherence.

$$\text{RT}(c) = \text{Intercept} + a_c \log_e(c).$$

We fit this regression model and report the slope as well as the $R^2$ values from the fit. $a_c$ is the slope of the regression. We complemented this simple characterization of how RT changed with coherence and fit both choice behavior and RT using a DDM or UGM, which we describe in the following section (Supplementary Note 1, Supplementary Figs. 1–2).

**Fitting behavior using a DDM.** We first fit the behavioral observations of discrimination accuracy and RT with a DDM that proposes the integration of momentary evidence (e.g., noisy spike trains in sensory areas in response to the red squares of the checkerboard cue) to one of two predefined bounds (Supplementary Fig. 1a). Even though our stimulus is a static checkerboard cue, the spike trains in response to static stimuli are time varying in nature and DDMs have been used in other tasks with static stimuli and shown to capture key features of the discrimination behavior[42].

A succinct modeling description is provided here; detailed descriptions of the model are available elsewhere[42]. The DDM is mathematically equivalent to the one-dimensional movement of a particle undergoing Brownian motion to one of two absorbing boundaries. The model provides a quantitative account of both the choice (which boundary was reached) and the decision time (when the boundary was reached) on each trial. The DDM model uses the formulation of a putative decision variable ($V$) that depends on the SC of the checkerboard cue according to the stochastic differential equation:

$$dV = \mu dt + dW.$$

A decision occurs when $V$ reaches a decision bound ($A$ or 0, corresponding to a red choice or a green choice). The quantity $\mu dt$ is the momentary evidence (termed drift rate) and is distributed as a random variable with a Gaussian distribution ($N(\mu, 1)$). The mean of this momentary evidence distribution ($\mu$) is different for each condition. $dW$ refers to Gaussian noise which is added at each time step.

We did not want to make assumptions about how stimulus strength mapped to the mean of the momentary evidence. Moreover, we wanted to fit full RT distributions and not just the mean RTs because very different models can predict the same mean RT[42]. For these reasons, we chose to use the formulation of the DDM for the decision time with the following parameters: slope of the diffusion process for each color coherence level ($v$), threshold separation ($A$), and starting point ($z$)[60]. We also included variability in the drift rate ($s_v$) but not the starting point. Overall both monkeys showed very little red-green bias.

The diffusion model only provides an estimate of the decision time and does not incorporate non-decision times, such as other sensory encoding and motor preparation latencies. The common approach is to consider decision time and other residual times as independent additive contributions to the total RT. Therefore, mean RT for every condition is the sum of the mean decision time and the non-decision time ($t_0$). We also assumed trial-to-trial variability in this non-decision time parameter ($s_{t0}$). This improved overall RT fits but tended to increase the mean non-decision time (fits with this parameter often led to a 40–60 ms increase in the mean non-decision time, and we report this number in the main text). Our decision times are therefore better understood as a lower bound. We chose to report fits with this parameter included in the main text because they provided the best fits to the overall RT distribution as measured by the $\chi^2$ goodness of fit. The goodness of fits for other comparison models that excluded these parameters are shown in Supplementary Table 1. The parameters from the best fit DDM are shown in Supplementary Tables 2 and 3.

Finally, non-decision times are often assumed to be equal for both choices. However, this assumption may not apply if other cognitive factors are involved. For example, if one of the response types occurs more frequently, or is perceived to be more likely in a given situation, it might result in a faster motor response[42]. Overall, the fits to the RT data shown in Supplementary Fig. 1b–c were better when including the static parameter ($d$) which shifts the non-decision time for one choice by this amount. This use of variable non-decision times for different responses is often used in decision-making studies to explain RT.

Model fits were performed using the fast-dm toolbox with minimization of the $\chi^2$ criterion which is robust with large numbers of trials[42]. For input to the model, we included all trials pooled over all sessions (correct as well as incorrect) and labeled it with the choice the monkey made as well as RT (128,989 trials for T and 108,344 trials for O). The $\chi^2$ goodness of fit for the DDM from this best fit model can be found in Supplementary Table 1. The parameters from this best fit model are provided in Supplementary Table 2.

Plots of decision times in Supplementary Fig. 1d were generated by subtracting the estimates of non-decision times from the mean and 95th percentiles of the RT distributions. The 95th percentile of the RT distribution provides an upper bound on the decision times used by the monkey and facilitates comparison to the data from variable duration tasks. For our monkeys, approximately 35–40% of trials lie between the mean and the 95th percentile of RTs.

**Comparing fits from the UGM and the DDM**. We also compared fits between a DDM and an UGM[44, 48, 49]. For this comparison, we used a toolbox developed as part of a recent study that investigated the differences between these two models[47]. This toolbox is available on request from Dr. Guy Hawkins. The models being compared here are a DDM with variability in drift rate[44, 47] and an UGM that incorporates variability in drift rate and an urgency signal[44]. The time constant for the UGM was chosen to be 100 ms[44]. An extensive description of these models is provided in ref. [47] and here we only briefly describe the models.

We used the same basic framework from refs. [44, 47] to implement our DDMs and UGMs. The first variant of the UGM was implemented by an urgency function that was linear in time ($U(t) = b \times t$, $b = 1$) and included a low pass filter with a time constant of 100 ms[44, 47]. Further details of model implementation are provided in ref. [47]. We assumed variability in the drift rate for both the DDM and the UGM. Both models have equal numbers of estimated parameters, and thus goodness of fit can be compared without any penalization for model complexity. The parameters estimated for the UGM (DDM) and the parameters kept constant are reported in the Supplementary Table 4 and replicated from ref. [47]. We also explored a second variant of the UGM that included an additional intercept term ($U(t) = a + b \times t$, with $b = 1$). The inclusion of this additional intercept parameter allows the UGM to model fast RTs observed in discrimination tasks.

The models were implemented in a combination of R and C++. To ensure that we were correctly performing the model fitting, we first fit the behavior of the monkeys from Roitman and Shadlen[45] and verified that the QMPE statistics[72] were near identical (small variations occur due to the random number generators in different iterations) to the statistics reported for this data set in ref. [47]. We then applied this analysis to the behavioral data collected in this study.

**Peristimulus/perimovement time histograms (PSTH/PMTH)**. For each unit, we calculated mean FR over trials to define a PSTH/PMTH over time. Trials were aligned to several events such as checkerboard cue onset, target onset, and movement. Spike trains for each trial were binned in 1 ms bins, convolved with a causal boxcar and averaged over trials depending on condition identities. For single examples shown in Figs. 2, 4 and Supplementary Fig. 4, we used a 75 ms causal boxcar. For population data, we convolved the spike trains with a 50 ms causal boxcar. When aligned to checkerboard cue onset, for both single examples as well as population FR plots, we removed spikes starting 100 ms before movement onset until the end of the trial and assigned them to not a number (NaN). Similarly, when aligned to movement onset, we used the RT and set all spikes before the onset of the checkerboard to NaN. We then averaged across the trials and reported the mean and standard error across trials (for single examples) and neurons (population level).

**Visuomotor index for separation into unit classes**. We partitioned our unit database into a continuum of unit classes by developing a visuomotor index based on the observation that before movement onset there are systematic FR modulations in some units that covary with RT. We exploited the existence/absence as well as the direction (increases or decreases) of this main effect of RT to define the index.

The index is estimated as the trial-by-trial correlation between the RT and time averaged FR in the −600 to −200 ms epoch before movement onset. We only included FR until 200 ms before movement based on two observations. First, most perimovement units often began modulating their responses around that time. Second, when using the regression analysis, the number of units with RT as a significant predictor at −150 to −200 ms before movement onset began to approach chance levels (Supplementary Fig. 10a).

Before computing this correlation coefficient, we first subtracted the hold period FR (from the 200 ms epoch before checkerboard cue onset). Then for each trial, to ensure that FRs before checkerboard cue onset do not bias the index, we set spikes before the onset of the checkerboard relative to movement onset to be NaN. We then averaged the FR (across time) from −600 to −200 ms for each trial and then correlated this quantity to the RT. For $n$ trials of one unit, we had $n$ RTs and $n$ time-averaged FRs. Both the left and right choices were included in the computation of the correlation. The correlation coefficient was significant if the 90% confidence intervals ($p < .05$, for each tail) estimated by bootstrapping did not overlap with zero. Changing confidence intervals to 95% did not result in a qualitative change in the results.

The perimovement units were defined as the ones with insignificant indices. We were concerned that some of the decreased units, which had smaller values of the index, could be mistakenly classified as perimovement units and vice versa. To address this concern, we used the Bayes Factor method that provides the ratio of the likelihood of two competing hypotheses or models[73] and is typically interpreted as evidence for one model over the other. In our case the ratio is between classifying a unit as decreased (or increased, $H_1$) vs. perimovement ($H_0$)[73]. A large value of Bayes Factor suggests that model $H_1$ is more likely than the model $H_0$ and in our case would provide strong support that the unit is correctly classified as increased or decreased. In contrast, a low Bayes Factor would suggest strong evidence for $H_0$ and support the classification of the unit as perimovement in nature. We examined the Bayes Factors computed for the different broad unit categories and found that the Bayes Factors for the units classified as perimovement based on the visuomotor index were very low suggesting that they were correctly classified (Supplementary Fig. 3b). The Bayes Factors computed for perimovement units also had very little overlap with Bayes Factors for both the decreased and increased units (Supplementary Fig. 3b)[73]. This suggests that there was minimal contamination of the perimovement group by the decreased units and vice versa.

**Slope analyses**. For every level of color coherence, we regressed the 150–350 ms epoch of the boxcar smoothed choice selectivity signal (e.g., Fig. 3b) to time to estimate the rate at which the choice selectivity increased in this epoch (plotted in Fig. 3c for each color coherence level). Using 150–250 ms after checkerboard cue onset (which would suggest an even shorter average decision time of 100 ms) did not change our conclusions.

**Population timing analyses**. We identified the time at which the population activity significantly changed as a function of the choice of the monkey by using a paired $t$-test comparing responses during the epoch after checkerboard cue onset to the responses of the same neurons during the hold period (−500 ms to 0 before checkerboard cue onset). This latency for choice selectivity is termed discrimination time and is plotted in Fig. 3e. We chose a paired $t$-test for this analysis because the decreased neurons did not have a significant choice latency with more stringent measures. The discrimination time was identified as the first time point after checkerboard cue onset for which at least 50 ms was significantly different from hold period FR (paired $t$-test vs. hold period FR, $p < .05$ false discovery rate corrected). When aligned to checkerboard cue onset FR from 100 ms before movement onset onward were set to NaN. To derive error bars around this estimate of the time at which significant choice selectivity emerged, we bootstrapped trials for each unit to come up with 50 surrogate distributions of FRs for each unit. We then reran the timing analyses and used the standard deviation of the bootstrap distribution to estimate the error in our estimate of the discrimination time. Alternative methods by bootstrapping over units yielded similar results.

**Estimate of the discrimination time for each single neuron**. We also performed the same analysis identifying the discrimination time on a neuron-by-neuron basis. We used the 95th percentile of the bootstrap estimates of baseline FRs for each neuron to identify the time at which the neuron demonstrated significant choice selectivity. The FRs should be significantly different as a function of choice for at least 50 ms after this first identified time to be included as an estimate of a time of significant choice selectivity. The analysis was performed aligned to both checkerboard cue onset and movement onset. Data are shown aligned to checkerboard cue onset in Supplementary Fig. 4g–i.

We computed the slope of this discrimination latency vs. RT when aligned to both checkerboard cue onset and movement onset and used this to perform the analysis described in ref. [9] and in Supplementary Fig. 7b. The mean slopes aligned to checkerboard cue onset and aligned to movement onset are shown in Supplementary Fig. 7c, d, respectively.

**Neural response latency ($\lambda$) and covariation with RT ($\beta$)**. We also used the method developed in DiCarlo and Maunsell[33] to characterize our neural population. This method helps determine the response onset (termed Neuronal Latency or NL, $\lambda$) and is thought to estimate where in the "processing chain" the neurons might operate. Smaller values of $\lambda$ suggest that the neuron is more sensory (visual in our case) and larger values of $\lambda$ signify that the neuron is closer to the motor output.

The method also provides an estimate of the covariance between the neuronal latency and the RT ($\beta$). $\beta$ is unitless and ranges from near zero for neurons whose FRs have no correlation with RT (neurons involved in the early stage of information processing) to near unity for neurons with activity closely correlated with the timing of the behavioral response (the RT). Note that $\lambda$ and $\beta$ do not distinguish whether the neuron increased or decreased it FR after checkerboard cue onset. Code and a Jupyter notebook explaining this technique are available at https://github.com/mailchand/ephysutils.

**Unsupervised clustering of PMd FRs**. Another analysis we performed was inspired by a recent study that used a clustering technique to suggest the presence

of diversity in neural responses in LIP during a perceptual decision-making task[26]. Using the tuning metric measured around movement onset (described above), we first identified for each neuron a FR averaged over all trials in the preferred direction and a FR over all trials for the non-preferred direction aligned to the checkerboard cue onset. We included 100 ms before checkerboard cue onset and 600 ms after checkerboard cue onset in our averages. We then concatenated these FRs and then performed a k-means clustering analysis.

To decide on k, the number of clusters to show, we used the gap statistic method[74]. The gap statistic quantifies the change in within-cluster dispersion with that expected under an appropriate reference null distribution[74]. The number of clusters is chosen to be the minimal value of k that involves the largest separation between the true distribution and the reference distribution. With very small numbers of clusters, the difference between the within group dispersion and the reference distribution is low. At some point, this difference increases and roughly saturates. The first such value is a good indication of a reasonable clustering solution for the data[74].

**Principal component analysis (PCA) of PMd FRs.** We also used an alternative unsupervised method, PCA to examine the FR variance recorded in our neural population. The FRs of all units were first "softmax" normalized by dividing the FR for each unit across all conditions by the range of the FR for the neuron across all conditions. Softmax normalization reduced the bias induced by high FR units. This ensures that each unit has roughly the same overall variability across conditions. Dimensionality reduction was performed on a FR matrix of dimension (units $(n) \times \sum t_i$), where $n$ is the number of recorded units, and $t_i$ denotes the number of time points for the ith condition over time. We performed PCA on this space to reduce these dimensions to $k \times \sum c_i \times t_i$, where $k$ represents the dimensions across which the most neural variance was explained.

PCA makes few assumptions about the underlying structure of the data, simply revealing dimensions that explain a large percentage of the variance. PCA does not guarantee that the dimensions extracted are meaningful. However, often they can align quite well with behavioral variables of interest as seen in Supplementary Fig. 8. Other methods such as factor analysis can be used. However, they often require additional assumptions about the data. Several studies have employed PCA as a dimensionality reduction technique for trial-averaged data[63]. The RT nature of the task precluded us from using techniques such as targeted dimensionality reduction[23].

**Examination of loading matrices from principal components.** Analysis of the loading matrices from PCA provided us with additional credence to the claim that there are broad unit populations in PMd that demonstrate a specific relationship to the decision formation process. We tested if the distribution of loadings in this two-dimensional space was consistent with a uniform random distribution by using a $\chi^2$ test that assumes an equal number of neurons in each octant or by using the more sophisticated PAIRS (Projection Angle Index of Response Similarity) test developed in a recent study[27].

The PAIRS test compares each neurons loading on the first two (or more) principal components with its "nearest neighbors" in the principal component space by computing the angle between the vectors. If there is a nonuniformity in the loading in this space, then the distribution of angles would not be uniform, and a median of this distribution is compared to the median from 10,000 simulated data sets from a two-dimensional Gaussian distribution. The simulated data sets are used to estimate a p-value for this statistic.

**Estimating correlation between RT and neuronal responses.** We also adopted an analysis method developed for heterogeneous neural populations by Erlich and collaborators[53]. This method measures the correlations between RT and neuronal responses by using an alignment algorithm to find a temporal offset for each trial that would best align that trial's FR with the average over all the other trials.

The algorithm first computes a trial-averaged FR aligned to a marker of interest (in our case movement onset). The FR of every trial is then cross-correlated with the trial-averaged signal, and the location of the peak of the cross-correlation is measured as the offset needed to align the trial to the average. The trial was then shifted accordingly, and the trial-averaged FR was recomputed. This process was then repeated until the variance of the trial-average signal converged, typically within less than 10 iterations for a criterion of 1% variance. The final output of this alignment procedure was an offset that was then correlated to RT.

When aligned to movement onset, the expectation is that the perimovement units would require minimal shifts and would thus show the lowest correlation between the offset time and RT. In contrast, the increased and decreased units would show a larger degree of shift, and this would emerge as a significant correlation. Because the analysis was performed aligned to movement onset, longer RT trials would require a positive shift (to match the average). The shorter RT trials would need a negative shift resulting in a significant positive correlation between RT and offsets estimated from the algorithm.

One limitation of this metric is that it does not provide any indication of whether the neuron was decreasing its FR during the decision-formation process or increasing its FR. To estimate the direction of FR change, we examined the FR in the 100 ms epoch before movement onset and compared it to the FR in a reference epoch (−800 to −700 ms before movement onset). If the perimovement period FR was lower than the reference epoch FR, we multiplied the sign of the correlation coefficient estimate from the offset method by −1. We then correlated this modified coefficient to the visuomotor index we proposed.

**Classification analysis.** We also examined if the superficial layers (electrodes 1–8) of PMd signaled choice earlier than the deeper layers (electrodes 9–16) by using linear classifiers. A separate naïve Bayes classifier (from classify in MATLAB) with 10-fold cross-validation (over trials) was used for the superficial and the deep layers. For classification analysis, we only included sessions where we had more than 10 units recorded for this classification analysis to give us sufficient power and be confident in the classification results. This criterion left us with 19 sessions in monkey T and 12 sessions in monkey O. To reduce the bias in the classification results we equalized the number of units for each classifier to be the same. When either the superficial or deep had more units, we randomly selected from the greater of the two pools, a number of units that was equal to the number of units in the other pool. We repeated this selection 10 times and then averaged over both folds (10 folds) and selections. All RTs from 300 to 1000 ms were included in this analysis. Input to the classifier was 50 ms boxcar spike counts and retrained on each time point. We chose a 2 ms spacing between adjacent time points for the classifier.

To estimate the reduction in discrimination time for the superficial compared to the deep layers, we used a threshold of 55% accuracy and estimated the first time at which the accuracy increased above this value for the superficial and deep layers. We then defined the benefit as $t_{Deep} - t_{Superficial}$ and used it as our estimate of the discrimination time delay between the superficial vs. deep layers. We also estimated through bootstrapping (10,000 repeats) by randomly drawing with replacement from this population of sessions and obtained a distribution of these delays. We report the mean of this randomly sampled distribution as our estimate of the reduction in discrimination time. To estimate if this reduction in discrimination time was significant, we used the bootstrapped distribution and estimated the fraction of delays that were lower than 0 and report this as the p-value for the reduction in discrimination time.

**Control: regression analysis.** We assumed that FR at each time point is a linear combination of several predictors such as RT, reach direction choice, the color of the target chosen, color coherence, interaction terms between these predictors, as well as several potential nuisance variables related to the arm and eye movements. The regression equation for the ith trial when aligned to movement onset was as follows:

$$
\begin{aligned}
r_i(t) = \ & \beta_0 + \beta_{coh} \times coh_i + \beta_{choice} \times choice_i \\
& + \beta_{color} \times \text{Chosen Color}_i + \beta_{RT} \times RT_i \\
& + \beta_{handX} \times X_{handX,i}(t) + \beta_{handY} \times X_{handY,i}(t) \\
& + \beta_{EyeX} \times EyeX_i(t) + \beta_{EyeY} \times EyeY_i(t) \\
& + \beta_{speed} \times (\text{Hand Speed}_i) \\
& + \beta_{RT \times coherence} \times (RT_i \times coh_i) \\
& + \beta_{RT \times choice} \times (RT_i \times Choice_i).
\end{aligned}
$$

The different predictors were first Z-scored and then input into the regression. The predictor variables are defined as follows:

$\beta_0$ − Intercept term

$coh_i$ − Coherence on the i$^{th}$ trial

$Choice_i$ − Choice of the monkey on the i$^{th}$ trial(L, R)(−1, 1)

$\text{Chosen Color}_i$ − The color of the target touched by the monkey on the i$^{th}$ trial (−1, 1)

$RT_i$ − Reaction time for the trial

$X_{HandX,i}(t)$ − Hand position in X for i$^{th}$ trial and as a function of time

$X_{HandY,i}(t)$ − Hand position in Y for i$^{th}$ trial and as a function of time

$X_{EyeX,i}(t)$ − Eye position in X for i$^{th}$ trial and as a function of time

$X_{EyeY,i}(t)$ − Eye position in Y for i$^{th}$ trial and as a function of time

Hand Speed$_i$ − Speed of the reach on the i$^{th}$ trial

We used 95% confidence intervals at each time point to estimate if the betas for predictor variables were significantly different from chance. We used this to generate plots of the number of significant units as a function of time aligned to movement onset (Supplementary Fig. 10a, b).

**Code availability.** All analyses were performed using custom code written in MATLAB to process spike trains and perform statistical analyses. Code used in this paper is available on request. Some of this code is already available in github repositories for which links are provided in the paper.

**Data availability**. All data examined in the paper are available from us on request.

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

## Acknowledgements

We would like to thank Dr. Guy Hawkins for guidance and R code for fitting Urgency Gating and Drift Diffusion Models and Prof. Jeffrey Erlich for assistance and code for implementing analyses examining correlations between RT and FRs[53] (code available at https://github.com/erlichlab/elutils). We would also like to thank Ms. Megan Wang and other members of the Shenoy Lab for helpful discussions and comments on earlier versions of this paper. The views and comments from three anonymous reviewers greatly improved the paper. C.C. was supported by an NIH/NINDS K99/R00 grant NS092972.

D.P. was supported by the Champalimaud Foundation, Portugal, and Howard Hughes Medical Institute. B.N. was supported by the Howard Hughes Medical Institute. K.V.S. was supported by the following: Christopher and Dana Reeve Paralysis Foundation, Burroughs Welcome Fund Career Awards in the Biomedical Sciences, Defense Advanced Research Projects Agency Reorganization and Plasticity to Accelerate Injury Recovery N66001-10-C-2010, US National Institutes of Health Institute of Neurological Disorders and Stroke Transformative Research Award R01NS076460, US National Institutes of Health Director's Pioneer Award 8DP1HD075623-04, US National Institutes of Health Director's Transformative Research Award (TR01) from the NIMH #5R01MH09964703, Defense Advanced Research Projects Agency NeuroFAST award from BTO #W911NF-14-2-0013, the Simons Foundation, and the Howard Hughes Medical Institute.

## Author contributions

All authors contributed extensively to the experimental design and analyses presented in this study. C.C. trained animals, performed all electrophysiological experiments, analyses, and wrote initial drafts of the paper. All authors contributed analytical insights and commented on statistical tests, discussed the results and implications, participated in interpretation, responded to criticisms, and helped finalize the drafts of the paper.

## Additional information

**Competing interests:** The authors declare no competing financial interests.

