## [Peer Review File · Nature Communications]

Reviewers' Comments:

Reviewer #1 (Remarks to the Author):

note: comments also attached as formatted word doc for easier reading.

Laminar differences in decision-related neural activity in dorsal premotor cortex

Summary

The authors trained 2 monkeys to perform a visual discrimination reach task. Despite using a static stimuli, the monkeys behavior was well fit by a drift-diffusion model (DDM). Recordings from PMd revealed diverse neuronal populations, with some being more highly correlated with reaction time and others less correlated. The authors claim that the units that were positively correlated with reaction time and also choice and stimulus difficulty. But the units that were negatively correlated with RT did not show "decision-variable" like characteristics. Using a laminar probe the authors claim that superficial layer contain more of the "increasing" neurons (more correlated with RT.)

General Comments

The laminar organization of cortical function/computation is of great interest to neuroscientists. This paper is one of the first to describe laminar distinctions in a perceptually difficult reach task, which (in theory) permits the separation of perceptual, decision, and action related signals.

There are a few problems that prevent me from recommending the paper for publication in Nature Comm. in its current form.

- 1) The behavior of the two monkeys is quite different, but all the analyses lump the neurons together, calling into question the generality of the results. The authors need to show that the main results and claims hold up independently in both animals.
- 2) The logic of the paper is convoluted. I feel that it could be substantially improved with new analyses and writing. I detail the issues below, but the main problem is with the overuse of the "index" (correlation between pre-movement firing rate and reaction time).

3) Of the 3 "concrete goals" the authors describe in the discussion, only the third one (laminar distribution of decision encoding) merits enough interest for publication in Nature Comm. However, the authors did not perform any analyses of the simultaneously recorded units that would provide stronger evidence that the decision-related activity in superficial (and also the deepest) layers of cortex drive the more movement related neurons.

Major Issues

The index used for quantifying a neuron's position along a visuo-motor continuum is unconvincing. It turns out that for the "decreased" population, choice signals emerge later, but it did not have to turn out that way, thus the index is not itself the property of interest. Rather than report the laminar distribution of the "index" why not directly report the laminar distribution of a perceptual, decision and motor signals, since it seems that that is what the authors are more interested in reporting anyway. The last sentence of the results: "At the population level, choice selectivity appeared earlier in the superficial compared to the deep units".

I would recommend to divide the neurons by depth and then use a decoding analysis to reveal the perceptual/decision/motor information content as a function of time at different depths. The authors rely heavily on firing rate differences (using regression / ttests) but something more sophisticated (see for example papers from Pagan and Rust on decoding information in IT) would improve the manuscript.

A major reference for the "visuo-motor continuum" type analysis is DiCarlos and Maunsell 2005 "Using Neuronal Latency to Determine Sensory-Motor Processing Pathways in Reaction Time Tasks". That paper provides a more robust technique for doing neural-behavior latency analysis (rather than just comparing spike-rates with RT). Another technique for neural-behavior latency analysis is described in Erlich, Bialek & Brody (2011) which is applicable to neurons with more complex dynamics.

Given that the behavior of the two monkeys is quite different, it is important to show that your main results (that "increased" neurons have earlier decision information and are more superficial) replicated independently in both animals. Figure 3e, 4b, Extended Data 3.

****Use of parametric statistics.**** I don't think this would change the main results of the paper, but I would encourage the authors to use non-parametric statistic (like permutation tests) rather than rely on t-tests which have assumptions about the underlying distribution of the data. Especially since the authors do not check (or at least did not report) whether those assumptions were met. For example, simultaneously recorded neurons do not satisfy the independence assumption. Neurons generally have autocorrelations which violate the assumption of independent samples for regression. Firing rates may or may not satisfy the assumption that the data came from a normal distribution.

Some detailed comments

Line 15 - "... are not well understood"

: This phrasing has unfortunately become a universal technique for general motivation of any paper. I wish the authors could be a bit more specific in the abstract as to the hypothesis that they are testing. Your paper is about the laminar organization. So something like "we have previously described the response properties of neurons in PMd using techniques that did not allow us to identify the laminar location... "

Lines 19 - 25

: On line 19 "firing rates of PMd neurons are organized" but at line 24 "units at the center". Is it a continuum of units or of firing rates? If it a continuum of units then you should write (on 19): "Units in PMd can be organized along a continuum by their responses in our reach task." (or something to that effect.)

Line 19 "Consistent with a model, which"

: Remove comma after model

Figure 2: "Check"

: I would encourage the authors to just write "cue" rather than "check" in Figure 2. "Check" is not a standard short form for "checkerboard", but 'Cue' is pretty easy to interpret (especially if you use that term in fig 1 and the rest of the text... e.g. the "checkerboard cue".)

Figure 3e: Regression

: The regression should be done on the actual data, not the binned and averaged data. Why did the authors do that? This figure needs to be made for each animal separately.

Line 88-90: "..the DDM framework applies even when sensory evidence is 90 available all at once"

: This is not news. You cite two (relatively recent) papers. But much of the early psychophysics work related to the DDM is based on static visual discrimination. What is unique about your design?

Figure 4B

: Why use a line to fit this when it clearly looks non-monotonic? A quadratic fit (or two lines) seems more appropriate.

Reviewer #2 (Remarks to the Author):

Chandrasekaran et al. present an analysis of multi-unit recordings from laminar probes in premotor cortex during a perceptual decision making task. They found that some units whose activity positively correlated with reaction times, activity bears signatures of a decision variable as predicted by a drift-diffusion-type model. Depth measurements from the laminar probes revealed that decision-related units were preferentially distributed in the superficial layers, whereas units in the deeper layers reflected more motor-preparatory responses.

I find this to be an interesting and compelling study that provides a novel advance to our understanding of the cortical circuit dynamics during decision making. The analyses are thorough and well-designed, and the figures and text are carefully and clearly presented. The laminar dependence of decision vs. motor signals in primate premotor cortex is of particular novelty and importance, as they suggest a circuit-level transformation in the flow of information across layers during decision making. I have only two comments/concerns, which should be readily addressable, described below.

Comments:

1. Relationship between visuomotor index and choice selectivity: What is the relationship across units between the visuomotor index and choice selectivity? Fig. 3C speaks to this question at the group level, averaging within decreased, perimovement, and increased divisions, but not at how these may be correlated or clustered at the unit level. This could be shown with a scatter plot showing, for each unit, the visuomotor index and some measure of choice selectivity (e.g. for each unit, the immediate pre-movement difference in firing rates or slopes between choices, averaged across coherences, and maybe normalized by baseline/overall firing rate). This analysis could address the important question of whether this sample can be described a unimodal distribution showing a range of choice selectivity, or whether there is clustering with some units showing very little or very high choice selectivity. For instance, the example units at channel 11 in Fig. 4A show very little choice selectivity at the time before movement. In that sense they resemble the “fixation” neurons recorded in the frontal eye field (by Jeff Schall and others), which show omni-directional decreases preceding movement. So is there a subgroup of units that show a strong choice signal, beyond the expected continuum of a unimodal distribution?

2. Potential limitations of multi-unit analysis: A key limitation of the current work is that the analyses are done on only multi-unit data, without any validation that the main conclusions would hold if analyses were applied to only isolated single-neuron recordings. Indeed, with this limitation it is not even clear how one should interpret the selectivity and visuomotor indices of these units, when multiple neurons are combined to produce an analyzed unit. The Discussion section should note this limitation and describe what aspects of the conclusions may be sensitive

to the multi-unit vs. single-neuron distinction or which specific aspects would benefit from analysis of single neurons.

3. DDM parameters: It would be of definite interest to some readers to have the DDM fit parameter values provided in a supplementary table.

Reviewer #3 (Remarks to the Author):

This manuscript describes a study examining neural activity in dorsal premotor cortex (PMd) during a perceptual discrimination task. Monkeys were trained to determine which of two colors was predominant in a mixed color cue and indicate their choice by reaching to a target of the corresponding color. Recordings were made in caudal PMd using multisite linear probes. There were three aims: 1) To characterize the variety of firing patterns in PMd during this task, relating them to putative decision variables; 2) to test whether these firing patterns, and monkey behavior, conformed to the predictions of the drift diffusion model (DDM); 3) to examine whether there are specific trends in the distribution of different firing patterns in different cortical lamina. Three corresponding main results are reported: 1) PMd cells can be characterized as lying along a continuum that is indexed by their trial-to-trial relationship to reaction time. At one end there are “increasing cells” whose movement-aligned activity is larger when reaction times are longer, and which are most clearly related to the difficulty of the decision. At the other end are “decreasing cells” whose activity is larger for shorter reaction times, and which do not reflect the choice until just before movement onset. In-between there are “perimovement cells” that do not correlate with RT and simply indicate movement direction about 150ms prior to onset. 2) The activity of increasing cells, and the monkeys’ behavior, is consistent with the DDM. 3) There is a clear trend in the laminar distributions of the cell types, with increasing cells more common in superficial layers, while decreasing and perimovement cells are more common at deeper layers.

I will comment on each of these aims and findings in turn, but I will begin with aims 1 and 3, as they are related, before turning to aim 2.

Many previous studies have reported similar kinds of trends in PMd neural firing patterns, especially by Coallier et al. 2015 and Song & McPeck 2010 (both cited), but also earlier studies by Evarts, Tanji, Wise, Kalaska, Caminiti and others. These authors have described similar types of “stimulus-related” or “motor-related” cells, using a variety of indices and methods for quantifying response patterns. Similar trends have also often been reported in other regions, notably including FEF (e.g. Thompson et al. 1996 cited; as well as many other studies by Schall and colleagues, see in particular Sato & Schall 2003). The authors acknowledge all of this, but then it’s unclear why they state that a “detailed description of the temporal patterns” is lacking (p. 2), and unclear how their analyses provide a description that is more detailed than what was already shown previously.

Having said that, I do recognize that the present paper quantifies the specific relationship with decision-related variables, which has not been done before. However, the index that is used is perhaps not ideal to tease that apart from other stimulus-locking or burst vs. tonic properties. The index is straightforward and reasonable, but it could potentially confound important phenomena. In particular, the FR is averaged across a lengthy period of time during which cells exhibit temporal changes, some increasing over the course of the trial and some decreasing. The classification of a cell as having a positive index could be caused by several different scenarios: 1) it could reach a similar level of activity but for a longer time prior to movement onset in long RT trials (perhaps simply because it is “stimulus locked” and “tonic”; 2) it could exhibit a burst that always lasts the same amount of time but is stronger in long RT trials; 3) it could build up as a function of elapsed time independently of difficulty. All of the examples shown illustrate scenario 1, but were other scenarios observed? If so, how often? A negative index could likewise be produced by different scenarios, and it is very conceivable that stronger bursts are associated with shorter RTs (and/or higher velocities). The index might therefore inappropriately partition cells that are in fact closely functionally related, or group together cells that are very different. It seems to me that one needs to be more careful in characterizing these trends (aim 1), possibly through the use of multiple indices that measure different aspects of cell responses (e.g. duration of burst, amplitude of burst, time-locking with stimulus or movement onset, etc.). The analyses used by Sato & Schall and Song & McPeck, examining the relationship between RT and neural discrimination time, could be particularly insightful here, but only as one among several ways of quantifying response patterns.

Assuming that the trends in response patterns are well-characterized, mapping them to different cortical layers (aim 3) is novel and very interesting. In fact, I consider this to be the most significant advance presented in this paper. The result in Figure 4b is compelling, and it would be very good to show that for individual recording sessions, perhaps even as a 3-D map that examines how robust these trends are along a rostral-caudal or medial-lateral axes. Does the prevalence of superficial increasing cells decrease as you move caudally? Do they appear at different depths? It would also be desirable to have more detailed reconstructions so as to be able to judge what specific cortical layers correspond to the different depths. The online methods describe the careful placement of the probe to maximize consistency across sessions, but this cannot do anything about the different angles at which the probes intersect the curving cortical surface. A histological reconstruction would be best, but if the animals are still being used in experiments then perhaps a careful 3D reconstruction based on MRI data would suffice. I emphasize this because the laminar trends described here would be even more valuable if they could be more precisely related to specific cortical layers.

Regarding aim 2, the paper is rather underwhelming. The authors contrast their findings with a study (Thura & Cisek 2014), in which it was claimed that PMd activity is not compatible with the DDM but instead quickly tracks sensory information and combines it with a growing urgency signal. Why is that explanation not compatible with the present data? The central assumption of the DDM is that build-up of neural activity is caused by an integrator with a long time constant,

and Thura & Cisek argued that their data suggests the PMd time constant is short. Here, the authors argue that their data is compatible with the DDM (i.e. an integrator with a long time constant). But they can't actually test that proposal because it is not possible to distinguish between a long and short time constant with a task, like the present one, in which the stimulus information is static. All of the data shown here, and all of the fits, could be done with a variety of settings of time constants and urgency signals. The authors propose that perhaps the brain adjust time constants in a task-dependent way, using long ones when the stimulus is static. This is plausible, but it has also been claimed that time constants are short even for the classic random-dot motion task (Carland et al. 2016). Regardless, no case can be made either for or against the DDM using data, like that presented here, which does not actually test the DDM. Consequently, I would recommend that the authors just remove the DDM from the manuscript, including the discussion as well as the model fits. It merely distracts from the main contribution, which (at least in my mind) is the characterization of differences in activity patterns at different cortical layers. Furthermore, given the ambiguous status of the DDM as an explanation for this kind of experiment, I would recommend not basing any of the analyses on fits to that model. This specifically pertains to estimates of decision times, which are derived from parametric fits that assume the DDM as the correct model, even though it is not tested. In fact, the non-decision times of 326 and 360ms for the two monkeys, respectively, are quite long. If that is the sum of sensory plus motor delays, then it would be hard to explain how the monkeys could perform a simple reaction time task with an RT that is less than 300ms, which presumably they often do.

Specific comments:

Line 168: It is reported that the neural data comes from 546 units in monkey T and 450 in monkey O, and that this includes both single units and multi-units. Why include the multi-units? One critical prerequisite to addressing aims 1 and 3 is to characterize the diversity of firing patterns of real neurons, and so averaging across neurons is exactly the wrong thing to do. It only raises the concern that many important phenomena were missed.

Line 176: It is interesting that stimuli with higher coherence were associated with earlier divergence of FRs, but the statistics that test this should be reported (In fact, the strength of this correlation would make a very interesting index for characterizing response patterns, as noted above). Similarly, statistical tests should be reported for all of the other claims made as well.

Response to reviewers for “Laminar differences in decision-related neural activity in dorsal premotor cortex”

We thank all three reviewers for the positive evaluation of the novelty of the results presented here. We are also grateful to all the reviewers for the insightful comments and the suggestions that we believe have greatly improved the manuscript. In this revised manuscript, we have made every attempt to address these comments and clarify the novelty as well as strength of the results presented in the manuscript. Please find below the comments from the reviewers in **black** and our response to the reviewers in **red**. The text/analysis changes we have made in the manuscript to respond to the reviewer comments are provided in blue in this document (as well as in the manuscript). Before addressing each comment, we provide a brief preamble to list the major additions or clarifications we have made in the manuscript.

In our original manuscript, we relied heavily on the index as the primary way to separate neuronal firing rates because we believe that it is simple and readily understood. We did not include in the original manuscript other analyses we had performed because of the worry that it may be unwieldy or overwhelming for the reader. However, we agree with the reviewers that adding additional depth to these analyses improves the impact, readability, and relevance of this manuscript. We have attempted to perform the appropriate analyses and include them in the manuscript in the appropriate location. The new analyses included in this revised manuscript are described below here first and also when appropriate for the questions separately for each reviewer.

1. Performed a neural discrimination time analysis on *single neurons* for a range of RTs and show that discrimination time correlates with the visuomotor index and other metrics of interest (Extended Data Fig. 4d-f). Choice selectivity in both monkeys emerges first for the increased neurons and later for the decreased and perimovement neurons (Extended Data Fig. 4d-f). Choice selectivity as measured by this single-neuron analysis emerges earlier for the superficial compared to the deeper layers of PMd (Fig. 4d, Extended Data Fig. 11d).
2. Added a decoding analysis to demonstrate that information about choice is available earlier in superficial compared to the deeper layers of PMd. This analysis was performed by separating out electrodes according to superficial (electrodes 1-8) and deep (9-16) and decoding choice using a linear classifier (Fig. 4e, Extended Data Fig. 12b).
3. Performed analyses that estimate “Normalized Neural Latency (λ)” and “Covariation with RT (β)” from DiCarlo and Maunsell (2005). We correlated these metrics to the visuomotor index used in this manuscript and show that it provides very similar results (Extended Data Fig. 5).

- 4. Performed the correlation analyses to relate neural responses to RT from Erlich, Bialek
and Brody, 2011. We show that these new correlations are consistent with the
visuomotor index we used (Extended Data Fig. 6).
- 5. Applied Principal Component Analysis (PCA) to examine if our separation into these
broad unit classes is justified and to show that “increased” units have stronger
projections on a principal component that strongly covaries with choice compared to the
decreased and perimovement neurons. We then used the PCA analysis to quantify the
variance explained by our index. We found that Principal Component 1 (PC1) and the
index were highly correlated and PC1 explained ~50% of the variance. Thus the index,
by proxy explains nearly 50% of the variance in the FRs (Extended Data Fig. 8).
- 6. Included a clustering analysis on the trial averaged firing rates for the population of PMd
neurons as performed by Meister et al. (2013). We show that even with this unsupervised
clustering analysis that the trends are largely consistent with our previous findings using
the visuomotor index and support the broad distinction into decreased, increased and
perimovement neurons (Extended Data Fig. 9).
- 7. Reduced the emphasis on the DDM given the previous literature on using the DDM for
static stimuli and also the recent unresolved debate on whether an urgency gating model
(UGM) or a DDM better fits discrimination behavior (Hawkins et al., 2015b; Hawkins et
al., 2015a). We also included the fits of a UGM and DDM to the behavioral data for
comparison and completeness (Extended Data Fig. 2).
- 8. Provided evidence that almost all of the main effects replicate in both monkeys
(Extended Data Figs. 1, 2, 3, 4, 11, and 12). The only analysis that failed to reach
significance in monkey O was the delay in discrimination time for the deeper compared
to the superficial layers estimated using the classifiers. The list below shows the analyses
that we include for each monkey separately.
- i. Extended Data Fig. 1 shows the fit of the DDM to the behavior of the
two monkeys separately. Table 1 shows the goodness of fit statistics for
the 4 candidate DDM models we considered for each monkey separately.
- ii. Extended Data Fig. 2 shows the fit of a DDM and the UGM to the
behavior of the two monkeys separately.
- iii. Both monkeys show a visuomotor continuum for PMd unit responses
(Extended Data. Fig. 3a)
- iv. Main effects of coherence (Extended Data Fig. 3c-e) and RT (Extended
Data Fig. 4a-f) are also present in both monkeys.
- v. At the population level, increased units also signal choice earlier than the
decreased and perimovement units (Extended Data Fig. 4a-c). Similarly

- on a neuron-by-neuron basis, the increased units signal choice earlier
than both the decreased and perimovement units (Extended Data Fig.
4d-f).
- vi. Regression analysis shows main effects of choice, RT, and coherence in
both monkeys (Extended Data Fig. 10c, d and Extended Data Fig. 10e,
f).
- vii. Both monkeys show a decrease in the visuomotor index as a function of
cortical depth (Extended Data Fig. 11c).
- viii. Superficial layers in both monkeys signal choice earlier than the deeper
layers (Extended Data Fig. 11d). The laminar trends are stronger in
monkey T than in monkey O.
- ix. The only analysis which was not significant in monkey O was the delay in
discrimination time for the deeper layers vs. the superficial layers as
estimated by the classifier.

9. Added nonparametric statistical analyses (comparison of medians, and permutation tests)
to address concerns about normality and correlations between units. Results did not
change when using these nonparametric methods.

10. Included details on the number of single and multiunits reported in the manuscript. Also
included a caveat in the discussion section on how multiunits could modestly change
some of the analyses and conclusions presented in the manuscript.

Having provided this preamble, we now address each comment in turn by pointing out the analyses
or figures or replicating text written in the manuscript in response to the question.

**Specific Reviewers' comments:**

**1. Reviewer #1 (Remarks to the Author):**

The authors trained 2 monkeys to perform a visual discrimination reach task. Despite using a static
stimuli, the monkeys behavior was well fit by a drift-diffusion model (DDM). Recordings from PMd
revealed diverse neuronal populations, with some being more highly correlated with reaction time
and others less correlated. The authors claim that the units that were positively correlated with
reaction time and also choice and stimulus difficulty. But the units that were negatively correlated
with RT did not show "decision-variable" like characteristics. Using a laminar probe the authors
claim that superficial layer contain more of the "increasing" neurons (more correlated with RT.)

##### General Comments

**1.1. The laminar organization of cortical function/computation is of great interest to**
**neuroscientists. This paper is one of the first to describe laminar distinctions in a**
**perceptually difficult reach task, which (in theory) permits the separation of perceptual,**
**decision, and action related signals. There are a few problems that prevent me from**
**recommending the paper for publication in Nature Comm. in its current form.**

We thank the reviewer for the positive evaluation of the novelty of the results presented here. We
have attempted to perform appropriate analyses and rewritten parts of the manuscript to address the
issues raised by you. Please find below our point-by-point response to your comments.

**1.2. The behavior of the two monkeys is quite different, but all the analyses lump the**
**neurons together, calling into question the generality of the results. The authors need to**
**show that the main results and claims hold up independently in both animals.**

We thank the reviewer for this comment. We carefully described in the original manuscript that
there are differences between the monkeys' behavior primarily at the level of the RT. Without the
RT curve and the DDM, and in the revised manuscript the UGM, the thresholds, and psychometric
curves appear similar and in a fixed discrimination task the behavior of the monkeys might be
considered near equivalent.

Nevertheless, we concur with the reviewer that the differences justify the need to show that the
neurophysiological effects are largely similar in both animals and we have done so where appropriate
by plotting data separately for each animal (Extended Data Figs. 1, 2, 3, 4, 11, and 12).

a. Extended Data Fig. 1 shows the fit of the DDM to the behavior of the two monkeys
separately. Table 1 shows the goodness of fit statistics for the 4 candidate DDM
models we considered for each monkey separately.

b. Extended Data Fig. 2 shows the fit of a DDM and the UGM to the behavior of the
two monkeys separately.

c. Both monkeys show a visuomotor continuum for PMd unit responses (Extended
Data. Fig. 3a)

133 d. Main effects of coherence (Extended Data Fig. 3c-e) and RT (Extended Data Fig.
4a-f) are also present in both monkeys.

e. At the population level, increased units also signal choice earlier than the decreased
and perimovement units (Extended Data Fig. 4a-c). Similarly on a neuron-by-neuron
basis, the increased units signal choice earlier than both the decreased and
perimovement units (Extended Data Fig. 4d-f).

f. Regression analysis shows main effects of choice, RT, and coherence in both
monkeys (Extended Data Fig. 10c, d and Extended Data Fig. 10e, f).

- 141 g. Both monkeys show a decrease in the visuomotor index as a function of cortical
depth (Extended Data Fig. 11c).
- 143 h. Superficial layers in both monkeys signal choice earlier than the deeper layers
(Extended Data Fig. 11d). The laminar trends are stronger in monkey T than in
monkey O.
- i. The only analysis which was not significant in monkey O was the delay in
discrimination time for the deeper layers vs. the superficial layers as estimated by the
classifier.

**1.3. The logic of the paper is convoluted. I feel that it could be substantially improved**
**with new analyses and writing. I detail the issues below, but the main problem is with the**
**overuse of the "index" (correlation between pre-movement firing rate and reaction time).**

We apologize to the reviewer for the lack of clarity. As suggested by the reviewer, we have now
added several new analyses to extend the main conclusions of this paper that relied on the index. We
have also attempted to clarify and streamline the text throughout.

**1.4. Of the 3 "concrete goals" the authors describe in the discussion, only the third one**
**(laminar distribution of decision encoding) merits enough interest for publication in Nature**
**Comm. However, the authors did not perform any analyses of the simultaneously recorded**
**units that would provide stronger evidence that the decision-related activity in superficial**
**(and also the deepest) layers of cortex drive the more movement related neurons.**

We appreciate that the reviewer concurs with us on the importance of the laminar differences in
PMd during a decision-making task. Regarding the other two points, we respectfully feel that the
systematic characterization of the relationship between different types of response patterns in PMd
and the decision-formation process is quite important and was largely lacking in the literature. For
instance, as reviewer 2 points out, the decreased neurons are a novel finding for PMd and have only
been previously described in the FEF and the superior colliculus.

Similarly, Coallier et al. (2015), the study with the most similar design as ours, documented decision-
related responses in ~60 PMd neurons (many hundreds of neurons fewer than in the present
manuscript) but did not describe the response diversity that we attempt to show. They also did not
show that different units have different choice selectivity profiles and they provided *one pooled estimate*
of the time of choice selectivity. These types of analyses are a major component of the manuscript,
and our conclusions do not rest entirely on the laminar differences discussed here. The laminar
differences are the final analyses we perform to investigate the complexity observed in the FRs of
PMd neurons during the decision-making task.

Another important feature of this revised manuscript is that we also have now modeled the behavior
of the monkeys using the DDM and an urgency-gating model. We show how both models explain
some features of the data, but neither model is perfect (Extended Data Fig. 2). The DDM

overestimates the highest RT quantiles (70 and 90th percentile). The UGM predicts a shape for the
quantile probability plots that is inconsistent with the data.

We have also added a decoding analysis to show that information about choice emerges earlier in the
superficial compared to the deeper layers of PMd (Fig. 4e, Extended Data Fig. 12b). Choice
information emerges ~16 ms earlier in the superficial compared to the deeper layers of the dorsal
premotor cortex.

##### Major Issues

**1.5. The index used for quantifying a neuron's position along a visuo-motor continuum is**
**unconvincing. It turns out that for the "decreased" population, choice signals emerge later,**
**but it did not have to turn out that way. Thus the index is not itself the property of interest.**
**Rather than report the laminar distribution of the "index" why not directly report the**
**laminar distribution of a perceptual, decision and motor signals, since it seems that that is**
**what the authors are more interested in reporting anyway. The last sentence of the results:**
**"At the population level, choice selectivity appeared earlier in the superficial compared to**
**the deep units."**

**I would recommend to divide the neurons by depth and then use a decoding analysis to**
**reveal the perceptual/decision/motor information content as a function of time at different**
**depths. The authors rely heavily on firing rate differences (using regression / ttests), but**
**something more sophisticated (see for example papers from Pagan and Rust on decoding**
**information in IT) would improve the manuscript.**

We thank the reviewer for pointing this out, we appreciate it, and we agree. We have now shown
that on a single neuron basis the timing of signals emerges earlier in the superficial compared to the
deeper layers of PMd (Fig. 4d, Extended Data Fig. 11d for each monkey). There is a positive
correlation between the timing of significant choice selectivity and cortical depth in both monkeys
and in the pooled data (Fig. 4d).

We have also added a decoding analysis to show that information about choice emerges earlier in the
superficial compared to the deeper layers of PMd (Fig. 4e, Extended Data Fig. 12b). Choice
information emerges ~16 ms earlier in the superficial compared to the deeper layers of the dorsal
premotor cortex.

As regards the perceived unconvincing nature of the index, we have now incorporated an analysis
where we show that there is a near-perfect relationship between our correlation-derived visuomotor
index and the first principal component of an unbiased PCA analysis. The first PC explains ~ 50%
of the variance in the neural data set, substantially bolstering confidence in our correlation-derived
index (Extended Data Fig. 8e).

**1.6. A major reference for the "visuo-motor continuum" type analysis is DiCarlos and**
**Maunsell 2005 "Using Neuronal Latency to Determine Sensory–Motor Processing Pathways**

**in Reaction Time Tasks." That paper provides a more robust technique for doing neural-**
**behavior latency analysis (rather than just comparing spike-rates with RT). Another**
**technique for neural-behavior latency analysis is described in Erlich, Bialek & Brody (2011)**
**which is applicable to neurons with more complex dynamics.**

We thank the reviewer for drawing our attention to both of these analysis techniques. We were
obtaining reasonable results with simple regressions between firing rates and RT and thus did not
pursue more sophisticated techniques for estimating this relationship. We have now added the
analysis performed in DiCarlo and Maunsell as an extended data figure (Extended Data Fig. 5).

We have now also used the technique from Erlich, Bialek and Brody (2011) in the manuscript (with
minor modifications) and reassuringly found very similar results. Neurons with positive visuomotor
index values are the most correlated with the metric provided in Erlich et al. (2011) (Extended Data
Fig. 6). Because the technique from Erlich et al. (2011) does not distinguish between increases and
decreases in FR, we have now made a minor modification to the correlation coefficient and show
that this more sophisticated analysis again suggests a very organized relationship between RT and
responses of PMd neurons. Both the method from Erlich et al. (2011) and DiCarlo and Maunsell
(2005) largely lead to similar conclusions and their metrics are well correlated with the simple index
we propose here.

We also pursued yet another method for characterizing a visuomotor continuum originally
developed in (Sato and Schall, 2003) and recently used in (Song and McPeck, 2010). This data is
shown in Extended Data Fig 7. The conclusions from this alternative analysis are again consistent
with the results using the simple visuomotor index.

**1.7. Given that the behavior of the two monkeys is quite different, it is important to show**
**that your main results (that "increased" neurons have earlier decision information and are**
**more superficial) replicated independently in both animals. Figure 3e, 4b, Extended Data 3.**

Please see the list a-i provided in answer to question 1.2 and also in the preamble to the responses
(point 8).

**1.8. **Use of parametric statistics.** I don't think this would change the main results of**
**the paper, but I would encourage the authors to use non-parametric statistic (like**
**permutation tests) rather than rely on t-tests which have assumptions about the underlying**
**distribution of the data. Especially since the authors do not check (or at least did not report)**
**whether those assumptions were met. For example, simultaneously recorded neurons do not**
**satisfy the independence assumption. Neurons generally have autocorrelations which**
**violate the assumption of independent samples for regression. Firing rates may or may not**
**satisfy the assumption that the data came from a normal distribution.**

Thank you for this suggestion. Where applicable we have now included nonparametric statistics that
measure whether the medians are significantly different from one another to assuage the concern
that the means of these distributions may not be meaningful because of the non-Gaussian nature of
these distributions. As the reviewer has surmised, adopting tests that compare between the medians
did not change the main results of the manuscript. We have also included nonparametric statistics
using permutation tests where necessary. Again using these methods did not change the results
presented in the manuscript.

##### Some detailed comments

**1.9. Line 15 - "... are not well understood."**

**: This phrasing has unfortunately become a universal technique for general motivation of**
**any paper. I wish the authors could be a bit more specific in the abstract as to the**
**hypothesis that they are testing. Your paper is about the laminar organization. So something**
**like "we have previously described the response properties of neurons in PMd using**
**techniques that did not allow us to identify the laminar location... "**

Many studies in the oculomotor system have delineated the role of different brain regions in
perceptual decision-making. We feel that the mechanisms are not well understood for the
somatomotor system. That's why we included this opening sentence. However, we fully agree that
these types of sentences have become standard opening or motivating statement for many papers
and we apologize if the opening sentence of this manuscript is trite. We have therefore edited the
first two sentences to better set up the rationale for our study.

**Lesion and physiological experiments in monkeys and examination of stroke patients**
**suggest a role for the dorsal premotor cortex (PMd) in somatomotor decisions. However,**
**we currently lack a description of the temporal patterns of decision-related firing rates (FRs)**
**in PMd. We also do not understand the laminar organization of these decision-related**
**responses.**

**1.10. Lines 19 - 25**

**: On line 19 "firing rates of PMd neurons are organized" but at line 24 "units at the**
**center." Is it a continuum of units or of firing rates? If it a continuum of units then you**
**should write (on 19): "Units in PMd can be organized along a continuum by their responses**
**in our reach task." (or something to that effect.)**

We thank the reviewer for this suggestion and have rewritten the sentence in the abstract

**"We show that PMd neurons can be organized along a bidirectional visuomotor continuum**
**according to their responses during the reach decision task."**

**1.11. Line 19 "Consistent with a model, which."**

**: Remove comma after model**

We thank the reviewer for this suggestion. We have rewritten the sentence.

**Together, the results are consistent with a laminar model in which decision-related signals**
**are received and processed in the upper layers of PMd, subsequently informing action**
**selection and execution processes centered in the lower layers.**

**1.12. Figure 2: "Check."**

**: I would encourage the authors to just write "cue" rather than "check" in Figure 2.**
**"Check" is not a standard short form for "checkerboard", but 'Cue' is pretty easy to**
**interpret (especially if you use that term in fig 1 and the rest of the text... e.g. the**
**"checkerboard cue".)**

We thank the reviewer for the suggestion and have amended the text as well as the methods to be
clearer. We have replaced the word check in the figures with the word “cue” and refer to it as
“checkerboard cue” in the text.

**1.13. Figure 3e: Regression**

**: The regression should be done on the actual data, not the binned and averaged data. Why**
**did the authors do that? This figure needs to be made for each animal separately.**

We chose to perform the regression on latencies estimated from the average population data
because we were concerned that single neuron estimates of this regression might be noisy. However,
the analysis was just as robust when we performed the analysis using the individual neurons and
found very similar results (Extended Data Fig. 4d-f, Extended Data Fig. 7, Fig. 4d, Extended Data
Fig. 11d). We thank the reviewer for the suggestion and agree that this is statistically a more robust
measure of the latency differences in PMd. We now include these neuron-by-neuron estimates as
figures separately for each monkey (Extended Data Fig. 4e-f) and also pooled across both monkeys
(Extended Data Fig. 4d). We also include a correlation between the discrimination time measured on
a single neuron basis over all RTs and our simple visuomotor index (Extended Data Fig. 4d-f).

As advised by the reviewer, we also included analyses from DiCarlo and Maunsell (2005) as
well as the analysis from Erlich et al. (2011). As suggested by reviewer 3, we also included an analysis
based on the slopes as devised by Sato and Schall (2003) and Song and McPeck (2010). Conclusions
from all of these different analyses were consistent with one another.

**1.14. Line 88-90: "..the DDM framework applies even when sensory evidence is 90**
**available all at once"**

**: This is not news. You cite two (relatively recent) papers. But much of the early**
**psychophysics work related to the DDM is based on static visual discrimination. What is**
**unique about your design?**

We thank the reviewer for this question. We agree that the DDM has been applied in settings even
when the sensory evidence is available all at once. Thus our design is not unique. We perhaps
inadvertently overreached in the original paper. However, recent studies, especially from Cisek,

Thura, and collaborators, have now challenged this claim (Cisek et al., 2009; Thura et al., 2012).
Thura et al. (2012) claimed that the UGM and DDM make identical behavioral predictions for these
types of stimuli. However, a recent set of studies showed that the UGM and the DDM only make
identical predictions for mean RTs. The two models make different predictions about the
distribution of RTs for correct and error trials (Hawkins et al., 2015b; Hawkins et al., 2015a; Boehm
et al., 2016; Thura and Cisek, 2016). We have now attempted to use both of these models to identify
if one of them describes the data better than the other. The DDM predicts the shape of the quantile
probability plots better but overestimates the RTs for the highest quantiles (Extended Data Fig. 2a,
left panel). In contrast, the UGM predicts a different shape but predicts faster RTs for the highest
quantiles (Extended Data Fig. 2b, right panel). Depending on the relative balance, either the DDM
or the UGM describes the data better.

Nevertheless, we agree with both Reviewer 1 and Reviewer 3 that the DDM vs. UGM is not the
main focus of this manuscript. The more novel results in the manuscript are the temporal
heterogeneity of decision-related responses and the demonstration of a laminar structure in PMd.
We have rewritten this introductory paragraph and deemphasized the DDM in the results and
provided an explicit discussion on the DDM vs. the UGM.

We have adopted the strategy of keeping the modeling results but minimizing their emphasis in the
manuscript that the DDM is the only model that can fit this data. All of these decision-making
models predict very similar FR profiles for trial-averaged data with the stimuli used by us.

**1.15. Figure 4B**
**: Why use a line to fit this when it clearly looks non-monotonic? A quadratic fit (or two lines)**
**seems more appropriate.**

We thank the reviewer for this suggestion to use quadratic and higher order fits to describe the
dependence of visuomotor index on cortical depth (Fig. 4c, Extended Data Fig. 11c). We have made
this appropriate change in the methods and also amended the manuscript to incorporate these new
statistics. The pooled data were better explained by using a cubic function than a quadratic or linear
function.

**2. Reviewer #2 (Remarks to the Author):**

Chandrasekaran et al. present an analysis of multi-unit recordings from laminar probes in
premotor cortex during a perceptual decision making task. They found that some units whose
activity positively correlated with reaction times, activity bears signatures of a decision variable as
predicted by a drift-diffusion-type model. Depth measurements from the laminar probes
revealed that decision-related units were preferentially distributed in the superficial layers,
whereas units in the deeper layers reflected more motor-preparatory responses.

I find this to be an interesting and compelling study that provides a novel advance to our
understanding of the cortical circuit dynamics during decision making. The analyses are
thorough and well-designed, and the figures and text are carefully and clearly presented. The
laminar dependence of decision vs. motor signals in primate premotor cortex is of particular
novelty and importance, as they suggest a circuit-level transformation in the flow of information
across layers during decision making. I have only two comments/concerns, which should be
readily addressable, described below.

We thank the reviewer for the kind words and positive evaluation of the novelty of the results
presented here.

Comments:

**2.1. Relationship between visuomotor index and choice selectivity: What is the**
**relationship across units between the visuomotor index and choice selectivity? Fig. 3C**
**speaks to this question at the group level, averaging within decreased, perimovement, and**
**increased divisions, but not at how these may be correlated or clustered at the unit level.**
**This could be shown with a scatter plot showing, for each unit, the visuomotor index and**
**some measure of choice selectivity (e.g. for each unit, the immediate pre-movement**
**difference in firing rates or slopes between choices, averaged across coherences, and maybe**
**normalized by baseline/overall firing rate). This analysis could address the important**
**question of whether this sample can be described a unimodal distribution showing a range**
**of choice selectivity, or whether there is clustering with some units showing very little or**
**very high choice selectivity. For instance, the example units at channel 11 in Fig. 4A show**
**very little choice**
**selectivity at the time before movement.**

We also agree that this is an interesting question. We have now attempted to answer this question
using the following analysis.

We have added an extended data figure (Extended Data Fig. 4h) that plots the correlation between
the index and the choice probability (measured using an ROC analysis on the smoothed firing rates)
at each time point. There is a reliable positive correlation between the visuomotor index we defined
and choice probability. The correlation plot shows that as the visuomotor index becomes more
positive (that is towards a more increased neuron) the choice selectivity also increases. We also show
three scatter plots for each of the three different time points shown in the left panel. The data are
more consistent with a continuum and less consistent with clusters of neurons with stronger vs.
weaker choice selectivity.

**2.2. In that sense, they resemble the “fixation” neurons recorded in the frontal eye field**
**(by Jeff Schall and others), which show omni-directional decreases preceding movement. So**
**is there a subgroup of units that show a strong choice signal, beyond the expected**
**continuum of a unimodal distribution?**

We thank the reviewer for alerting us to the fixation neurons reported by Jeff Schall and others. We
have added this as a sub section in the discussion portion of the manuscript:

The decreased neurons had tonic levels of FRs that began to reduce after the checkerboard cue
onset and showed minimal choice selectivity. FRs of these neurons were similar for left and for right
reaches and only diverged as a function of choice just before movement onset. These FR
characteristics share some characteristics with fixation neurons documented in the frontal eye
fields (Izawa et al., 2009; Schall, 2009) and the superior colliculus. Fixation neurons, like the
decreased neurons described here, reduce their activity before the initiation of a saccade (Hanes et al.,
1998). They also form a small proportion of neurons described in FEF an observation consistent
with the result that decreased neurons were the least common of the broad categories (~15% of the
neural population). Finally, fixation neurons are thought to be colocalized with movement related
neurons in layer 5 of the FEF — a result consistent with our report here of finding these decreased
and perimovement neurons largely in the deeper electrodes of the U-probes and thus the deeper
layers of PMd (Schall, 2009).

**2.3. Potential limitations of multi-unit analysis: A key limitation of the current work is**
**that the analyses are done on only multi-unit data, without any validation that the main**
**conclusions would hold if analyses were applied to only isolated single-neuron recordings.**
**Indeed, with this limitation, it is not even clear how one should interpret the selectivity and**
**visuomotor indices of these units when multiple neurons are combined to produce an**
**analyzed unit. The Discussion section should note this limitation and describe what aspects**
**of the conclusions may be sensitive to the multi-unit vs. single-neuron distinction or which**
**specific aspects would benefit from analysis of single neurons.**

We apologize for the lack of clarity in the results and methods section about the number of single
neurons and multiunits reported in our population. We have now added this text to the methods
section. ~80% of the units reported here are single neurons. The remainder are multiunits (~20%).
We have also added a caveat in the discussion section about what additional insights may be gained
from even better isolation of single neurons.

**Methods Section**

We report here the activity of 996 units recorded in PMd of two monkeys while they performed the
task (546 units in T, 450 units in O; single neurons and multi-units which responded during any
epoch in the task). The majority of the *units* in the database were well-isolated single neurons, and as
431 per convention, we term them single neurons. Some of these single neurons were collected using
high impedance (i.e., small electrode contact area, $> 6 \text{ M}\Omega$) sharp FHC electrodes. When using these
electrodes, every attempt was made to isolate and track single neurons and to stably record from
them.

We also had excellent success recording from isolated single neurons from the U-probes.
The U-probes are low impedance electrodes ($\sim 100 \text{ k}\Omega$) with a small contact area and were thus
excellent for isolation of single units. We used a conservative threshold to maximize the number of
clearly defined waveforms and minimize contamination from spurious non-neuronal events. Online,
we used the hoops provided by the software client for our Cerebus system (BlackRock
MicroSystems) to delineate single neurons after the electrodes had been placed in the cortex for at
least half an hour to 45 minutes. Every time a spike was detected by the threshold method, a 1.6 ms
snippet was stored and used for subsequent evaluation of the clusters as well as adjustments needed
for spike sorting. Recordings from some electrodes in the U-probes consisted of mixtures of 2 or
more neurons well separated from the noise and from one another. In a majority of these cases, the
waveforms were clearly separated, and these were labeled as single units. We verified these
separations visually by viewing the waveforms in a principal component space using custom code
written in MATLAB and adjusted the clusters that were isolated online using the MatClust
MATLAB based clustering tool that allows the drawing of polygons to label clusters (O'Connor et
al., 2013) or the Plexon Offline Sorter. Finally, we had two other types of recordings that we labeled
as multiunits. The first category consisted of the multi neuron mixtures that were not separable
using a principal components method. The second rarer class consisted of recordings with
waveforms only weakly separated from the noise. Both of these types of units were classed as
multiunits. We included all of these different units in our analysis here.

To reduce the subjectivity in our assessments we used the number of ISI violations (after
clustering and sorting) to distinguish between single and multiunits. We labeled a unit as a single
neuron if the percentage of ISI violations (refractory period of $\leq 2\text{ms}$) was less than 1.5% and a
unit with ISI violations exceeding 1.5% as a multiunit. Using this criterion, 801 of the 996 PMd units
reported here are identified as single neurons (384 from O, 417 from T, mean ISI violation = 0.43%,
~ 0.13 additional spikes per trial). Similarly, 195 out of 996 are classified as multiunits (129 from T,
66 from O, mean ISI violation = 3.36%, ~ 1.4 additional spikes per trial).

**Discussion Section**

The majority of the electrophysiological data reported in this manuscript were single neurons
($\sim 80\%$) recorded in PMd during the decision-making task. However, we also included a substantial
fraction of multiunits in the data ($\sim 20\%$). The multiunits certainly provided us with additional power
for the analyses presented here, but it may have also lead to spurious misclassification of units in the
continuum. First, in the worst case, there is the possibility of combining the FRs of an increased
neuron with a decreased neuron and a perfect cancelling in the decision-formation period could
result in a spurious perimovement unit. Fortunately, as our laminar recordings show, the increased,

decreased and perimovement like FRs appear to be roughly segregated as a function of cortical
depth, so this type of spurious mixing will be minimized due to this topographic organization.
Second, because a multiunit contains additional spikes, there is a slightly greater chance that a
decreased neuron will be misclassified as a perimovement or an increased unit and this will increase
the preponderance of increased units in our database. Third, there is also the possibility that some
finer grained temporal patterns are smeared because of combining multiple neurons into a unit.
Finally, inclusion of the multiunits could have resulted in smoother visuomotor continuum than
what is actually present in PMd. Future studies that use a laminar electrode with a tetrode
configuration that will improve isolation or the next generation of silicon electrodes that provide
high-density recordings in combination with automated sorting methods may further illuminate the
microcircuit in PMd and other structures involved in perceptual decision-making (Berenyi et al.,
2014; Rossant et al., 2016). These techniques may allow us to derive so-called “electroanatomical”
maps of brain regions (Berenyi et al., 2014).

**2.4. DDM parameters: It would be of definite interest to some readers to have the DDM**
**fit parameter values provided in a supplementary table.**

We have added the DDM goodness of fit metrics (Table 1) as well as the parameters of the fit
obtained from the fast-dm implementation (Tables 2, 3). In the revised manuscript, we have also fit
the urgency-gating model to the data and show that the UGM predicts a shape for the quantile
probability plot that is inconsistent with the behavior of the monkeys but can successfully capture
the highest quantiles of the RT distributions. Neither model is sufficient to completely describe the
behavioral data.

**3. Reviewer #3**

This manuscript describes a study examining neural activity in dorsal premotor cortex (PMd) during
 a perceptual discrimination task. Monkeys were trained to determine which of two colors was
 predominant in a mixed color cue and indicate their choice by reaching to a target of the
 corresponding color. Recordings were made in caudal PMd using multisite linear probes.
 There were three aims: 1) To characterize the variety of firing patterns in PMd during this task,
 relating them to putative decision variables; 2) to test whether these firing patterns, and monkey
 behavior, conformed to the predictions of the drift diffusion model (DDM); 3) to examine whether
 there are specific trends in the distribution of different firing patterns in different cortical lamina.
 Three corresponding main results are reported: 1) PMd cells can be characterized as lying along a
 continuum that is indexed by their trial-to-trial relationship to reaction time. At one end there are
 “increasing cells” whose movement-aligned activity is larger when reaction times are longer, and
 which are most clearly related to the difficulty of the decision. At the other end are “decreasing
 cells” whose activity is larger for shorter reaction times, and which do not reflect the choice until
 just before movement onset. In-between there are “perimovement cells” that do not
 correlate with RT and simply indicate movement direction about 150ms prior to onset. 2) The
 activity of increasing cells, and the monkeys’ behavior, is consistent with the DDM. 3) There is a
 clear trend in the laminar distributions of the cell types, with increasing cells more common in
 superficial layers, while decreasing and perimovement cells are more common at deeper layers.
 I will comment on each of these aims and findings in turn, but I will begin with aims 1 and 3, as they
 are related, before turning to aim 2.

 **3.1. Many previous studies have reported similar kinds of trends in PMd neural firing**
 **patterns, especially by Coallier et al. 2015 and Song & McPeck 2010 (both cited), but also**
 **earlier studies by Evarts, Tanji, Wise, Kalaska, Caminiti and others. These authors have**
 **described similar types of “stimulus-related” or “motor-related” cells, using a variety of**
 **indices and methods for quantifying response patterns. Similar trends have also often been**
 **reported in other regions, notably including FEF (e.g. Thompson et al. 1996 cited; as well as**
 **many other studies by Schall and colleagues, see in particular Sato & Schall 2003). The**
 **authors acknowledge all of this, but then it’s unclear why they state that a “detailed**
 **description of the temporal patterns” is lacking (p. 2), and unclear how their analyses**
 **provide a description that is more detailed than what was already shown previously. Having**
 **said that, I do recognize that the present paper quantifies the specific relationship with**
 **decision-related variables, which has not been done before.**

We apologize for our lapse in including papers from classical motor studies. We have now included
 the following appropriate citations to Tanji, Caminiti, Evarts, Kalaska, Wise and others in the
 introduction.

- 20. Crammond, D.J. & Kalaska, J.F. Modulation of preparatory neuronal activity in
dorsal premotor cortex due to stimulus-response compatibility. *J Neurophysiol* **71**,
1281-1284 (1994).
- 21. Crammond, D.J. & Kalaska, J.F. Prior information in motor and premotor cortex:
activity during the delay period and effect on pre-movement activity. *J Neurophysiol*
**84**, 986-1005 (2000).
- 22. Kurata, K. & Wise, S.P. Premotor cortex of rhesus monkeys: set-related activity
during two conditional motor tasks. *Exp Brain Res* **69**, 327-343 (1988).
- 23. di Pellegrino, G. & Wise, S.P. Visuospatial versus visuomotor activity in the
premotor and prefrontal cortex of a primate. *J Neurosci* **13**, 1227-1243 (1993).
- 24. Johnson, P.B., Ferraina, S., Bianchi, L. & Caminiti, R. Cortical networks for visual
reaching: physiological and anatomical organization of frontal and parietal lobe arm
regions. *Cereb Cortex* **6**, 102-119 (1996).
- 25. Ferraina, S., *et al.* Visual control of hand-reaching movement: activity in parietal area
7m. *Eur J Neurosci* **9**, 1090-1095 (1997).
- 26. Weinrich, M. & Wise, S.P. The premotor cortex of the monkey. *J Neurosci* **2**, 1329-
1345 (1982).
- 27. Tanji, J. & Evarts, E.V. Anticipatory activity of motor cortex neurons in relation to
direction of an intended movement. *J Neurophysiol* **39**, 1062-1068 (1976).
- 28. Mushiake, H., Inase, M. & Tanji, J. Neuronal activity in the primate premotor,
supplementary, and precentral motor cortex during visually guided and internally
determined sequential movements. *J Neurophysiol* **66**, 705-718 (1991).

As the reviewer has recognized by our citations of some of the most relevant papers, we were well
aware of the many different studies that had previously demonstrated stimulus related and
movement related units in PMd and the frontal eye fields. Indeed, we built upon these ideas that a
visuomotor continuum can describe the FR patterns and used it as the simplest way to tease apart
the different types of units involved in the decision-formation process. **We agree that we were**
**remiss in NOT citing these key references.** We have now included them in the revised
manuscript.

However as the reviewer has recognized, first there is a gap in our understanding of the types of
responses in PMd during a difficult reach decision-making task and whether some of them are better
described by a decision-variable. Second, to our knowledge decreased neurons have not been
reported in PMd. As reviewer 2 points out, the decreased neurons are a novel finding for PMd and
have only been previously described in the FEF and the superior colliculus.

To illustrate, Coallier, Michelet, and Kalaska 2015, the most recent study with quite a similar
design as ours, documented decision-related responses in ~60 PMd neurons (hundreds of neurons
less than reported in the present manuscript) but did not describe the response diversity that we
attempt to show. They also did not show that different units had different choice selectivity profiles
and provided one pooled estimate of the time of choice selectivity (For instance, Table 3A of
Coallier et al. (2015)). We attempted to provide many different analyses that help us dissociate
between the different firing rate patterns.

Similarly, Song and McPeck (2010) described neurons in the visual and motor ends of the
continuum but did not describe any of the decreased neurons. Their task also involved a visual
search. Not only did we replicate their findings in this study, we also show that a similar visuomotor
continuum is present in PMd during a difficult decision-making task with an arm reach as the
behavioral report.

**3.2. However, the index that is used is perhaps not ideal to tease that apart from other**
**stimulus-locking or burst vs. tonic properties. The index is straightforward and reasonable,**
**but it could potentially confound important phenomena. In particular, the FR is averaged**
**across a lengthy period of time during which cells exhibit temporal changes, some**
**increasing over the course of the trial and some decreasing. The classification of a cell as**
**having a positive index could be caused by several different scenarios: 1) it could reach a**
**similar level of activity but for a longer time prior to movement onset in long RT trials**
**(perhaps simply because it is “stimulus locked” and “tonic”; 2) it could exhibit a burst that**
**always lasts the same amount of time but is stronger in long RT trials; 3) it could build up**
**as a function of elapsed time independently of difficulty. All of the examples shown**
**illustrate scenario 1, but were other scenarios observed? If so, how often? A negative index**
**could likewise be produced by different scenarios, and it is very conceivable that stronger**
**bursts are associated with shorter RTs (and/or higher velocities). The index might therefore**
**inappropriately partition cells that are in fact closely functionally related, or group together**
**cells that are very different. It seems to me that one needs to be more careful in**
**characterizing these trends (aim 1), possibly through the use of multiple indices that**
**measure different aspects of cell responses (e.g. duration of burst, amplitude of burst, time-**
**locking with stimulus or movement onset, etc.). The analyses used by Sato & Schall and**
**Song & McPeck, examining the relationship between RT and neural discrimination time,**
**could be particularly insightful here, but only as one among several ways of quantifying**
**response patterns.**

We thank the reviewer for raising this important concern. We have now attempted to use several
other analyses with varying degrees of supervision to describe these response patterns. We are
pleased to note that all of these alternative methods still provide very similar results to our main
conclusions using the straightforward and reasonable visuomotor index that we derived here.

- • We have now implemented the analyses originally developed by Sato and Schall (2003) and
further refined by Song and McPeck (2010) in this revised manuscript. This data is shown in
Extended Data Fig 7. The conclusions from this alternative analysis are also consistent with
the results using the simple visuomotor index.
- • We have included the correlation analysis from Erlich et al. (2011) that attempts to maximize
the correlation between the mean PSTH and individual PSTHs to understand the
relationship between RT and PSTHs of the neurons (Extended Data Fig. 6). This
sophisticated correlation is very well related to the visuomotor index we proposed here.

- • We have included an analysis from DiCarlo and Maunsell (2005) to identify the position of
these neurons along a visuomotor continuum (Extended Data Fig. 5).
- • It is entirely plausible that there are many different patterns that we have completely missed.
However, if this is the case, then the number of principal components should increase with
the number of distinct patterns. The variance in the FRs is well explained by the top 5
components (Extended Data Fig. 8).
- • As regards the unconvincing nature of the index, we have now incorporated an analysis
where we show that there is an almost near perfect correlation between the correlation-
derived visuomotor index and the first principal component that explains ~ 50% of the
variance (Extended Data Fig. 8).
- • Another analysis we performed was based on an analysis by Meister et al. (2013) that used
clustering to separate out neural signals in LIP during perceptual decisions. We find that
most of our data is well described by ~5 clusters. When we increase the number of clusters
we further fractionate the neural populations, but the gain in explanatory power is modest
(Extended Data Fig. 9).

**3.3. Assuming that the trends in response patterns are well-characterized, mapping them**
**to different cortical layers (aim 3) is novel and very interesting. In fact, I consider this to be**
**the most significant advance presented in this paper. The result in Figure 4b is compelling,**
**and it would be very good to show that for individual recording sessions.**

As requested by the reviewer, we have now added single-session examples of this index in Fig. 4b
and Extended Data Fig. 10b.

**3.4. Perhaps even as a 3-D map that examines how robust these trends are along a rostro-**
**caudal or medial-lateral axes. Does the prevalence of superficial increasing cells decrease as**
**you move caudally?**

Successfully making U-probe recordings meant that we needed to constantly monitor the Dura to
see if there was any dimpling or resistance. The use of a grid system would drastically reduce the
visibility in the chamber and compromise the quality/safety of the recordings. As a consequence, we
have very little access to the precise locations of the recordings.

However, we did perform a modest number of U-probe recordings in a more caudal location in our
chamber (putative M1). For these sessions, we did not observe the same trends that we observed in
dorsal premotor cortex (Extended Data Fig. 12a). The visuomotor index was nearly flat as a
function of cortical depth and if anything showed the opposite trend to what was observed in PMd.

**3.5. Do they appear at different depths? It would also be desirable to have more detailed**
**reconstructions so as to be able to judge what specific cortical layers correspond to the**
**different depths. The online methods describe the careful placement of the probe to**
**maximize consistency across sessions, but this cannot do anything about the different**
**angles at which the probes intersect the curving cortical surface. A histological**
**reconstruction would**
**be best, but if the animals are still being used in experiments then perhaps a careful 3D**
**reconstruction based on MRI data would suffice. I emphasize this because the laminar**
**trends described here would be even more valuable if they could be more precisely related to**
**specific cortical layers.**

Unfortunately, we do not have access to histology or MRIs of the recordings. Both animals are
actively participating in other decision-making and motor control experiments in the laboratory (See
SFN abstract by Wang et al. (2016)). The lack of MRIs was the reason why we attempted to
maximize consistency across sessions using physiological markers as much as possible.
Experimentally, this was much more successful and easier in Monkey T than in Monkey O.

**Layers:** We agree that it would be very helpful if we could identify from our physiological
recordings, the exact layer the responses are recorded from. However, we currently have no reliable
way to identify different cortical layers. In the primary visual cortex and areas such as V4, flashes
reliably elicit a visual response that can be used to map out different cortical layers via a current
source density method (Nandy et al., 2016). A corresponding method for the premotor cortex is
currently lacking because visual flashes do not elicit robust responses unless tied to an action. So this
type of analysis is unfortunately beyond the scope of this manuscript and study.

**3.6. Regarding aim 2, the paper is rather underwhelming. The authors contrast their**
**findings with a study (Thura & Cisek 2014), in which it was claimed that PMd activity is not**
**compatible with the DDM but instead quickly tracks sensory information and combines it**
**with a growing urgency signal. Why is that explanation not compatible with the present**
**data? The central assumption of the DDM is that build-up of neural activity is caused by an**
**integrator with a long time constant, and Thura & Cisek argued that their data suggests the**
**PMd time constant is short. Here, the authors argue that their data is compatible with the**
**DDM (i.e. an integrator with a long time constant). But they can't actually test that proposal**
**because it is not possible to distinguish between a long and short time constant with a task,**
**like the present one, in which the stimulus information is static. All of the data shown here,**
**and all of the fits, could be done with a variety of settings of time constants and**
**urgency signals. The authors propose that perhaps the brain adjust time constants in a task-**
**dependent way, using long ones when the stimulus is static. This is plausible, but it has also**
**been claimed that time constants are short even for the classic random-dot motion task**
**(Carland et al. 2016). Regardless, no case can be made either for or against the DDM using**
**data, like that presented here, which does not actually test the DDM.**

We are very receptive to this concern, and we thank the reviewer for pointing out this lack of rigor
in the manuscript. We were remiss in loosely claiming that the DDM provides a better fit to the data
*without* testing the alternative Urgency Gating Model (UGM). As the reviewer correctly identifies,
many candidate models with different assumptions can potentially explain discrimination behavior,
and this is an active, fascinating debate that is currently occurring between Cisek and Colleagues and
Ratcliff and colleagues (Hawkins et al., 2015a; Boehm et al., 2016; Thura and Cisek, 2016).

In particular, the predictions for the *mean RTs* are the same for the DDM and the UGM. However,
as Hawkins et al. point out, the DDM and the UGM make different predictions when examining
quantile plots of RT and probability of correct responses (Hawkins et al., 2015a). In particular, this
study identified that the UGM predicts that the error trials would have longer RTs than correct trials
especially for the easiest conditions (Extended Data Fig. 2a).

We, therefore, examined the quantile probability plots for the two different monkeys (Extended
Data Fig. 2b). Visually, the data were more consistent with a DDM than an UGM with a 100 ms
time constant. We went one step further and used the same toolbox developed by Hawkins et al. to
fit the data (Code available from Guy Hawkins on request, Extended Data Fig. 2c). We tested the
UGM developed in Thura et al. (2012) and an evidence accumulation model (or DDM) developed
by Stone et al. (Stone, 1960). Consistent with the observation from the quantile plots we found that
the DDM fits the data slightly better than the UGM for the fast RTs. In contrast, the UGM fits the
slowest RTs better (Extended Data Figs. 2c, d). We nevertheless agree with the reviewer that neither
the DDM nor the UGM can completely describe the data. We have removed the overemphasis on
the DDM from the introduction and wrote the discussion to be broader and incorporate the two
different viewpoints espoused by the Ratcliff and colleagues approach and also Cisek and others'
UGM approach. We have also amended the various portions of the text to better reflect this
distinction and wrote an explicit discussion, which now expands on this relevant debate and show
how our increased neurons could be explained by both of these mechanisms.

We did not want to remove the whole section on DDMs because Reviewer 2 suggested that these
parameters are of interest. This is also a very relevant and interesting topic for interpreting the
myriad reports of decision-related firing rates in many brain regions which usually are thought to be
described by the DDM. Another reason is that detailed modeling of behavior from decision-making
experiments in different species appears to support different mechanisms (Boehm et al., 2016). Our
hope is that these new analyses and discussions suffice to ameliorate the concern of the reviewer
that we were not precise enough in our results and discussion section about the different models
that can explain the data.

**3.7. Consequently, I would recommend that the authors just remove the DDM from the**
**manuscript, including the discussion as well as the model fits. It merely distracts from the**
**main contribution, which (at least in my mind) is the characterization of differences in**
**activity patterns at different cortical layers. Furthermore, given the ambiguous status of the**
**DDM as an explanation for this kind of experiment, I would recommend not basing any of**
**the analyses on fits to that model.**

We thank you for this comment. We have taken an approach where we have fit both models to the
behavioral data and then shown that neither model provides a complete account of the behavior of
the monkeys. Fortunately, none of the analyses of the neural responses presented here depend on
whether the DDM or the UGM is the right model. As we understand it, when trial averaging, both
models predict very similar FR profiles. We have elected to retain the DDM (and UGM) discussion,
for the reasons described directly above.

**3.8. This specifically pertains to estimates of decision times, which are derived from**
**parametric fits that assume the DDM as the correct model, even though it is not tested. In**
**fact, the non-decision times of 326 and 360ms for the two monkeys, respectively, are quite**
**long. If that is the sum of sensory plus motor delays, then it would be hard to explain how**
**the monkeys could perform a simple reaction time task with an RT that is less than 300ms,**
**which presumably they often do.**

We note that the best DDM fits (measured by the chi square estimate in Table 1) were obtained
when a variability parameter was also included for the non-decision time. This variability parameter
is the range of the uniform distribution centered on the mean non-decision time. This variability
range was ~140 ms for monkey T and ~120 ms for monkey O. So non-decision times could be in
principle as low as ~256 ms for monkey T and ~290 ms for monkey O.

Thus our estimates of decision-times are likely a lower bound. Even if our estimates of non-decision
736 times are higher than expected, it only bolsters confidence that the monkeys are monitoring the
737 stimulus more to make their decisions. It might mean that our “200 ms” decision time for the
738 monkeys is actually a lower limit on what the monkeys are using to form their decisions.

Regarding absolute magnitudes for non-decision times, we do not feel that these numbers are hugely
inconsistent with other reports of non-decision times. Thura and Cisek (2014) estimated non-
decision times using the RTs from a simple delayed reach task that is similar to the simple RT task
which the reviewer has suggested. These numbers for RTs and thus by proxy non-decision times
for the two monkeys reported in Thura and Cisek were as follows (mean \pm SD: 291 \pm 40 ms, 335 \pm 93
744 ms). These numbers were not too far from the mean non-decision times we report here. Similarly,
Coallier, Michelet and Kalaska (Coallier et al., 2015) provided mean RT estimates of 290.7 \pm 54.5 ms
and 274 \pm 51.2 ms for their monkeys in a single-target task. Finally estimates from other monkeys in
our lab performing delayed reach tasks (monkeys K and N reported in (Ames et al., 2014)) suggest
mean RTs of ~310 ms, 245 ms, and 260 ms. Thura and Cisek (2014) in their discussion also suggest
that non-decision times may be 35-50 ms larger in tasks that involve volitional commitment (as in
our RT discrimination task) compared to simple delayed reach tasks. Together, even though our
estimates seem slightly on the higher side, they are roughly in the same regions for monkeys
performing a very simple delayed reach task which in principle should only involve minimal sensory
and motor delays.

Specific comments:

**3.9. Line 168: It is reported that the neural data comes from 546 units in monkey T and**
**450 in monkey O and that this includes both single units and multi-units. Why include the**
**multi-units? One critical prerequisite to addressing aims 1 and 3 is to characterize the**
**diversity of firing patterns of real neurons, and so averaging across neurons is exactly the**
**wrong thing to do. It only raises the concern that many important phenomena were missed.**

We included the multi units because we were circumspect about excluding any recorded data and we
wanted as much power as we could for understanding how cortical depth influences decision-related
activity. We have now provided a more detailed account of the types of units that were included in
the database in the methods section. As suggested by reviewer 2 we have placed a caveat at an
appropriate location in the discussion section. We note that the multiunits were only 20% of the
units recorded in our full dataset and thus unlikely to dominate the results presented here. 80% of
the units are true single neuron recordings as determined by the usual electrophysiology standards in
the field.

The text included in the revised manuscript can be found in the answer to question 3 posed by
reviewer 2 (2.3).

**3.10. Line 176: It is interesting that stimuli with higher coherence were associated with**
**earlier divergence of FRs, but the statistics that test this should be reported (In fact, the**
**strength of this correlation would make a very interesting index for characterizing response**
**patterns, as noted above). Similarly, statistical tests should be reported for all of the other**
**claims made as well.**

We apologize for this lack of clarity in the manuscript. We meant to say faster *rate* of divergence
which is to say that choice information increases faster for easier compared to harder coherences as
in Figs. 3b-c. We report the correlation between the slope of the FR dependence on coherence and
the visuomotor index in this revised manuscript. (positive correlation between the visuomotor index
and the dependence of slopes on color coherence, spearman's $r=0.25$, $p=6.35e-16$)

References

- Ames KC, Ryu SI, Shenoy KV (2014) Neural dynamics of reaching following incorrect or absent
motor preparation. *Neuron* 81:438-451.
- Berenyi A, Somogyvari Z, Nagy AJ, Roux L, Long JD, Fujisawa S, Stark E, Leonardo A, Harris TD,
Buzsaki G (2014) Large-scale, high-density (up to 512 channels) recording of local circuits in
behaving animals. *J Neurophysiol* 111:1132-1149.
- Boehm U, Hawkins GE, Brown S, van Rijn H, Wagenmakers EJ (2016) Of monkeys and men:
Impatience in perceptual decision-making. *Psychon Bull Rev* 23:738-749.
- Cisek P, Puskas GA, El-Murr S (2009) Decisions in changing conditions: the urgency-gating model.
*J Neurosci* 29:11560-11571.

- Coallier E, Michelet T, Kalaska JF (2015) Dorsal premotor cortex: neural correlates of reach target
decisions based on a color-location matching rule and conflicting sensory evidence. *J*
*Neurophysiol* 113:3543-3573.
- DiCarlo JJ, Maunsell JH (2005) Using neuronal latency to determine sensory-motor processing
pathways in reaction time tasks. *J Neurophysiol* 93:2974-2986.
- Erlich JC, Bialek M, Brody CD (2011) A cortical substrate for memory-guided orienting in the rat.
*Neuron* 72:330-343.
- Hanes DP, Patterson WF, 2nd, Schall JD (1998) Role of frontal eye fields in countermanding
saccades: visual, movement, and fixation activity. *J Neurophysiol* 79:817-834.
- Hawkins GE, Wagenmakers EJ, Ratcliff R, Brown SD (2015a) Discriminating evidence
accumulation from urgency signals in speeded decision making. *J Neurophysiol* 114:40-47.
- Hawkins GE, Forstmann BU, Wagenmakers EJ, Ratcliff R, Brown SD (2015b) Revisiting the
evidence for collapsing boundaries and urgency signals in perceptual decision-making. *J*
*Neurosci* 35:2476-2484.
- Izawa Y, Suzuki H, Shinoda Y (2009) Response properties of fixation neurons and their location in
the frontal eye field in the monkey. *J Neurophysiol* 102:2410-2422.
- Meister ML, Hennig JA, Huk AC (2013) Signal multiplexing and single-neuron computations in
lateral intraparietal area during decision-making. *J Neurosci* 33:2254-2267.
- Nandy AS, Nassi JJ, Reynolds JH (2016) Laminar Organization of Attentional Modulation in
Macaque Visual Area V4. *Neuron*.
- O'Connor DH, Hires SA, Guo ZV, Li N, Yu J, Sun QQ, Huber D, Svoboda K (2013) Neural
coding during active somatosensation revealed using illusory touch. *Nat Neurosci* 16:958-
965.
- Rossant C, Kadir SN, Goodman DF, Schulman J, Hunter ML, Saleem AB, Grosmark A, Belluscio
817 M, Denfield GH, Ecker AS, Tolias AS, Solomon S, Buzsaki G, Carandini M, Harris KD
(2016) Spike sorting for large, dense electrode arrays. *Nat Neurosci* 19:634-641.
- Sato TR, Schall JD (2003) Effects of stimulus-response compatibility on neural selection in frontal
eye field. *Neuron* 38:637-648.
- Schall JD (2009) Frontal eye fields. In: *Encyclopedia of neuroscience*, pp 1635-1638: Springer Berlin
Heidelberg.
- Song JH, McPeck RM (2010) Roles of narrow- and broad-spiking dorsal premotor area neurons in
reach target selection and movement production. *J Neurophysiol* 103:2124-2138.
- Stone M (1960) Models for choice-reaction time. *Psychometrika* 25:251-260.
- Thura D, Cisek P (2014) Deliberation and commitment in the premotor and primary motor cortex
during dynamic decision making. *Neuron* 81:1401-1416.
- Thura D, Cisek P (2016) On the difference between evidence accumulator models and the urgency
gating model. *J Neurophysiol* 115:622-623.
- Thura D, Beauregard-Racine J, Fradet CW, Cisek P (2012) Decision making by urgency gating:
theory and experimental support. *J Neurophysiol* 108:2912-2930.
- Wang M, Chandrasekaran C, Peixoto D, Newsome WT, Shenoy KV (2016) Dorsal premotor cortex
activity reflects a candidate decision variable during the action selection epoch of an abstract
perceptual decision-making task In: *Society For Neuroscience*. San Diego, CA.

Reviewers' Comments:

Reviewer #1 (Remarks to the Author):

NCOMMS-16-15643A

Summary

The manuscript is improved. There are still a few places where the text needs clarification, but I overall I recommend it for publication.

Things to address

line 347 & 390 (see details below)

Specific Comments

line 1461: Missing a word. "with"?

> We gently restrained the arm the monkey was not using ...

Figure 1C: Generally, the negative coherence (more green) should be to the left of the more positive coherence.

line 159: "than, the" remove comma

line 291: "Both left, and right" remove comma

line 303: The perimovement group is not defined in the same way as the other groups, since it is by definition the "leftovers". I wonder if the authors could "clean-up" that group? I recognize that this could be a bit complicated, but the techniques in Rouder et al (2009) could be of use. This may help to distinguish the decreased and the perimovement groups.

line 337: Not sure how you are estimating p in your permutation test but with a 2-tailed test and 10000 repeats your p should not be less than 1/5000. MY guess is that your data is completely outside of the shuffled distribution, so you put $p \sim 0$, but it should just be $2/(\# \text{ of shuffles})$.

line 347: Is that really true? Does this sentence even make sense? The RT is a property of the trial (the specific trajectory that was taken), so how can you say "choice ... signal should emerge

on average ... regardless of RT". I think you mean this for a given level of difficulty. That's important to say. But I still think this sentence is problematic. The DDM and UGM are behavioral models. They don't make strong predictions about how neural circuits instantiate the models. In order for this sentence to be meaningful a lot of unstated assumptions should be stated (such as the correlation between neurons and how the DV is read out.) Maybe easier to remove?

line 390: I think the models predict crossing a threshold, which is not the same as convergence (the neurons could converge to 0 FR. Would that support the models?). Even if convergence was sufficient, is absence of correlation evidence for convergence? In Murakami et al (2014) they do a lot of analyses to argue for a rise to threshold mechanism.

References

Rouder et al., Bayesian t tests for accepting and rejecting the null hypothesis. *Psychonomic Bulletin & Review* 2009, 16 (2), 225-237 doi:10.3758/PBR.16.2.225

Murakami et al. Neural antecedents of self-initiated actions in secondary motor cortex. *Nature Neuroscience* 17, 1574–1582 (2014) doi:10.1038/nn.3826

Reviewer #2 (Remarks to the Author):

In their resubmission, the authors have performed extensive and thorough additional analyses, which address all major concerns raised in the initial review. This has greatly clarified and strengthened the manuscript. I now find this manuscript suitable for publication.

Reviewer #3 (Remarks to the Author):

In my opinion, the authors have significantly improved the manuscript. In particular, all three reviewers requested additional analyses and made a variety of suggestions, and the authors have heroically implemented all of these. Furthermore, these additional analyses generally support the initial conclusions that used the simple visuomotor index. While some of these are not terribly strong (e.g. Extended Figs 5 and 7), some are very compelling (e.g. Extended Fig 6f and 8e), and all are very worthwhile. Thus, my initial concerns about the suitability of the visuomotor index are well-addressed, and the new analyses make this a stronger paper. However, as a consequence the manuscript is now very large. By my estimate, the main text is more than twice over the word limit of Nature Communications. I would recommend that the details of most of the new analyses be relegated into the online methods and supplemental materials, and only mentioned in the main text as generally confirming the conclusions gained by the visuomotor index. Don't get

me wrong – I like these new analyses and think they should remain (all the details can be in online methods and extended figures). I just think the reader should be allowed to move on more quickly to the main finding, the laminar trends.

A more serious flaw with the paper is the issue of DDM versus UGM. I understand the authors wanted to leave this in because they use modeling to estimate non-decision times. But what does it actually add? First of all, this issue is completely orthogonal to the real message of the manuscript, which quantifies the differences in cell types in different layers in PMd. That quantification is accomplished with the visuomotor index and all of the methods requested by the reviewers, none of which require a DDM or UGM or any other additional assumptions.

But second of all, the results of the model fits are incredibly inconclusive! The DDM works better for the early part of the RT distribution in error trials; the UGM better for the late part of RT distributions. The DDM is better for monkey T; the UGM is better for monkey O. So what are we to conclude? That animals sometimes use the DDM and make early errors, but as time passes they switch to a UGM? That different animals use different mechanisms? I'm sure the authors do not intend to make such proposals. In my opinion, the only thing that this model fitting exercise demonstrates is its own futility. There is no value in doing model fits to data from experiments that do not discriminate the models. Let's be honest here – these models are very simplified cartoons and surely nobody would expect them to capture the richness of the true (recurrent, non-linear) dynamics of the real brain. Thus, each of them captures a lot of the data but suffers “at the edges”. Comparing how much each suffers at each edge is pointless, in my opinion. All it does is add noise to an already noisy debate. A better approach is to use “strong inference” (Platt JR 1964 Science) – i.e., to design experiments that force the models to make different predictions. This has been done, and the authors cite some of the relevant papers.

Thus, I again strongly recommend that the discussion of DDM vs UGM is removed from the manuscript, at least from the main body. It would be useful to include the modeling analyses in the supplemental data, but only as an instructive demonstration of the futility of doing such model fits on this kind of data. I'm sorry but there is nothing else you can conclude from these. In the main text, the authors should remove all statements and conclusions based on these utterly inconclusive results.

A final concern I had was with the use of “multiunits”. The authors now report that single, well-isolated units made up 80% of their data, and multiunits only 20%. That's great. But then what is the point of including the multiunits? They only raise concerns (Lines 743-761) and I doubt that with all of the single units (N=881, which is excellent) you really need the additional power. Why not just include the nice clean data that you worked so hard to obtain? Do any of your results change or lose significance when you remove the 20% multiunits?

Specific comments:

Line 192-206: If you do choose to keep the model fitting in the supplemental, then you should give both models the same flexibility. For example, to produce early decisions, the urgency signal in the UGM needs to have both a slope and intercept parameter, otherwise it always starts at zero (Line 1259). Also, I'm confused why you say that the models have the same number of parameters (Line 1262). Shouldn't the UGM have at least one or two more? What are they?

Line 250: I was surprised that you did not find onset-related activity. Could it be because the target locations were always the same, so that onset was not informative? Nevertheless, the previous study cited (Song & McPeck 2010) did show activity related to cue onset, so the statement is incorrect (unless I misunderstood what you meant).

Line 314: The phrase "the accumulation of choice information" should be replaced with a more model-agnostic phrase such as "the build-up of neural activity in favor of a choice".

Line 339: Remove the phrase "are most consistent with the predictions of the DDM".

Line 392: XJ Wang's papers could be cited here as well.

Line 552: It is interesting that the average visuomotor index of the first cluster is negative. But what is the range? Do all or most cells in this cluster have negative indices? The same question applies to the other clusters. It would be useful to show the distributions of indices for each cluster (just in the supplemental figure).

Line 796: Indeed, in the tokens task all sensory evidence remains visible. But is that not also the case in the present task, in which the color checkerboard is always visible? Is that also not the case in the random-dot motion discrimination task, in which the motion (the thing the subject is deciding about) is always present in the stimulus? Perhaps I don't understand the distinction that is being made here, in which case, please explain.

Line 1755: You say "estimated" twice.

**Responses to reviewers for NCOMMS-16-15643B: “Laminar**
**differences in decision-related neural activity in dorsal**
**premotor cortex”**

We thank all three reviewers for the positive evaluation of the novelty of the results
presented here. We are also grateful to all the reviewers for the new comments and the
suggestions on the revised manuscript. In this re-revision, we have made every attempt to
address these comments and clarify the novelty as well as strength of the results presented in
the manuscript. Please find below the comments from the reviewers in **black** and our
response to the reviewers in **blue**. The text/analysis changes we have made in the
manuscript to respond to the reviewer comments are provided in **orange** in this document
(as well as in the manuscript).

**Reviewers' comments:**

**Reviewer #1 (Remarks to the Author):**

# NCOMMS-16-15643A

### Summary

The manuscript is improved. There are still a few places where the text needs clarification,
but I overall I recommend it for publication.

Thank you for the kind evaluation. The manuscript is much improved after your suggestions
for rewriting and analysis. We believe that we have now addressed the new comments
provided by you.

### Things to address

line 347 & 390 (see details below)

Please find our responses to these issues below.

### Specific Comments

**1. line 1461: Missing a word. "with"**

> We gently restrained the arm the monkey was not using ...

Thank you for identifying this typographical error. We have now corrected this sentence and
it now reads

We gently restrained the arm the monkey was not using with a plastic tube and cloth sling

2. **Figure 1C: Generally, the negative coherence (more green) should be to the left of**
**the more positive coherence.**

We thank the reviewer for this suggestion and have now changed Figure 1C.

3. **line 159: "than, the" remove comma,**
**line 291: "Both left, and right" remove comma**

We have now fixed these punctuation errors in the revised manuscript.

4. **line 303: The perimovement group is not defined in the same way as the other**
**groups, since it is by definition the "leftovers". I wonder if the authors could**
**"clean-up" that group? I recognize that this could be a bit complicated, but the**
**techniques in Rouder et al (2009) could be of use. This may help to distinguish**
**the decreased and the perimovement groups.**

As the reviewer has recognized, this is a difficult problem. As per the suggestion of the
reviewer, we examined the Bayes Factors provided by the Rouder toolbox and found that
our index and method of separation into these neural populations was quite robust (Supp.
Fig. 3b). The Bayes Factors for the perimovement groups were significantly different from
the Bayes Factors for the increased and decreased units (Supp. Fig. 3b). We found minimal
overlap between the perimovement and decreased groups in terms of Bayes Factors ruling
out significant contamination of the perimovement group by the decreased units and vice
versa. We have now included the following text in the methods section.

The perimovement units were defined as the ones with insignificant indices. We were
concerned that some of the decreased units, which had smaller values of the index, could be
mistakenly classified as perimovement units and vice versa. To address this concern, we used
the Bayes Factor method which provides the ratio of the likelihood of two competing
hypotheses or models⁶⁸ and is typically interpreted as evidence for one model over the other.
In our case the ratio is between classifying a unit as decreased (or increased, H_1) vs.
perimovement (H_0)⁶⁸. A large value of Bayes Factor suggests that model H_1 is more likely
than the model H_0 and in our case would provide strong support that the unit is correctly
classified as increased or decreased. In contrast, a low Bayes Factor would suggest strong
evidence for H_0 and support the classification of the unit as perimovement in nature. We
examined the Bayes Factors computed for the different broad unit categories and found that
the Bayes Factors for the units classified as perimovement based on the visuomotor index
were very low suggesting that they were correctly classified (Supp. Fig. 3b). The Bayes
Factors computed for perimovement units also had very little overlap with Bayes Factors for
both the decreased and increased units (Supp. Fig. 3b)⁶⁸. This suggests that there was
minimal contamination of the perimovement group by the decreased units and vice versa.

5. **line 337: Not sure how you are estimating p in your permutation test but with a 2-**
**tailed test and 10000 repeats your p should not be less then 1/5000. MY guess is**
**that your data is completely outside of the shuffled distrubution, so you put $p \sim 0$,**
**but it should just be $2/(\# \text{ of shuffles})$.**

Thank you for pointing this out. The data indeed were outside the shuffled distribution, and
we have used the formula of $2/(\# \text{ of shuffles})$ for the permutation statistics.

**6. line 347: Is that really true? Does this sentence even make sense? The RT is a**
**property of the trial (the specific trajectory that was taken), so how can you say**
**"choice ... signal should emerge on average ... regardless of RT". I think you**
**mean this for a given level of difficulty. That's important to say. But i still think**
**this sentence is problematic. The DDM and UGM are behavioral models. They**
**don't make strong predictions about how neural circuits instantiate the models.**
**In order for this sentence to be meaningful a lot of unstated assumptions should**
**be stated (such as the correlation between neurons and how the DV is read out.)**
**Maybe easier to remove?**

We thank the reviewer for alerting us to the conceptual complexities of this section in the
manuscript. Having considered it, we concur with the reviewer that this is unnecessary
preamble that has several assumptions about readout mechanisms built into it. We are only
making statements about the discrimination time of populations of neurons and individual
neurons. The readout mechanism is unknown. The paragraph now reads as follows.

We next examined the latency of the population level choice selectivity signal for each of the
broad unit categories. This latency which is termed *discrimination time* was defined as the first
time at which the choice selective signal significantly departed from the FR in the hold
period before the onset of the checkerboard cue as estimated using a paired t-test that
corrected for multiple comparisons¹. Error bars on discrimination time were obtained
through bootstrapping. Population level choice selectivity in increased units appeared ~100-
150 ms after checkerboard cue onset, and this discrimination time increased only modestly
with RT (Mean slope of regression (M) ± SE: 0.2 ± 0.06 ms/1 ms of RT, Figs. 3d-e, Extended
Data Fig. 4a-c). Furthermore, choice selectivity was observed earlier for the increased
compared to decreased and perimovement units. (Bootstrapped time estimates do not
overlap). In contrast to the weak relationship between discrimination time and RT for the
increased units, decreased and perimovement units exhibited a more pronounced increase in
discrimination time with RT (decreased: $M \pm SE = 0.53 \pm 0.15$ ms/1 ms of RT,
Perimovement: 0.4 ± 0.11 ms/1 ms of RT, Figs. 3e, Extended Data Fig. 4a-c). Again these
trends were observed separately in each monkey (Extended Data Figs. 4b, c).

**7. line 390: I think the models predict crossing a threshold, which is not the same as**
**convergence (the neurons could converge to 0 FR. Would that support the**
**models?). Even if convergence was sufficient, is absence of correlation evidence**
**for convergence? In Murakami et al (2014) they do a lot of analyses to argue for a**
**rise to threshold mechanism.**

136 **Convergence to 0 FR**

We thank the reviewer for this comment. This comment is actually very closely related to the
previous comment regarding readouts for decision-related activity. In principle, if the
readout mechanism involves inhibition of neurons in a downstream structure by PMd
neurons, then if firing rates in PMd go down to zero, it would allow the FR of the

downstream neuron to reach its threshold. In this case, our statements about convergence of
the firing rates would still be reasonable.

**Rise to Threshold Hypothesis**

However, it is unclear if the analyses performed in Murakami et al. (a reference
already cited in the manuscript) are relevant for the PMd data reported here. A threshold is
usually defined as a one dimensional mechanism for individual neurons, and it is unclear if in
PMd, a rise to threshold mechanism is operating, even in simple tasks that do not require
decision-making^{2, 3}. We believe it would be somewhat convoluted to argue for a rise-to-
threshold mechanism in PMd for decision-making but not in simple delayed-reach tasks. We
also emphasize that in this figure we are plotting the choice selective signal which is the
difference in firing rates and thus cannot rule out the presence or absence of a threshold per
se which in principle would operate at the level of firing rates. Finally, all three neuronal
types seem to show very similar properties. Whether the neurons actually rise to a threshold
does not materially change the conclusions we make here in this paper about laminar
differences in firing rate profiles in PMd during perceptual decisions.

The revised section from the paper is provided below.

**Magnitude of choice selective signal does not depend on coherence at the** 162 **time of movement initiation**

We next examined how FRs at the time of movement initiation depended on coherence and
RT. At the time of movement onset, regardless of broad unit category, the magnitude of the
average choice selective signal in the -100 ms to movement epoch was only weakly
modulated by color coherence/RT (For instance in the PMTHs shown in Figs. 2d-f; $M \pm SE$,
sign-test to compare median slopes for the average FR in -100 ms to move as a function of
color coherence and the slopes for a shuffled curve, increased: 0.30 ± 0.24 spks/s²/100%
color coherence, $p < 0.1$; decreased: 0.49 ± 0.33 spks/s²/100% color coherence, $p < .16$;
perimovement: 0.54 ± 0.25 spks/s²/100% color coherence, $p < 0.09$; Figs. 3g, h, Supp. Fig.
4j). These results were again observed in both monkeys (Supp. Fig. 4j, sign test, Monkey T:
Increased, $p < .07$, Decreased: $p < .14$, Perimove: 0.59; Monkey O: Increased, $p < 0.51$,
Decreased: $p < 0.67$, Perimove: $p < 0.37$).

This observation that the magnitude of the choice selective signal at the time of
movement onset only changes weakly with color coherence or RT is consistent with
proposals of a commitment state in the network prior to the initiation of the choice^{13, 53, 70, 71,}
⁷². In some contexts, additional analysis has been used to argue for a threshold implemented
by single neurons^{53, 73}, perhaps mimicking the thresholds used in behavioral models. Our
analysis showing a lack of correlation between the magnitude of the choice selective signal
(which is the difference in FRs for different reach directions) and coherence cannot be
considered as evidence for the rise-to-threshold mechanism in PMd for decisions. Previous
studies suggest that a rise-to-threshold mechanism provides a poor explanation of FRs of
PMd neurons even in simple delayed-reach tasks that do not involve perceptual decisions^{74,}
⁷⁵.

**Reviewer #2 (Remarks to the Author):**

In their resubmission, the authors have performed extensive and thorough additional
analyses, which address all major concerns raised in the initial review. This has greatly
clarified and strengthened the manuscript. I now find this manuscript suitable for
publication.

We thank reviewer 2 for the kind words and feedback on the revised manuscript and are
delighted that the manuscript has been recommended for publication.

Reviewer #3 (Remarks to the Author):

We thank you for all the suggestions that have greatly improved the paper. As a preamble, here are the concrete steps we have taken based on your advice from the comments shown below.

- Moved the entire modeling section to Supplementary Note 1 and Supp. Figs 1-2.
- Included an explicit UGM with both an intercept and slope term. This seems to provide the best description of the behavior of the monkeys albeit with one extra parameter.
- The many different analyses which we performed are now in Supp. Notes. 2-4 and in Supp. Figs. 3-12.
- The main results for just the single neurons are now shown in Supp. Fig. 13 and described in Supp. Note 5.
- We have now clarified the statements made in the paper about the Tokens task, the lack of responses to target onset and included box plots of the distributions of indices.

All of these manipulations have made the paper significantly shorter by removing nearly 5500 words from the main text. We hope this assuages any remaining concerns about the manuscript. Please find below our point by point response to your comments.

1. **In my opinion, the authors have significantly improved the manuscript. In particular, all three reviewers requested additional analyses and made a variety of suggestions, and the authors have heroically implemented all of these. Furthermore, these additional analyses generally support the initial conclusions that used the simple visuomotor index. While some of these are not terribly strong (e.g. Extended Figs 5 and 7), some are very compelling (e.g. Extended Fig 6f and 8e), and all are very worthwhile. Thus, my initial concerns about the suitability of the visuomotor index are well-addressed, and the new analyses make this a stronger paper.**

We thank you for the positive evaluation of our effort to make the analyses and conclusions watertight. Suggestions from all three reviewers were excellent, and we made every effort to address them and where possible perform the analysis.

2. **However, as a consequence the manuscript is now very large. By my estimate, the main text is more than twice over the word limit of Nature Communications. I would recommend that the details of most of the new analyses be relegated into the online methods and supplemental materials, and only mentioned in the main text as generally confirming the conclusions gained by the visuomotor index. Don't get me wrong – I like these new analyses and think they should remain (all the details can be in online methods and extended figures). I just think the reader should be allowed to move on more quickly to the main finding, the laminar trends.**

Thank you for this suggestion. We have now moved the entire section on the DDM vs.
UGM to the supplementary methods. We have also moved sections that use alternative
analyses to examine these FRs to the supplementary materials. Please find the text we wrote
in the manuscript below.

We also further investigated the behavior of the monkeys by fitting the RT
distributions and accuracy using the drift-diffusion (DDM) and urgency gating models
(UGM) that have been developed to explain behavior in two alternative forced choice tasks⁵⁴,
257 ^{55, 56, 57, 58} (Supp. Note 1 and Supp. Figs. 1,2). We performed this model-fitting analysis to
258 identify if these candidate computational frameworks could help us interpret decision-related
responses in PMd, and if the behavior was better explained by the DDM, estimate decision
260 times for the monkeys. In comparison to the well-studied random dots discrimination task³,
⁵⁹, quantitative modeling of how monkeys perform discrimination of static stimuli such as
the checkerboard used here^{55, 56} is lacking^{55, 60}. We found that, while both the UGM and the
DDM provided very reasonable fits, neither model was completely sufficient to describe the
RT and accuracy of the monkeys performing discrimination of static checkerboard stimuli
used here (Supp. Figs 1 and 2). The UGM with an intercept and slope term provided the
best fit of all three models considered here¹³. Our results here further highlight the increasing
realization that differentiating between these models of decision-making behavior using
purely statistical techniques is currently very difficult^{55, 56, 57, 58, 61, 62} — explicit stimulus
manipulations are necessary. Additional elaboration of these models is likely needed to better
describe the behavior of these monkeys in this static checkerboard discrimination task⁶³.

Choice selectivity is distributed in a continuum in this PMd neural population and is
not well described as clusters of high and low choice selectivity (Supp. Note 2, Supp. Fig.
4h). One concern is that the visuomotor index we use to partition and understand our large
dataset of neurons is too simplistic and collapses over many key features of the data. To
ensure that increased, decreased and perimovement units were not a spurious artifact
specific to the index that we developed here, we also tested if our results were consistent
when applying other methods for describing a visuomotor continuum^{12, 40} as well as a
novel technique reported by Erlich and collaborators⁶⁹ for estimating a correlation between
RT and neural responses (Supp. Note 3, Supp. Figs 5-9). These results were also not
explained as a simple effect of differences in the kinematics of the eventual arm movement
(e.g. speed) or eye position (Supp. Note 4, Supp Fig. 10). Finally, the conclusions did not
change even when we only analyzed the single neuron responses (Supp. Note 5, Supp. Fig.
13).

**3. A more serious flaw with the paper is the issue of DDM versus UGM. I**
**understand the authors wanted to leave this in because they use modeling to**
**estimate non-decision times. But what does it actually add? First of all, this issue**
**is completely orthogonal to the real message of the manuscript, which quantifies**
**the differences in cell types in different layers in PMd. That quantification is**
**accomplished with the visuomotor index and all of the methods requested by the**
**reviewers, none of which require a DDM or UGM or any other additional**
**assumptions.**

But second of all, the results of the model fits are incredibly inconclusive! The
DDM works better for the early part of the RT distribution in error trials; the
UGM better for the late part of RT distributions. The DDM is better for monkey
297 T; the UGM is better for monkey O. So what are we to conclude? That animals
sometimes use the DDM and make early errors, but as time passes they switch to
a UGM? That different animals use different mechanisms? I'm sure the authors
do not intend to make such proposals. In my opinion, the only thing that this
model fitting exercise demonstrates is its own futility. There is no value in doing
model fits to data from experiments that do not discriminate the models. Let's be
honest here – these models are very simplified cartoons and surely nobody would
expect them to capture the richness of the true (recurrent, non-linear) dynamics
of the real brain. Thus, each of them captures a lot of the data but suffers “at the
edges”. Comparing how much each suffers at each edge is pointless, in my
opinion. All it does is add noise to an already noisy debate. A better approach is
to use “strong inference” (Platt JR 1964 Science) – i.e., to design experiments that
force the models to make different predictions. This has been done, and the
authors cite some of the relevant papers.

Thus, I again strongly recommend that the discussion of DDM vs UGM is
removed from the manuscript, at least from the main body. It would be useful to
include the modeling analyses in the supplemental data, but only as an
instructive demonstration of the futility of doing such model fits on this kind of
data. I'm sorry but there is nothing else you can conclude from these. In the main
text, the authors should remove all statements and conclusions based on these
utterly inconclusive results.

We thank the reviewer for this recommendation and have now moved this entire section in
the results to the Supplementary Materials. We left all the analyses in the main text for the
previous revision because the reviewer found our earlier manuscript which only used the
DDM to be inadequate. The feedback provided by you helped us even more critically
evaluate our behavioral data: We are now using the lens provided by the UGM and
appreciate even further the role urgency has in the behavior of the monkeys performing
these types of tasks. We have now taken on board your suggestion and placed all of these
results in a supplementary note (Supp. Note 1). We only refer to them briefly in the main
text. The revised text is available in the answer to question 2 posed by you.

4. A final concern I had was with the use of “multiunits”. The authors now
report that single, well-isolated units made up 80% of their data, and
multiunits only 20%. That's great. But then what is the point of including the
multiunits? They only raise concerns (Lines 743-761) and I doubt that with all
of the single units (N=881, which is excellent) you really need the additional
power. Why not just include the nice clean data that you worked so hard to
obtain? Do any of your results change or lose significance when you remove
the 20% multiunits?

We are circumspect about excluding neural responses. As we said in the manuscript, we were
concerned that we did not have enough power to investigate laminar differences. However,
none of our results changed when only using the single units.

We now note in the results and the methods section that none of the conclusions change when excluding the multiunits from key analyses and have now created a new supplementary note (Supp. Note 5) and a Supplementary Figure (Supp. Fig. 13) that details several key results (dependence on coherence, discrimination time vs. RT, discrimination time for the three broad neuronal categories, scatter plot of discrimination time vs. index, laminar distribution of the visuomotor index, and discrimination time as a function of cortical depth) for just the single units.

Methods statement:

All main claims made in the paper were unchanged when only restricting the analysis to single neurons (Supp. Fig. 13 and Supp. Note 5).

Supplementary Note 5: Conclusions do not change when restricting analyses to just isolated single neurons

We also confirmed that our results were not influenced by the inclusion of the multi-units in the database. We examined these effects in four different key analyses that we report in the main results. We focused on the results describing the dependence of slopes on coherence, discrimination time, laminar distribution of the index and discrimination time as a function of cortical depth. None of the conclusions change when only considering the isolated single units (Supp. Fig. 13a-f).

First, the dependence on coherence was stronger for the increased compared to the decreased and perimovement neurons (Supp. Fig. 13a, Increased: 45.29 (3.59), Decreased: 25.38 (3.33), Perimovement: 19.09 (1.99), Increased vs. Decreased: Wilcoxon ranksum $p = 5.62e-03$, increased vs. perimovement: Wilcoxon ranksum $p = 2.69e-07$, decreased vs. perimovement: Wilcoxon ranksum $p = 0.2$, correlation between visuomotor index and slope: spearman's $r = 0.33$, $p < 4.77e-22$). Second, increased neurons in PMd signaled the choice earlier than the decreased and perimovement neurons (Supp. Figs. 13b-d, Correlation between visuomotor index and discrimination time for all RTs from 300 ms to 1s: Spearman's $r=0.36$, $p < 3.47e-25$; Increased vs. Decreased: ranksum $p=1.23e-10$, increased vs. perimove: $p < 8.91e-12$). Third, the visuomotor index again showed the same dependence as a function of cortical depth for both single neurons and multi-units (Supp. Fig. 13e, goodness of fit for single neurons: $R^2 = 0.88$, $p < 2e-3$, 1000 shuffles). Finally, the discrimination time also increases as a function of cortical depth (Supp. Fig. 13f, $r=0.69$, $p < 0.0042$). Together, the results for single neurons are largely consistent with the results reported including both single neurons and multiunits in PMd.

**Specific comments**

- **5. Line 192-206: If you do choose to keep the model fitting in the supplemental,**
- **then you should give both models the same flexibility. For example, to**
- **produce early decisions, the urgency signal in the UGM needs to have both a**
- **slope and intercept parameter, otherwise it always starts at zero (Line 1259).**
- **Also, I'm confused why you say that the models have the same number of**
- **parameters (Line 1262). Shouldn't the UGM have at least one or two more?**
- **What are they?**

We thank the reviewer for the suggestion. We have now implemented the new model

incorporating an intercept term for the urgency signal. This was an excellent suggestion as it

indeed improved the model and now the urgency gating model provides the most consistent

explanation for both monkeys. However, the urgency gating model now has additional

parameters compared to the DDM. We did not want to further go into the intricacies of

model comparison and have just made a statement that the UGM provides the best

description of the behavior of the monkeys.

We have now clarified in a table about the number of parameters for the two models.

What we meant to say is that an equivalent number of parameters are estimated from the

two models (UGM without an intercept and the DDM), and thus they could be compared

without penalization of one model or the other (Table 4). However, in the revised

manuscript the addition of an intercept term for the UGM provides it with one additional

parameter and the fit is improved and now consistent across both monkeys.

- **6. Line 250: I was surprised that you did not find onset-related activity. Could it**
- **be because the target locations were always the same, so that onset was not**
- **informative? Nevertheless, the previous study cited (Song & McPeck 2010)**
- **did show activity related to cue onset, so the statement is incorrect (unless I**
- **misunderstood what you meant).**

From our reading of the Fig. 2B of Song and McPeck 2010 paper and the corresponding text

in page 2127 of the published paper, they make the following statements. The bolding is our

emphasis.

“Figure 2 shows the activity of representative a cell in both tasks. Activity is aligned on the

presentation of the search array, and reach onset is indicated in each trial by a thick tick

mark. In the neuron on the left, activity shows a transient increase after search array onset

when the target was in the PD. In the single-target task (Fig. 2A), the cell shows a prominent

burst of activity in the first 100 ms after the onset of a target in its PD. However, in search

(Fig. 2B), this initial visual activity is substantially reduced, despite the fact that the position

and visual properties of the target in the PD were identical in the two tasks.

Similar results are seen in the mean population activity across our sample of neurons in the

single-target (Fig. 2C) and search tasks (Fig. 2D). **This reduction in the initial burst of**

**activity occurs despite the fact that monkeys are free to respond immediately in the**

**search task, whereas they must wait to respond in the single-target delay task. A**

similar reduction in the initial visual response for search tasks has been found for the
oculomotor system (Basso and Wurtz 1998; McPeck and Keller 2002; Schall et al. 1995).”

The way we interpret this result from Song and McPeck 2010 is that in one-target tasks you
can see robust delay period and modulation to target onset but not in two target tasks at least
in this type of task context and our recordings. We have two hypotheses about the lack of
neural responses to target onset in our task.

First, we suspect that the weak responses to target onset in our case might be
because our recordings were performed in more Caudal regions of PMd and not Rostral
PMd. As Figure 6B from Cisek and Kalaska 2005²⁴ shows, target related responses in Caudal
PMd were far more modest (~3 spikes/s) than the responses in rostral PMd (~7 spikes/s).
Moreover, the potential response cells^{24,25} are the ones which respond to the target onset and
are found most commonly in the rostral part of PMd (Fig. 5. Of ²⁴) Most of our recordings
were done in the caudal locations in PMd (Stereotaxic coordinates +16, +15). In over a
hundred sessions of recordings from caudal PMd, we have not seen activity in response to
target onset.

Second, there are legitimate task differences between ref. ²⁴ and the task we use here.
In the 2-Target task used by ²⁴, the monkey is very sure that an unambiguous color cue that
allows it to select the correct target with near 100% success rates will be available to it. This
might encourage the monkey to co-activate both populations of neurons in response to
target onset or behaviorally co-plan to potential reach targets. In contrast, in our task in
nearly (5/7) cases the animal is not sure he will receive an unambiguous color cue. This may
discourage the co-activation of neural populations and co-planning of movements. Of
course, all of this is just conjecture on our part.

We now qualify this statement by adding the suggestion from the reviewer and have
rewritten this paragraph.

Caudal PMd units did not typically modulate their FRs to target onset in this task
(data not shown). Modest changes in firing rates in response to target onset might be either a
result of the targets always being shown at predictable spatial locations. Other studies have
suggested that suppressive effects on FRs may occur when there is movement uncertainty
due to the presence of multiple targets^{12,67}.

**7. Line 314: The phrase “the accumulation of choice information” should be**
**replaced with a more model-agnostic phrase such as “the build-up of neural**
**activity in favor of a choice”.**

We thank you for this comment. We have now replaced it with the suggested text.

**8. Line 339: Remove the phrase “are most consistent with the predictions of the**
**DDM”.**

We have removed this line from the manuscript.

**9. Line 392: XJ Wang’s papers could be cited here as well.**

We have included citations to papers from XJ Wang’s papers in the relevant location of the manuscript. We have also clarified the interpretation of these results because activity in PMd is unlikely to be explained simply by a rise-to-threshold mechanism.

10. Line 552: It is interesting that the average visuomotor index of the first cluster is negative. But what is the range? Do all or most cells in this cluster have negative indices? The same question applies to the other clusters. It would be useful to show the distributions of indices for each cluster (just in the supplemental figure).

Fig. 3a and Supp. Figs. 3a, b show the distribution of the index for each of the monkeys already. By definition, all cells in the decreased cluster have negative perimovement indices because we tested whether the index is significantly different from zero. We have also incorporated on the basis of suggestions from reviewer 1, a distribution of the Bayes Factors for the three broad categories. Our method of separation was able to broadly separate out these neural populations.

11. Line 796: Indeed, in the tokens task all sensory evidence remains visible. But is that not also the case in the present task, in which the color checkerboard is always visible? Is that also not the case in the random-dot motion discrimination task, in which the motion (the thing the subject is deciding about) is always present in the stimulus? Perhaps I don’t understand the distinction that is being made here, in which case, please explain.

We note that the three tasks have different features.

- Random dot motion, only momentary evidence is available at each frame. The integrated sensory evidence (if computed) is only available in the brain of the monkey. Similarly, in the clicks task devised in ²⁷, the integrated evidence is available only in the brain of the rats or the humans, the momentary evidence is provided by each click.
- For the checkerboard discrimination task used here, the momentary evidence is the same as the overall evidence at every time point.
- For the tokens task, overall evidence in favor of the choices are available at each point in time in the form of spatially separated sets of tokens, momentary evidence then is available at each time point.

We have clarified these sentences to read:

One purported limitation of the “tokens” task design is that the stimulus by itself provides the integral of the sensory evidence²⁸. This is in comparison to the random-dots stimuli where only the momentary evidence is available and not the integral of the sensory evidence. However, recent efforts in humans have addressed this issue by using a variant of the random dot motion discrimination task and suggesting that the UGM might apply in other stimulus contexts as well¹⁴.

**12. Line 1755: You say “estimated” twice.**

We thank you for alerting us to this typographical error. This is fixed in the revised
manuscript.

**References**

- 1. Sato TR, Schall JD. Effects of stimulus-response compatibility on neural selection in
frontal eye field. *Neuron* **38**, 637-648 (2003).
- 2. Churchland MM, Cunningham JP, Kaufman MT, Ryu SI, Shenoy KV. Cortical
preparatory activity: representation of movement or first cog in a dynamical
machine? *Neuron* **68**, 387-400 (2010).
- 3. Churchland MM, Santhanam G, Shenoy KV. Preparatory activity in premotor and
motor cortex reflects the speed of the upcoming reach. *J Neurophysiol* **96**, 3130-3146
(2006).
- 4. Thura D, Cisek P. Deliberation and commitment in the premotor and primary motor
cortex during dynamic decision making. *Neuron* **81**, 1401-1416 (2014).
- 5. Wang XJ. Probabilistic decision making by slow reverberation in cortical circuits.
*Neuron* **36**, 955-968 (2002).
- 6. Wang XJ. Decision making in recurrent neuronal circuits. *Neuron* **60**, 215-234 (2008).
- 7. Wang XJ. Neural dynamics and circuit mechanisms of decision-making. *Curr Opin*
*Neurobiol* **22**, 1039-1046 (2012).
- 8. Roitman JD, Shadlen MN. Response of neurons in the lateral intraparietal area
during a combined visual discrimination reaction time task. *J Neurosci* **22**, 9475-9489
(2002).
- 9. Murakami M, Vicente MI, Costa GM, Mainen ZF. Neural antecedents of self-
initiated actions in secondary motor cortex. *Nat Neurosci* **17**, 1574-1582 (2014).
- 10. Ratcliff R, McKoon G. The diffusion decision model: theory and data for two-choice
decision tasks. *Neural Comput* **20**, 873-922 (2008).
- 11. Hawkins GE, Forstmann BU, Wagenmakers EJ, Ratcliff R, Brown SD. Revisiting
the evidence for collapsing boundaries and urgency signals in perceptual decision-
making. *J Neurosci* **35**, 2476-2484 (2015).
- 12. Hawkins GE, Wagenmakers EJ, Ratcliff R, Brown SD. Discriminating evidence
accumulation from urgency signals in speeded decision making. *J Neurophysiol* **114**, 40-
47 (2015).
- 13. Carland MA, Thura D, Cisek P. The urgency-gating model can explain the effects of
early evidence. *Psychon Bull Rev* **22**, 1830-1838 (2015).
- 14. Carland MA, Marcos E, Thura D, Cisek P. Evidence against perfect integration of
sensory information during perceptual decision making. *J Neurophysiol* **115**, 915-930
(2016).
- 15. Palmer J, Huk AC, Shadlen MN. The effect of stimulus strength on the speed and
accuracy of a perceptual decision. *J Vis* **5**, 376-404 (2005).
- 16. de Lafuente V, Jazayeri M, Shadlen MN. Representation of accumulating evidence
for a decision in two parietal areas. *J Neurosci* **35**, 4306-4318 (2015).
- 17. Middlebrooks PG, Schall JD. Response inhibition during perceptual decision making
in humans and macaques. *Atten Percept Psychophys* **76**, 353-366 (2014).
- 18. Boehm U, Hawkins GE, Brown S, van Rijn H, Wagenmakers EJ. Of monkeys and
men: Impatience in perceptual decision-making. *Psychon Bull Rev* **23**, 738-749 (2016).
- 19. Thura D, Cisek P. On the difference between evidence accumulator models and the
urgency gating model. *J Neurophysiol* **115**, 622-623 (2016).
- 20. Purcell BA, Kiani R. Neural Mechanisms of Post-error Adjustments of Decision
Policy in Parietal Cortex. *Neuron* **89**, 658-671 (2016).

- 21. DiCarlo JJ, Maunsell JH. Using neuronal latency to determine sensory-motor
processing pathways in reaction time tasks. *J Neurophysiol* **93**, 2974-2986 (2005).
- 22. Song JH, McPeck RM. Roles of narrow- and broad-spiking dorsal premotor area
neurons in reach target selection and movement production. *J Neurophysiol* **103**, 2124-
2138 (2010).
- 23. Erlich JC, Bialek M, Brody CD. A cortical substrate for memory-guided orienting in
the rat. *Neuron* **72**, 330-343 (2011).
- 24. Cisek P, Kalaska JF. Neural correlates of reaching decisions in dorsal premotor
cortex: specification of multiple direction choices and final selection of action.
*Neuron* **45**, 801-814 (2005).
- 25. Coallier E, Michelet T, Kalaska JF. Dorsal premotor cortex: neural correlates of
reach target decisions based on a color-location matching rule and conflicting
sensory evidence. *J Neurophysiol* **113**, 3543-3573 (2015).
- 26. Basso MA, Wurtz RH. Modulation of neuronal activity by target uncertainty. *Nature*
**389**, 66-69 (1997).
- 27. Brunton BW, Botvinick MM, Brody CD. Rats and humans can optimally accumulate
evidence for decision-making. *Science* **340**, 95-98 (2013).
- 28. Winkel J, Keuken MC, van Maanen L, Wagenmakers EJ, Forstmann BU. Early
evidence affects later decisions: why evidence accumulation is required to explain
response time data. *Psychon Bull Rev* **21**, 777-784 (2014).

Reviewers' Comments:

Reviewer #1 (Remarks to the Author):

I commend the authors on their excellent revisions.
I recommend the paper for publication.

Reviewer #3 (Remarks to the Author):

The revised manuscript responds very well to all of my comments. The comparison of the full data set to the data restricted to single units (Supp. Fig. 13) is very persuasive, and lays to rest my concerns that inclusion of multi-unit data might have been problematic. The DDM-vs-UGM discussion is now mostly in the supplemental materials, so it does not distract from the main message of the paper. The authors also provide a nice response to my question regarding the data of Song & McPeck. Overall, I think this is an excellent paper and support its publication in its present form.

REVIEWER COMMENTS

Reviewer #1 (Remarks to the Author):

I commend the authors on their excellent revisions.
I recommend the paper for publication.

Reviewer #3 (Remarks to the Author):

The revised manuscript responds very well to all of my comments. The comparison of the full data set to the data restricted to single units (Supp. Fig. 13) is very persuasive, and lays to rest my concerns that inclusion of multi-unit data might have been problematic. The DDM-vs-UGM discussion is now mostly in the supplemental materials, so it does not distract from the main message of the paper. The authors also provide a nice response to my question regarding the data of Song & McPeck. Overall, I think this is an excellent paper and support its publication in its present form.

We thank both reviewers for the kind comments and are delighted that the paper is suitable for publication in its present form.